# Generalization Error Bounds on Deep Learning with Markov Datasets

**Lan V. Truong**[*]
Department of Engineering
University of Cambridge
Cambridge, CB2 1PZ
lt407@cam.ac.uk

## Abstract

In this paper, we derive upper bounds on generalization errors for deep neural networks with Markov datasets. These bounds are developed based on Koltchinskii and Panchenko's approach for bounding the generalization error of combined classifiers with i.i.d. datasets. The development of new symmetrization inequalities in high-dimensional probability for Markov chains is a key element in our extension, where the absolute spectral gap of the infinitesimal generator of the Markov chain plays a key parameter in these inequalities. We also propose a simple method to convert these bounds and other similar ones in traditional deep learning and machine learning to Bayesian counterparts for both i.i.d. and Markov datasets. Extensions to $m$-order homogeneous Markov chains such as AR and ARMA models and mixtures of several Markov data services are given.

## 1 Introduction

In statistical learning theory, understanding generalization for neural networks is among the most challenging tasks. The standard approach to this problem was developed by Vapnik [1], and it is based on bounding the difference between the prediction error and the training error. These bounds are expressed in terms of the so called VC-dimension of the class. However, these bounds are very loose when the VC-dimension of the class can be very large, or even infinite. In 1998, several authors [2, 3] suggested another class of upper bounds on generalization error that are expressed in terms of the empirical distribution of the margin of the predictor (the classifier). Later, Koltchinskii and Panchenko [4] proposed new probabilistic upper bounds on generalization error of the combination of many complex classifiers such as deep neural networks. These bounds were developed based on the general results of the theory of Gaussian, Rademacher, and empirical processes in terms of general functions of the margins, satisfying a Lipschitz condition. They improved previously known bounds on generalization error of convex combination of classifiers.

In the context of supervised classification, PAC-Bayesian bounds have proved to be the tightest [5–7]. Several recent works have focused on gradient descent based PAC-Bayesian algorithms, aiming to minimise a generalisation bound for stochastic classifiers [8–10]. Most of these studies use a surrogate loss to avoid dealing with the zero-gradient of the misclassification loss. Several authors used other methods to estimate of the misclassification error with a non-zero gradient by proposing new training algorithms to evaluate the optimal output distribution in PAC-Bayesian bounds analytically [11–13]. Recently, there have been some interesting works which use information-theoretic approach to find PAC-bounds on generalization errors for machine learning [14, 15] and deep learning [16].

---

[*]Use footnote for providing further information about author (webpage, alternative address)—*not* for acknowledging funding agencies.

All of the above-mentioned bounds are derived based on the assumption that the dataset is generated by an i.i.d. process with unknown distribution. However, in many applications in machine learning such as speech, handwriting, gesture recognition, and bio-informatics, the samples of data are usually correlated. Some of these datasets are time-series ones with stationary distributions such as samples via MCMC, finite-state random walks, or random walks on graph. In this work, we develop some upper bounds on generalization errors for deep neural networks with Markov or hidden Markov datasets. Our bounds are derived based on the same approach as Koltchinskii and Panchenko [4]. To deal with the Markov structure of the datasets, we need to develop some new techniques in this work. The development of new symmetrization inequalities in high-dimensional probability for Markov chains is a key element in our extension, where the absolute spectral gap of the infinitesimal generator of the Markov chain plays as a key parameter in these inequalities. Furthermore, we also apply our results to $m$-order Markov chains such as AR and ARMA models and mixtures of Markov chains. Finally, a simple method to convert all our bounds for traditional deep learning to counterparts for Bayesian deep learning is given. Our method can be applied to convert other similar bounds for i.i.d. datasets in the research literature as well. Bayesian deep learning was introduced by [20, 21]. The key distinguishable property of a Bayesian approach is marginalization, rather than using a single setting of weights in (traditional) deep learning [22].

Bayesian marginalization can particularly improve the accuracy and calibration of modern deep neural networks, which are typically underspecified by the data, and can represent many compelling but different solutions. Analysis of machine learning algorithms for Markov and Hidden Markov datasets already appeared in research literature [17–19]. In practice, some real-world time-series datasets are not stationary Markov chains. However, we can approximate time-series datasets by stationary Markov chains in many applications. There are also some other methods of approximating non-stationary Markov chains by stationary ones via MA and ARMA models in the statistical research literature. The i.i.d. dataset is a special case of the Markov dataset with stationary distribution.

## 2 Preliminaries

### 2.1 Mathematical Backgrounds

Let a Markov chain $\{X_n\}_{n=1}^{\infty}$ on a state space $\mathcal{S}$ with transition kernel $Q(x, dy)$ and the initial state $X_1 \sim \nu$, where $\mathcal{S}$ is a Polish space in $\mathbb{R}$. In this paper, we consider the Markov chains which are irreducible and positive-recurrent, so the existence of a stationary distribution $\pi$ is guaranteed. An irreducible and recurrent Markov chain on an infinite state-space is called Harris chain [23]. A Markov chain is called *reversible* if the following detailed balance condition is satisfied:

$$\pi(dx)Q(x, dy) = \pi(dy)Q(y, dx), \qquad \forall x, y \in \mathcal{S}. \tag{1}$$

Define

$$d(t) = \sup_{x \in \mathcal{S}} d_{\text{TV}}(Q^t(x, \cdot), \pi), \qquad t_{\text{mix}}(\varepsilon) := \min\{t : d(t) \leq \varepsilon\}, \tag{2}$$

and

$$\tau_{\min} := \inf_{0 \leq \varepsilon \leq 1} t_{\text{mix}}(\varepsilon)\left(\frac{2-\varepsilon}{1-\varepsilon}\right)^2, \qquad t_{\text{mix}} := t_{\text{mix}}(1/4). \tag{3}$$

Let $L_2(\pi)$ be the Hilbert space of complex valued measurable functions on $\mathcal{S}$ that are square integrable w.r.t. $\pi$. We endow $L_2(\pi)$ with inner product $\langle f, g \rangle := \int fg^* d\pi$, and norm $\|f\|_{2,\pi} := \langle f, f \rangle_{\pi}^{1/2}$. Let $E_\pi$ be the associated averaging operator defined by $(E_\pi)(x, y) = \pi(y), \forall x, y \in \mathcal{S}$, and

$$\lambda = \|Q - E_\pi\|_{L_2(\pi) \to L_2(\pi)}, \tag{4}$$

where $\|B\|_{L_2(\pi) \to L_2(\pi)} = \max_{v:\|v\|_{2,\pi}=1} \|Bv\|_{2,\pi}$. $Q$ can be viewed as a linear operator (infinitesimal generator) on $L_2(\pi)$, denoted by $\mathbf{Q}$, defined as $(\mathbf{Q}f)(x) := \mathbb{E}_{Q(x, \cdot)}(f)$, and the reversibility is equivalent to the self-adjointness of $\mathbf{Q}$. The operator $\mathbf{Q}$ acts on measures on the left, creating a measure $\mu\mathbf{Q}$, that is, for every measurable subset $A$ of $\mathcal{S}$, $\mu\mathbf{Q}(A) := \int_{x \in \mathcal{S}} Q(x, A)\mu(dx)$. For a Markov chain with stationary distribution $\pi$, we define the *spectrum* of the chain as

$$S_2 := \left\{\xi \in \mathbb{C} : (\xi\mathbf{I} - \mathbf{Q}) \text{ is not invertible on } L_2(\pi)\right\}. \tag{5}$$

It is known that $\lambda = 1 - \gamma^*$ [24], where

$$\gamma^* := \begin{cases} 1 - \sup\{|\xi| : \xi \in \mathcal{S}_2, \xi \neq 1\}, \\ \qquad \text{if eigenvalue 1 has multiplicity 1,} \\ 0, \qquad \text{otherwise} \end{cases}$$

is the *the absolute spectral gap* of the Markov chain. The absolute spectral gap can be bounded by the mixing time $t_{\mathrm{mix}}$ of the Markov chain by the following expression:

$$\left( \frac{1}{\gamma^*} - 1 \right) \log 2 \leq t_{\mathrm{mix}} \leq \frac{\log(4/\pi_*)}{\gamma_*}, \tag{6}$$

where $\pi_* = \min_{x \in \mathcal{S}} \pi_x$ is the *minimum stationary probability*, which is positive if $Q^k > 0$ (entry-wise positive) for some $k \geq 1$. See [25] for more detailed discussions. In [25, 26], the authors provided algorithms to estimate $t_{\mathrm{mix}}$ and $\gamma^*$ from a single trajectory.

Define

$$\mathcal{M}_2 := \left\{ \nu \in \mathcal{M}(\mathcal{S}) : \left\| \frac{d\nu}{d\pi} \right\|_2 < \infty \right\}, \tag{7}$$

where $\| \cdot \|_2$ is the standard $L_2$ norm in the Hilbert space of complex valued measurable functions on $\mathcal{S}$.

## 2.2 Problem settings

In this paper, we consider a uniformly bounded class of functions: $\mathcal{F} := \{ f : \mathcal{S} \to \mathbb{R} \}$ such that $\sup_{f \in \mathcal{F}} \|f\|_\infty \leq M$ for some finite constant $M$.

Define the probability measure $P(A) := \int_A \pi(x) dx$, for any measurable set $A \in \mathcal{S}$. In addition, let $P_n$ be the empirical measure based on the sample $(X_1, X_2, \cdots, X_n)$, i.e., $P_n := \frac{1}{n} \sum_{i=1}^n \delta_{X_i}$. We also denote $Pf := \int_{\mathcal{S}} f dP$ and $P_n f := \int_{\mathcal{S}} f dP_n$. Then, we have

$$Pf = \int_{\mathcal{S}} f(x) \pi(x) dx \quad \text{and} \quad P_n f = \frac{1}{n} \sum_{i=1}^n f(X_i). \tag{8}$$

On the Banach space of uniformly bounded functions $\mathcal{F}$, define an infinity norm: $\|Y\|_{\mathcal{F}} = \sup_{f \in \mathcal{F}} |Y(f)|$. Let

$$G_n(\mathcal{F}) := \mathbb{E}\left[ \left\| n^{-1} \sum_{i=1}^n g_i \delta_{X_i} \right\|_{\mathcal{F}} \right], \tag{9}$$

where $\{g_i\}$ is a sequence of i.i.d. standard normal variables, independent of $\{X_i\}$. We will call $n \mapsto G_n(\mathcal{F})$ *the Gaussian complexity function* of the class $\mathcal{F}$.

Similarly, we define

$$R_n(\mathcal{F}) := \mathbb{E}\left[ \left\| n^{-1} \sum_{i=1}^n \varepsilon_i \delta_{X_i} \right\|_{\mathcal{F}} \right], \tag{10}$$

and

$$P_n^0 := n^{-1} \sum_{i=1}^n \varepsilon_i \delta_{X_i}, \tag{11}$$

where $\{\varepsilon_i\}$ is a sequence of i.i.d. Rademacher (taking values $+1$ and $-1$ with probability $1/2$ each) random variables, independent of $\{X_i\}$. We will call $n \mapsto R_n(\mathcal{F})$ *the Rademacher complexity function* of the class $\mathcal{F}$.

For times-series datasets in machine learning, we can assume that feature vectors are generated by a Markov chain $\{X_n\}_{n=1}^\infty$ with stochastic matrix $Q$, and $\{Y_n\}_{n=1}^\infty$ is the corresponding sequence of labels. Furthermore, $Q$ is irreducible and recurrent on some finite set $\mathcal{S}$. An i.i.d. sequence of

feature vectors can be considered as a special Markov chain where $Q(x, x')$ only depends on $x'$. In the supervised learning, the sequence of labels $\{Y_n\}_{n=1}^{\infty}$ can be considered as being generated by a Hidden Markov Model (HMM), where the emission probability $P_{Y_n|X_n}(y|x) = g(x, y)$ for all $n \geq 1$ and $g : \mathcal{S} \times \mathcal{Y} \to \mathbb{R}_+$. It is easy to see that $\{(X_n, Y_n)\}_{n=1}^{\infty}$ is a Markov chain with the transition probability

$$P_{X_{n+1}Y_{n+1}|X_nY_n}(x_{n+1}, y_{n+1}|x_n, y_n) = Q(x_n, x_{n+1})g(x_{n+1}, y_{n+1}). \tag{12}$$

Let $\tilde{Q}(x_1, y_1, x_2, y_2) := Q(x_1, x_2)g(x_2, y_2)$ for all $x_1, x_2 \in \mathcal{S}$ and $y_1, y_2 \in \mathcal{Y}$, which is the transition probability of the Markov chain $\{(X_n, Y_n)\}_{n=1}^{\infty}$ on $\tilde{S} := \mathcal{S} \times \mathcal{Y}$. Then, it is not hard to see that $\{(X_n, Y_n)\}_{n=1}^{\infty}$ is irreducible and recurrent on $\tilde{S}$, so it has a stationary distribution, say $\tilde{\pi}$. The associated following probability measure is defined as

$$P(A) := \int_{\mathcal{S} \times \mathcal{Y}} \tilde{\pi}(x, y)dxdy, \tag{13}$$

and the empirical distribution $P_n$ based on the observations $\{(X_k, Y_k)\}_{k=1}^{n}$ is

$$P_n := \frac{1}{n} \sum_{k=1}^{n} \delta_{X_k, Y_k}. \tag{14}$$

## 2.3 Contributions

In this paper, we aim to develop a set of novel upper bounds on *the generalization errors* for deep neural networks with Markov dataset. More specially, our target is to find a relationship between $Pf$ and $P_n f$ which holds for all $f \in \mathcal{F}$ in terms of Gaussian and Rademacher complexities. Our main contributions include:

- We develop general bounds on generalization errors for machine learning (and deep learning) on Markov datasets.

- Since the dataset is non-i.i.d., the standard symmetrization inequalities in high-dimensional probability can not be applied. In this work, we extend some symmetrization inequalities for i.i.d. random processes to Markov ones.

- We propose a new method to convert all the bounds for machine learning (and deep learning) models to Bayesian settings.

- Extensions to $m$-order homogeneous Markov chains such as AR and ARMA models and mixtures of several Markov services are given.

## 3 Main Results

### 3.1 Probabilistic Bounds for General Function Classes

In this section, we develop probabilistic bounds for general function classes in terms of Gaussian and Rademacher complexities.

First, we prove the following key lemma, which is an extension of the symmetrization inequality for i.i.d. sequences (for example, [27]) to a new version for Markov sequences $\{X_n\}_{n=1}^{\infty}$ with the stationary distribution $\pi$ and the initial distribution $\nu \in \mathcal{M}_2$:

**Lemma 1.** *Let $\mathcal{F}$ be a class of functions which are uniformly bounded by $M$. For all $n \in \mathbb{Z}^+$, define*

$$A_n := \sqrt{\frac{2M}{n(1-\lambda)} + \frac{64M^2}{n^2(1-\lambda)^2}\left\|\frac{dv}{d\pi} - 1\right\|_2}, \tag{15}$$

$$\tilde{A}_n := \frac{M}{2n}\left[\sqrt{2\tau_{\min}n\log n} + \sqrt{n} + 4\right]. \tag{16}$$

*Then, the following holds:*

$$\frac{1}{2}\mathbb{E}\big[\|P_n^0\|_{\mathcal{F}}\big] - \tilde{A}_n \leq \mathbb{E}\big[\|P_n - P\|_{\mathcal{F}}\big] \leq 2\mathbb{E}\big[\|P_n^0\|_{\mathcal{F}}\big] + A_n, \tag{17}$$

*where*

$$\|P_n^0\|_{\mathcal{F}} := \sup_{f \in \mathcal{F}} \left| \frac{1}{n} \sum_{i=1}^{n} \varepsilon_i f(X_i) \right|. \tag{18}$$

The proof of this lemma can be found in Appendix A of the supplement material. Compared with the i.i.d. case, the symmetrization inequality for Markov chain in Lemma 1 are different in two perspectives: (1) The expectation $\mathbb{E}\left[\|P_n^0\|_{\mathcal{F}}\right]$ is now is under the joint distributions of Markov chain and Rademacher random variables and (2) The term $A_n$ appears in both lower and upper bounds to compensate for the difference between the initial distribution $\nu$ and the stationary distribution $\pi$ of the Markov chain[2]. Later, we will see that $A_n$ represents the effects of data structures on the generalization errors in deep learning.

By applying Lemma 1, the following theorem can be proved. See a detailed proof in Appendix C in the supplement material.

**Theorem 2.** *Denote by*

$$B_n := \sqrt{\frac{2}{n(1-\lambda)} + \frac{64}{n^2(1-\lambda)^2} \left\| \frac{dv}{d\pi} - 1 \right\|_2}, \tag{19}$$

*Let $\varphi$ be a non-increasing function such that $\varphi(x) \geq \mathbf{1}_{(-\infty,0]}(x)$ for all $x \in \mathbb{R}$. For any $t > 0$,*

$$\mathbb{P}\left( \exists f \in \mathcal{F} : P\{f \leq 0\} > \inf_{\delta \in (0,1]} \left[ P_n \varphi\left(\frac{f}{\delta}\right) + \frac{8L(\varphi)}{\delta} R_n(\mathcal{F}) \right.\right.$$
$$\left.\left. + \left(t + \sqrt{\log \log_2 2\delta^{-1}}\right) \sqrt{\frac{\tau_{\min}}{n}} + B_n \right] \right) \leq \frac{\pi^2}{3} \exp(-2t^2) \tag{20}$$

*and*

$$\mathbb{P}\left( \exists f \in \mathcal{F} : P\{f \leq 0\} > \inf_{\delta \in (0,1]} \left[ P_n \varphi\left(\frac{f}{\delta}\right) + \frac{2L(\varphi)\sqrt{2\pi}}{\delta} G_n(\mathcal{F}) \right.\right.$$
$$\left.\left. + \left(t + \sqrt{\log \log_2 2\delta^{-1}}\right) \sqrt{\frac{\tau_{\min}}{n}} + \frac{2}{\sqrt{n}} + B_n \right] \right) \leq \frac{\pi^2}{3} \exp(-2t^2). \tag{21}$$

Since $B_n = O\left(1/\sqrt{n}\right)$, Theorem 2 shows that with high probability, the generalization error can be bounded by Rademacher or Gaussian complexity functions plus an $O\left(1/\sqrt{n}\right)$ term, where $n$ is the length of the training set. This fact also happens in i.i.d. case [4]. However, because the dependency among samples in Markov chain, the constant in $O(1/\sqrt{n})$ term is larger than the i.i.d. case.

It follows, in particular, in the example of the voting methods of combining classifiers [2], from Theorem 2, we achieve the following PAC-bound:

$$P\{\tilde{f} \leq 0\} \leq \inf_{\delta \in (0,1]} \left[ P_n\{\tilde{f} \leq \delta\} + \frac{8C}{\delta} \sqrt{\frac{V(\mathcal{H})}{n}} \right.$$
$$\left. + B_n + \left( \sqrt{\frac{1}{2} \log \frac{\pi^2}{3\alpha}} + \sqrt{\log \log_2 2\delta^{-1}} \right) \sqrt{\frac{\tau_{\min}}{n}} \right] \tag{22}$$

with probability at least $1 - \alpha$ (PAC-Bayes bound), where $V(\mathcal{H})$ is the VC-dimension of the class $\mathcal{H}$ and $C$ is some positive constant.

## 3.2 Bounding the generalization error in deep neural networks

In this section, we consider the same example as [4, Section 6]. However, we assume that feature vectors in the dataset are generated by a Markov chain instead of an i.i.d. process. Let $\mathcal{H}$ be a the class of all uniformly bounded functions $f : \mathcal{S} \to \mathbb{R}$. $\mathcal{H}$ is called the class of *base functions*.

---

[2]This difference causes a burn-in time [28] which is the time between the initial and first time that the Markov chain is stationary.

Consider a feed-forward neural network with $l$ layers of neurons $V = \{v_i\} \cup \bigcup_{j=0}^{l} V_j$ where $V_l = \{v_o\}$. The neurons $v_i$ and $v_o$ are called the input and the output neurons, respectively. To define the network, we will assign the labels to the neurons in the following way. Each of the base neurons is labelled by a function from the base class $\mathcal{H}$. Each neuron of the $j$-th layer $V_j$, where $j \geq 1$, is labelled by a vector $w := (w_1, w_2, \cdots, w_m) \in \mathbb{R}^m$, where $m$ is the number of inputs of the neuron. Here, $w$ will be called the vector of weights of the neuron.

Given a Borel function $\sigma$ from $\mathbb{R}$ into $[-1, 1]$ (for example, sigmoid function) and a vector $w := (w_1, w_2, \cdots, w_m) \in \mathbb{R}^m$ where $m = |\mathcal{H}| + 1$, let

$$N_{\sigma,w} : \mathbb{R}^m \to \mathbb{R}, N_{\sigma,w}(u_1, u_2, \cdots, u_m) := \sigma\left(\sum_{j=1}^{m} w_j u_j\right). \tag{23}$$

Let $\sigma_j : j \geq 1$ be functions from $\mathbb{R}$ into $[-1, 1]$, satisfying the Lipschitz conditions

$$\left|\sigma_j(u) - \sigma_j(v)\right| \leq L_j |u - v|, \qquad u, v \in \mathbb{R}. \tag{24}$$

The neural network works can be formed as the following. The input neuron inputs an instance $x \in \mathcal{S}$. A base neuron computes the value of the base function on this instance and outputs the value through its output edges. A neuron in the $j$-th layer ($j \geq 1$) computes and outputs through its output edges the value $N_{\sigma_j,w}(u_1, u_2, \cdots, u_m)$ (where $u_1, u_2, \cdots, u_m$ are the values of the inputs of the neuron). The network outputs the value $f(x)$ (of a function $f$ it computes) through the output edge.

We denote by $\mathcal{N}_l$ the set of such networks. We call $\mathcal{N}_l$ the class of feed-forward neural networks with base $\mathcal{H}$ and $l$ layers of neurons (and with sigmoid $\sigma_j$). Let $\mathcal{N}_\infty := \bigcup_{j=1}^{\infty} \mathcal{N}_j$. Define $\mathcal{H}_0 := \mathcal{H}$, and then recursively

$$\mathcal{H}_j := \left\{N_{\sigma_j,w}(h_1, h_2, \cdots, h_m) : m \geq 0, h_i \in \mathcal{H}_{j-1}, w \in \mathbb{R}^m\right\} \cup \mathcal{H}_{j-1}. \tag{25}$$

Denote $\mathcal{H}_\infty := \bigcup_{j=1}^{\infty} \mathcal{H}_j$. Clearly, $\mathcal{H}_\infty$ includes all the functions computable by feed-forward neural networks with base $\mathcal{H}$.

Let $\{b_j\}$ be a sequence of positive numbers. We also define recursively classes of functions computable by feed-forward neural networks with restrictions on the weights of neurons:

$$\mathcal{H}_j(b_1, b_2, \cdots, b_j) := \left\{N_{\sigma_j,w}(h_1, h_2, \cdots, h_m) : m \geq 0,\right.$$

$$\left. h_i \in \mathcal{H}_{j-1}(b_1, b_2, \cdots, b_{j-1}), w \in \mathbb{R}^m, \|w\|_1 \leq b_j\right\} \bigcup \mathcal{H}_{j-1}(b_1, b_2, \cdots, b_{j-1}), \tag{26}$$

where $\|w\|_1$ denotes the 1-norm of the vector $w$.

Clearly,

$$\mathcal{H}_\infty = \bigcup \left\{\mathcal{H}_j(b_1, \cdots, b_j) : b_1, \cdots, b_j < +\infty\right\}. \tag{27}$$

Denote by $\tilde{\mathcal{H}}$ the class of measurable functions $\tilde{f} : \mathcal{S} \times \mathcal{Y} \to \mathbb{R}$, where $\mathcal{Y}$ is the alphabet of labels. $\tilde{\mathcal{H}}$ is introduced for real machine learning applications where we need to work with a new Markov chain generated from both feature vectors and their labels $\{(X_n, Y_n)\}_{n=1}^{\infty}$ instead of the feature-based Markov chain $\{X_n\}_{n=1}^{\infty}$. See Subsection 2.2 for detailed discussions. For binary classification, $\tilde{\mathcal{H}} := \{\tilde{f} : f \in \mathcal{H}\}$, where $\tilde{f}(x, y) = yf(x)$. Let $\varphi$ be a function such that $\varphi(x) \geq I_{(-\infty,0]}(x)$ for all $x \in \mathbb{R}$ and $\varphi$ satisfies the Lipschitz condition with constant $L(\varphi)$. Then, the following is a direct application of Theorem 2.

**Theorem 3.** *For any $t \geq 0$ and for all $l \geq 1$,*

$$\mathbb{P}\left(\exists f \in \mathcal{H}(b_1, b_2, \cdots, b_l) : P\{\tilde{f} \leq 0\} > \inf_{\delta \in (0,1]} \left[P_n \varphi\left(\frac{\tilde{f}}{\delta}\right) + \frac{2\sqrt{2\pi}L(\varphi)}{\delta} \prod_{j=1}^{l}(2L_j b_j + 1)G_n(\mathcal{H})\right.\right.$$

$$\left.\left. + \left(t + \sqrt{\log \log_2 2\delta^{-1}}\right)\sqrt{\frac{\tau_{\min}}{n}} + B_n\right]\right) \leq \frac{\pi^2}{3}\exp(-2t^2), \tag{28}$$

*where $B_n$ is defined in* (19).

**Remark 4.** $P\{\tilde{f} \leq 0\}$ *represents the probability of mis-classification in the deep neural network.*

*Proof.* Let $\mathcal{H}'_l := \mathcal{H}(b_1, b_2, \cdots, b_l)$. As the proof of [4, Theorem 13], it holds that

$$G_n(\mathcal{H}'_l) \leq \prod_{j=1}^{l}(2L_j b_j + 1)G_n(\mathcal{H}). \tag{29}$$

Hence, (28) is a direct application of Theorem 2 and (29). $\square$

Now, given a neural network $f \in \mathcal{N}_\infty$, let

$$l(f) := \min\{j \geq 1 : f \in \mathcal{N}_j\}. \tag{30}$$

For any number $k$ such that $1 \leq k \leq l(f)$, let $V_k(f)$ be the set of all neurons of layer $k$ in the neural network which is represented by $f$. Denote by

$$W_k(f) := \max_{u \in V_k(f)} \|w^{(u)}\|_1 \vee b_k, \qquad k = 1, 2, \cdots, l(f) \tag{31}$$

where $w^{(u)}$ is the coefficient-vector associated with the neuron $u$ in this layer. Define

$$\Lambda(f) := \prod_{k=1}^{l(f)}(4L_k W_k(f) + 1), \qquad \Gamma_\alpha(f) := \sum_{k=1}^{l(f)}\sqrt{\frac{\alpha}{2}\log(2 + \log_2 W_k(f))}, \tag{32}$$

where $\alpha > 0$ is the number such that $\zeta(\alpha) < 3/2$, $\zeta$ being the Riemann zeta-function: $\zeta(\alpha) := \sum_{k=1}^{\infty} k^{-\alpha}$. Then, by using Theorem 3 with $b_k \to \infty$ and the same arguments as [4, Proof of Theorem 14], we obtain the following result. See Section 1.3 in the supplement material for more detailed derivations.

**Theorem 5.** *For any $t \geq 0$ and for all $l \geq 1$,*

$$\mathbb{P}\left(\exists f \in \mathcal{H}_\infty : P\{\tilde{f} \leq 0\} > \inf_{\delta \in (0,1]}\left[P_n\varphi\left(\frac{\tilde{f}}{\delta}\right) + \frac{2\sqrt{2\pi}L(\varphi)}{\delta}\Lambda(f)G_n(\mathcal{H}) + \frac{2}{\sqrt{n}}\right.\right.$$

$$\left.\left. + \left(t + \Gamma_\alpha(f) + \sqrt{\log\log_2 2\delta^{-1}}\right)\sqrt{\frac{\tau_{\min}}{n}} + B_n\right]\right) \leq \frac{\pi^2}{3}(3 - 2\zeta(\alpha))^{-1}\exp\left(-2t^2\right), \tag{33}$$

*where $B_n$ is defined in* (19).

### 3.3 Generalization Error Bounds on Bayesian Deep Learning

For Bayesian machine learning and deep learning, $\mathcal{F} := \{f : \mathcal{S} \times \mathcal{W} \to \mathbb{R}\}$, where $\mathcal{S}$ is the state space of the Markov chain and $\mathcal{W}$ is the domain of (random) coefficients. We assume that $\mathcal{S}$ and $\mathcal{W}$ are Polish spaces on $\mathbb{R}$, which include both discrete sets and $\mathbb{R}$. For example, in binary classification, $f(X, W) = \text{sgn}(W^T X + b)$ where the feature $X$ and the coefficient $W$ are random vectors with specific prior distributions. In practice, the distribution of $W$ is known which depends on our design method, and the distribution of $X$ is unknown. For example, $W$ is assumed to be Gaussian in Bayesian deep neural networks [22].

Since all the bounds on Subsections 3.1 and 3.2 hold for any function $f$ in $\mathcal{F}$ at each fixed vector $W = w$, hence, they can be directly applied to Bayesian settings where $W$ is random. However, these bounds are not expected to be tight enough since we don't use the prior distribution of $W$ when deriving them. In the following, we use another approach to derive new (and tighter) bounds for Bayesian deep learning and machine learning from all the bounds in Subsections 3.1 and 3.2. For illustration purposes, we only derive a new bound. Other bounds can be derived in a similar fashion. We assume that $W_1, W_2, \cdots, W_n$ are i.i.d. random variables as in [22].

Let

$$\tilde{P}_n := \frac{1}{n}\sum_{i=1}^{n}\delta_{X_i, W_i}, \tag{34}$$

and define a new probability measure $\tilde{P}$ on $\mathcal{S} \times \mathcal{W}$ such that

$$\tilde{P}(A) := \int_A \tilde{\pi}(x, w) dx dw, \tag{35}$$

for all (Borel) set $A$ on $\mathcal{S} \times \mathcal{W}$. Here, $\tilde{\pi}$ is the stationary distribution of the irreducible Markov process $\{(X_n, W_n)\}_{n=1}^\infty$ with stochastic matrix $\tilde{Q} := \{Q(x, w) P_W(w)\}_{x \in \mathcal{S}, w \in \mathcal{W}}$. In addition, define two new (averaging) linear functionals:

$$\hat{P}_n(f) = \frac{1}{n} \sum_{i=1}^n \int_\mathcal{W} f(X_i, w) dP_W(w), \tag{36}$$

and

$$\hat{P}(f) := \int_\mathcal{S} \int_\mathcal{W} f(x, w) \tilde{\pi}(x, w) dP_W(w) dP(x). \tag{37}$$

In practice, the prior distribution of $W$ is known, so we can estimate $\hat{P}_n(f)$ based on the training set $\{(X_1, Y_1), (X_2, Y_2), \cdots, (X_n, Y_n)\}$, which is a Markov chain on $\mathcal{X} \times \mathcal{Y}$ (cf. Section 2.2). The following result can be proved.

**Theorem 6.** *Let $\varphi$ be a sequence of function such that $\varphi(x) \geq I_{(-\infty, 0]}(x)$. For any $t > 0$,*

$$\mathbb{P}\left(\exists f \in \mathcal{F} : \hat{P}\{f \leq 0\} > \inf_{\delta \in (0,1]} \left[ \hat{P}_n \varphi\left(\frac{f}{\delta}\right) + \frac{8L(\varphi)}{\delta} R_n(\mathcal{F}) \right.\right.$$
$$\left.\left. + \left(t + \sqrt{\log \log_2 2\delta^{-1}}\right)\sqrt{\frac{\tau_{\min}}{n}} + B_n \right]\right) \leq \frac{\pi^2}{3} \exp(-2t^2) \tag{38}$$

*and*

$$\mathbb{P}\left(\exists f \in \mathcal{F} : \hat{P}\{f \leq 0\} > \inf_{\delta \in (0,1]} \left[ \hat{P}_n \varphi\left(\frac{f}{\delta}\right) + \frac{2L(\varphi)\sqrt{2\pi}}{\delta} G_n(\mathcal{F}) + \frac{2}{\sqrt{n}} \right.\right.$$
$$\left.\left. + \left(t + \sqrt{\log \log_2 2\delta^{-1}}\right)\sqrt{\frac{\tau_{\min}}{n}} + B_n \right]\right) \leq \frac{\pi^2}{3} \exp\left(-2t^2\right). \tag{39}$$

*Proof.* Let $W_1, W_2, \cdots, W_n$ be an $n$ samples of $W \sim P_W$ on $\mathcal{W}$ (or samples of some set of random coefficients). For simplicity, we assume that $\{W_n\}_{n=1}^\infty$ is an i.i.d. sequence. Then, it is obvious that $\{(X_n, W_n)\}_{n=1}^\infty$ forms a Markov chain with probability transition probability

$$\tilde{Q}(x_n, w_n; x_{n+1}, w_{n+1}) = \mathbb{P}(X_{n+1} = x_{n+1}, W_{n+1} = w_{n+1} | X_n = x_n, W_n = w_n) \tag{40}$$
$$= Q(x_n, x_{n+1}) P_W(w_{n+1}). \tag{41}$$

From Theorem 2, it holds that

$$\mathbb{P}\left(\exists f \in \mathcal{F} : \tilde{P}\{f \leq 0\} > \inf_{\delta \in (0,1]} \left[ \tilde{P}_n \varphi\left(\frac{f}{\delta}\right) + \frac{8L(\varphi)}{\delta} R_n(\mathcal{F}) \right.\right.$$
$$\left.\left. + \left(t + \sqrt{\log \log_2 2\delta^{-1}}\right)\sqrt{\frac{\tau_{\min}}{n}} + B_n \right]\right) \leq \frac{\pi^2}{3} \exp(-2t^2). \tag{42}$$

This means that with probability at least $1 - \frac{\pi^2}{3} \exp(-2t^2)$, it holds that

$$\tilde{P}\{f \leq 0\} \leq \inf_{\delta \in (0,1]} \left[ \tilde{P}_n \varphi\left(\frac{f}{\delta}\right) + \frac{8L(\varphi)}{\delta} R_n(\mathcal{F}) + \left(t + \sqrt{\log \log_2 2\delta^{-1}}\right)\sqrt{\frac{\tau_{\min}}{n}} + B_n \right]. \tag{43}$$

From (43), it holds that with probability at least $1 - \frac{\pi^2}{3} \exp(-2t^2)$,

$$\tilde{P}\{f \leq 0\} \leq \inf_{\delta \in (0,1]} \left[ \frac{1}{n} \sum_{i=1}^n \varphi\left(\frac{f(X_i, W_i)}{\delta}\right) + \frac{8L(\varphi)}{\delta} R_n(\mathcal{F}) \right.$$
$$\left. + \left(t + \sqrt{\log \log_2 2\delta^{-1}}\right)\sqrt{\frac{\tau_{\min}}{n}} + B_n \right]. \tag{44}$$

From (44), with probability at least $1 - \frac{\pi^2}{3}\exp(-2t^2)$, it holds that

$$
\mathbb{E}_W\big[\tilde{P}\{f \leq 0\}\big] \leq \mathbb{E}_W\bigg[\inf_{\delta \in (0,1]}\bigg[\frac{1}{n}\sum_{i=1}^{n}\varphi\bigg(\frac{f(X_i, W_i)}{\delta}\bigg) + \frac{8L(\varphi)}{\delta}R_n(\mathcal{F})
$$

$$
+ \big(t + \sqrt{\log\log_2 2\delta^{-1}}\big)\sqrt{\frac{\tau_{\min}}{n}} + B_n\bigg] \tag{45}
$$

$$
\leq \inf_{\delta \in (0,1]}\bigg[\frac{1}{n}\sum_{i=1}^{n}\mathbb{E}_W\bigg[\varphi\bigg(\frac{f(X_i, W)}{\delta}\bigg)\bigg] + \frac{8L(\varphi)}{\delta}R_n(\mathcal{F})
$$

$$
+ \big(t + \sqrt{\log\log_2 2\delta^{-1}}\big)\sqrt{\frac{\tau_{\min}}{n}} + B_n\bigg]. \tag{46}
$$

From (46), we obtain (38). Similarly, we can achieve (39). $\qquad\square$

## 4 Extension to High-Order Markov Chains

In this subsection, we extend our results in previous sections to $m$-order homogeneous Markov chain. The main idea is to convert $m$-order homogeneous Markov chains to 1-order homogeneous Markov chain and use our results in previous sections to bound the generalization error. We start with the following simple example.

**Example 7.** *[m-order moving average process without noise] Consider the following $m$-order Markov chain*

$$
X_k = \sum_{i=1}^{m}a_i X_{k-i}, \qquad k \in \mathbb{Z}_+. \tag{47}
$$

*Let $Y_k := [X_{k+m-1}, X_{k+m-2}, \cdots, X_k]^T$. Then, from (47), we obtain $Y_{k+1} = \mathbf{G}Y_k, \ \forall k \in \mathbb{Z}_+$ where*

$$
\mathbf{G} := \begin{pmatrix} a_1 & a_2 & \cdots & a_{m-1} & a_m \\ 1 & 0 & \cdots & 0 & 0 \\ 0 & 1 & \cdots & 0 & 0 \\ \vdots & \vdots & \ddots & \vdots & \vdots \\ 0 & 0 & \cdots & 1 & 0 \end{pmatrix}. \tag{48}
$$

*It is clear that $\{\mathbf{Y}_n\}_{n=1}^{\infty}$ is an order-1 Markov chain. Hence, instead of directly working with the $m$-order Markov chain $\{\mathbf{X}_n\}_{n=1}^{\infty}$, we can find an upper bound for the Markov chain $\{Y_n\}_{n=1}^{\infty}$.*

*To derive generalization error bounds for the Markov chain $\{Y_n\}_{n=1}^{\infty}$, we can use the following arguments. For all $f \in \mathcal{F}$ and $(x_k, x_{k+1}, \cdots, x_{k+m-1})$, by setting $\tilde{f}(x_k, x_{k+1}, \cdots, x_{k+m-1}) = f(x_k)$ where $\tilde{f} : \mathcal{S}^m \to \mathbb{R}$, we obtain*

$$
\frac{1}{n}\sum_{i=1}^{n}\mathbf{1}\{f(X_i) \leq 0\} = \frac{1}{n}\sum_{i=1}^{n}\mathbf{1}\{\tilde{f}(Y_i) \leq 0\}. \tag{49}
$$

*Hence, by applying all the results for 1-order Markov chain $\{Y_n\}_{n=1}^{\infty}$, we obtain corresponding upper bounds for the sequence of $m$-order Markov chain $\{X_n\}_{n=1}^{\infty}$.*

This approach can be extended to more general Markov chain. See Section 4 in the Supplementary Material for details.

## 5 Conclusions

In this paper, we derive upper bounds on generalization errors for machine learning and deep neural networks based on a new assumption that the dataset has Markov or hidden Markov structure. We also propose a new method to convert all these bounds to Bayesian deep learning and machine learning. Extension to $m$-order Markov chains and a mixture of Markov chains are also given. An interesting future research topic is to develop some new algorithms to evaluate performance of these bounds on real Markov datasets.

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
