# Suplement Material:
# Generalization Error Bounds on Deep Learning with Markov Datasets

**Lan V. Truong**[*]
Department of Engineering
University of Cambridge
Cambridge, CB2 1PZ
lt407@cam.ac.uk

## 1 Main and Supplemental Results

### 1.1 Probabilistic Bounds for General Function Classes

In this section, we develop probabilistic bounds for general function classes in terms of Gaussian and Rademacher complexities. First, we prove the following key lemma, which is an extension of the symmetrization inequality for i.i.d. sequences (for example, [1]) to a new version for Markov sequences $\{X_n\}_{n=1}^\infty$ with the stationary distribution $\pi$ and the initial distribution $\nu \in \mathcal{M}_2$:

**Lemma 1.** *Let $\mathcal{F}$ be a class of functions which are uniformly bounded by $M$. For all $n \in \mathbb{Z}^+$, define*

$$A_n := \sqrt{\frac{2M}{n(1-\lambda)} + \frac{64M^2}{n^2(1-\lambda)^2}\left\|\frac{dv}{d\pi} - 1\right\|_2}, \tag{1}$$

$$\tilde{A}_n := \frac{M}{2n}\left[\sqrt{2\tau_{\min}n\log n} + \sqrt{n} + 4\right]. \tag{2}$$

*Then, the following holds:*

$$\frac{1}{2}\mathbb{E}\big[\|P_n^0\|_{\mathcal{F}}\big] - \tilde{A}_n \leq \mathbb{E}\big[\|P_n - P\|_{\mathcal{F}}\big] \leq 2\mathbb{E}\big[\|P_n^0\|_{\mathcal{F}}\big] + A_n, \tag{3}$$

*where*

$$\|P_n^0\|_{\mathcal{F}} := \sup_{f \in \mathcal{F}}\left|\frac{1}{n}\sum_{i=1}^n \varepsilon_i f(X_i)\right|. \tag{4}$$

*Proof.* See Appendix A. $\square$

Next, we prove the following theorem.

**Theorem 2.** *Consider a countable family of Lipschitz function $\Phi = \{\varphi_k : k \geq 1\}$, where $\varphi_k : \mathbb{R} \to \mathbb{R}$ satisfies $\mathbf{1}_{(-\infty,0]}(x) \leq \varphi_k(x)$ for all $k$. For each $\varphi \in \Phi$, denote by $L(\varphi)$ its Lipschitz constant. Define*

$$B_n := \sqrt{\frac{2}{n(1-\lambda)} + \frac{64}{n^2(1-\lambda)^2}\left\|\frac{dv}{d\pi} - 1\right\|_2}. \tag{5}$$

[*]Use footnote for providing further information about author (webpage, alternative address)—*not* for acknowledging funding agencies.

36th Conference on Neural Information Processing Systems (NeurIPS 2022).

*Then, for any $t > 0$,*

$$\mathbb{P}\left(\exists f \in \mathcal{F} : P\{f \leq 0\} > \inf_{k>0}\left[P_n\varphi_k(f) + 4L(\varphi_k)R_n(\mathcal{F}) + \left(t + \sqrt{\log k}\right)\sqrt{\frac{\tau_{\min}}{n}} + B_n\right]\right)$$
$$\leq \frac{\pi^2}{3}\exp(-2t^2) \tag{6}$$

*and*

$$\mathbb{P}\left(\exists f \in \mathcal{F} : P\{f \leq 0\} > \inf_{k>0}\left[P_n\varphi_k(f) + \sqrt{2\pi}L(\varphi_k)G_n(\mathcal{F})\right.\right.$$
$$\left.\left. + \frac{2}{\sqrt{n}} + \left(t + \sqrt{\log k}\right)\sqrt{\frac{\tau_{\min}}{n}}\right]\right) \leq \frac{\pi^2}{3}\exp(-2t^2). \tag{7}$$

*Proof.* See Appendix B. □

**Theorem 3.** *Let $\varphi$ is a non-increasing function such that $\varphi(x) \geq \mathbf{1}_{(-\infty,0]}(x)$ for all $x \in \mathbb{R}$. For any $t > 0$,*

$$\mathbb{P}\left(\exists f \in \mathcal{F} : P\{f \leq 0\} > \inf_{\delta \in (0,1]}\left[P_n\varphi\left(\frac{f}{\delta}\right) + \frac{8L(\varphi)}{\delta}R_n(\mathcal{F}) + \left(t + \sqrt{\log\log_2 2\delta^{-1}}\right)\sqrt{\frac{\tau_{\min}}{n}} + B_n\right]\right)$$
$$\leq \frac{\pi^2}{3}\exp(-2t^2) \tag{8}$$

*and*

$$\mathbb{P}\left(\exists f \in \mathcal{F} : P\{f \leq 0\} > \inf_{\delta \in (0,1]}\left[P_n\varphi\left(\frac{f}{\delta}\right) + \frac{2L(\varphi)\sqrt{2\pi}}{\delta}G_n(\mathcal{F})\right.\right.$$
$$\left.\left. + \frac{2}{\sqrt{n}} + \left(t + \sqrt{\log\log_2 2\delta^{-1}}\right)\sqrt{\frac{\tau_{\min}}{n}} + B_n\right]\right) \leq \frac{\pi^2}{3}\exp(-2t^2), \tag{9}$$

*where $B_n$ is defined in* (5).

*Proof.* See Appendix C. □

In the next statements, we use Rademacher complexities, but Gaussian complexities can be used similarly. Now, assume that $\varphi$ is a function from $\mathbb{R}$ to $\mathbb{R}$ such that $\varphi(x) \leq I_{(-\infty,0]}(x)$ for all $x \in \mathbb{R}$ and $\varphi$ satisfies the Lipschitz with constant $L(\varphi)$. Then, the following theorems can be proved by using similar arguments as the proof of Theorem 3.

**Theorem 4.** *Let $\varphi$ is a nonincreasing function such that $\varphi(x) \leq \mathbf{1}_{(-\infty,0]}(x)$ for all $x \in \mathbb{R}$. For any $t > 0$,*

$$\mathbb{P}\left(\exists f \in \mathcal{F} : P\{f \leq 0\} < \sup_{\delta \in (0,1]}\left[P_n\varphi\left(\frac{f}{\delta}\right) + \frac{8L(\varphi)}{\delta}R_n(\mathcal{F})\right.\right.$$
$$\left.\left. + \left(t + \sqrt{\log\log_2 2\delta^{-1}}\right)\sqrt{\frac{\tau_{\min}}{n}} + B_n\right]\right) \leq \frac{\pi^2}{3}\exp(-2t^2) \tag{10}$$

*and*

$$\mathbb{P}\left(\exists f \in \mathcal{F} : P\{f \leq 0\} < \sup_{\delta \in (0,1]}\left[P_n\varphi\left(\frac{f}{\delta}\right) + \frac{2\sqrt{2\pi}L(\varphi)}{\delta}G_n(\mathcal{F}) + \frac{2}{\sqrt{n}}\right]\right.$$
$$\left. + \left(t + \sqrt{\log\log_2 2\delta^{-1}}\right)\sqrt{\frac{\tau_{\min}}{n}} + B_n\right]\right) \leq \frac{\pi^2}{3}\exp(-2t^2), \tag{11}$$

*where $B_n$ is defined in* (5).

By combining Theorem 3 and Theorem 4, we obtain the following result.

**Theorem 5.** *Let*

$$\Delta_n(\mathcal{F}; \delta) := \frac{8}{\delta} R_n(\mathcal{F}) + \sqrt{\frac{\tau_{\min} \log \log_2 2\delta^{-1}}{n}} + B_n. \tag{12}$$

*Then, for all $t > 0$,*

$$\mathbb{P}\Bigg( \exists f \in \mathcal{F} : \big| P_n\{f \leq 0\} - P\{f \leq 0\} \big| > \inf_{\delta \in (0,1]} \bigg( P_n\{|f| \leq \delta\} + \Delta_n(\mathcal{F}; \delta) + t\sqrt{\frac{\tau_{\min}}{n}} \bigg) \Bigg)$$

$$\leq \frac{2\pi^2}{3} \exp(-2t^2) \tag{13}$$

*and*

$$\mathbb{P}\Bigg( \exists f \in \mathcal{F} : \big| P_n\{f \leq 0\} - P\{f \leq 0\} \big| > \inf_{\delta \in (0,1]} \bigg( P\{|f| \leq \delta\} + \Delta_n(\mathcal{F}; \delta) + t\sqrt{\frac{\tau_{\min}}{n}} \bigg) \Bigg)$$

$$\leq \frac{2\pi^2}{3} \exp(-2t^2). \tag{14}$$

*Proof.* Equation (13) is drawn by setting $\varphi(x) = \mathbf{1}\{x \leq 0\} + (1-x)\mathbf{1}\{0 \leq x \leq 1\}$ in Theorem 3 and Theorem 4. Equation (14) is drawn by setting $\varphi(x) = \mathbf{1}\{x \leq -1\} - x\mathbf{1}\{-1 \leq x \leq 0\}$ in these theorems. $\square$

### 1.2 Conditions on Random Entropies and $\gamma$-Margins

As [2], given a metric space $(T, d)$, we denote by $H_d(T; \varepsilon)$ the $\varepsilon$-entropy of $T$ with respect to $d$, that is

$$H_d(T; \varepsilon) := \log N_d(T; \varepsilon), \tag{15}$$

where $N_d(T; \varepsilon)$ is the minimal number of balls of radius $\varepsilon$ covering $T$. Let $d_{P_n, 2}$ denote the metric of the space $L_2(\mathcal{S}; dP_n)$:

$$d_{P_n, 2}(f, g) := \big( P_n |f - g|^2 \big)^{1/2}. \tag{16}$$

For each $\gamma \in (0, 1]$, define

$$\delta_n(\gamma; f) := \sup \bigg\{ \delta \in (0, 1) : \delta^{\frac{\gamma}{2}} P(f \leq \delta) \leq n^{-\frac{1}{2} + \frac{\gamma}{4}} \bigg\} \tag{17}$$

and

$$\hat{\delta}_n(\gamma; f) := \sup \bigg\{ \delta \in (0, 1) : \delta^{\frac{\gamma}{2}} P_n(f \leq \delta) \leq n^{-\frac{1}{2} + \frac{\gamma}{4}} \bigg\}. \tag{18}$$

We call $\delta_n(\gamma; f)$ and $\hat{\delta}_n(\gamma; f)$, respectively, the *$\gamma$-margin* and *empirical $\gamma$-margin* of $f$.

**Theorem 6.** *Suppose that for some $\alpha \in (0, 2)$ and some constant $D > 0$,*

$$H_{d_{P_n}, 2}(\mathcal{F}; u) \leq D u^{-\alpha}, \qquad u > 0 \qquad a.s., \tag{19}$$

*Then, for any $\gamma \geq \frac{2\alpha}{2+\alpha}$, there exists some constants $\zeta, \upsilon > 0$ such that when $n$ is large enough,*

$$\mathbb{P}\bigg[ \forall f \in \mathcal{F} : \zeta^{-1} \hat{\delta}_n(\gamma; f) \leq \delta_n(\gamma; f) \leq \zeta \hat{\delta}_n(\gamma; f) \bigg] \geq 1 - 2\upsilon \big( \log_2 \log_2 n \big) \exp \big\{ - n^{\frac{\gamma}{2}}/2 \big\}. \tag{20}$$

*Proof.* See Appendix D for a detailed proof. $\square$

## 1.3 Convergence rates of empirical margin distributions

First, we prove the following lemmas.

**Lemma 7.** *For any class $\mathcal{F}$ of bounded measurable functions from $\mathcal{S} \to \mathbb{R}$, with probability at least $1 - 2\exp\left(-2t^2\right)$, the following holds:*

$$\sup_{f \in \mathcal{F}} \sup_{y \in \mathbb{R}} \left| P_n(f \leq y) - P(f \leq y) \right| \leq \sqrt{B_n} + t\sqrt{\frac{\tau_{\min}}{n}}, \tag{21}$$

*where $B_n$ is defined in (5).*

**Remark 8.** *By setting $t = \sqrt{2\log n}$, (21) shows that $\sup_{f \in \mathcal{F}} \sup_{y \in \mathbb{R}} \left| P_n(f \leq y) - P(f \leq y) \right| \to 0$ as $n \to \infty$.*

*Proof.* See Appendix E. $\qquad\qquad\qquad\qquad\qquad\qquad\qquad\qquad\qquad\qquad\qquad\qquad\qquad$ $\square$

Now, for each $f \in \mathcal{F}$, define

$$F_f(y) := P\{f \leq y\}, \qquad F_{n,f} := P_n\{f \leq y\}, \qquad y \in \mathbb{R}. \tag{22}$$

Let $L$ denote the Lévy distance between the distributions in $\mathbb{R}$:

$$L(F, G) := \inf\{\delta > 0 : F(t) \leq G(t+\delta) + \delta, \qquad G(t) \leq F(t+\delta) + \delta, \qquad \forall t \in \mathbb{R}\}. \tag{23}$$

**Lemma 9.** *Let $M > 0$ and $\mathcal{F}$ be a class of measurable functions from $\mathcal{S}$ into $[-M, M]$. Let $\varphi$ be equal to $1$ for $x \leq 0$, $0$ for $x \geq 1$ and linear between them. Define*

$$\tilde{\mathcal{G}}_\varphi := \left\{ \varphi \circ \left( \frac{f-y}{\delta} \right) - 1 : f \in \mathcal{F}, \quad y \in [-M, M] \right\} \tag{24}$$

*for some $\delta > 0$. Recall the definition of $B_n$ in (5). Then, for all $t > 0$ and $\delta > 0$, the following holds:*

$$\mathbb{P}\left\{ \sup_{f \in \mathcal{F}} L(F_f, F_{f,n}) \geq \delta + \mathbb{E}\left[\|P_n^0\|_{\tilde{\mathcal{G}}_\varphi}\right] + B_n + t\sqrt{\frac{\tau_{\min}}{n}} \right\} \leq 2\exp(-2t^2). \tag{25}$$

*Especially, for all $t > 0$, we have*

$$\mathbb{P}\left\{ \sup_{f \in \mathcal{F}} L(F_f, F_{f,n}) \geq 4\sqrt{\mathbb{E}[\|P_n^0\|_{\mathcal{F}}] + M/\sqrt{n}} + B_n + t\sqrt{\frac{\tau_{\min}}{n}} \right\} \leq 2\exp(-2t^2). \tag{26}$$

*Proof.* See Appendix F. $\qquad\qquad\qquad\qquad\qquad\qquad\qquad\qquad\qquad\qquad\qquad\qquad\qquad$ $\square$

In what follows, for a function $f$ from $\mathcal{S}$ into $\mathbb{R}$ and $M > 0$, we denote by $f_M$ the function that is equal to $f$ if $|f| \leq M$, is equal to $M$ if $f > M$ and is equal to $-M$ if $f < -M$. We set

$$\mathcal{F}_M := \left\{ f_M : f \in \mathcal{F} \right\}. \tag{27}$$

As always, a function $\mathcal{F}$ from $\mathcal{S}$ into $[0, \infty)$ is called an envelope of $\mathcal{F}$ iff $|f(x)| \leq F(x)$ for all $f \in \mathcal{F}$ and all $x \in \mathcal{S}$.

We write $\mathcal{F} \in \mathrm{GC}(P)$ iff $\mathcal{F}$ is a Glivenko-Cantelli class with respect to $P$ (i.e., $\|P_n - P\|_{\mathcal{F}} \to 0$ as $n \to \infty$ a.s.). We write $\mathcal{F} \in \mathrm{BCLT}(P)$ and say that $\mathcal{F}$ satisfies the Bounded Central Limit Theorem for $P$ iff

$$\mathbb{E}\left[\|P_n - P\|_{\mathcal{F}}\right] = O(n^{-1/2}). \tag{28}$$

Based on Lemma and Lemma 9, we prove the following theorems.

**Theorem 10.** *Suppose that*

$$\sup_{f \in \mathcal{F}} P\{|f| \geq M\} \to 0 \qquad as \qquad M \to \infty. \tag{29}$$

*Then, the following two statements are equivalent:*

- *(i) $\mathcal{F}_M \in \mathrm{GC}(\mathrm{P})$ for all $M > 0$*
  *and*

- *(ii) $\sup_{f \in \mathcal{F}} L(F_{n,f}, F_f) \to 0 \quad a.s. \quad n \to \infty$.*

*Proof.* See Appendix G. □

Next, the following theorems hold.

**Theorem 11.** *[2, Theorem 7] The following two statements are equivalent:*

- *(i) $\mathcal{F} \in \mathrm{GC}(\mathrm{P})$ for all $M > 0$*
  *and*

- *(ii) there exists a P-integrable envelope for the class $\mathcal{F}^{(c)} = \{f - Pf : f \in \mathcal{F}\}$ and $\sup_{f \in \mathcal{F}} L(F_{n,f}, F_f) \to 0 \quad n \to \infty$ and*

$$\sup_{f \in \mathcal{F}} L(F_{n,f}, F_f) \to 0 \quad a.s. \quad n \to \infty. \tag{30}$$

Now, we prove the following theorem.

**Theorem 12.** *Suppose that the class $\mathcal{F}$ is uniformly bounded. If $\mathcal{F} \in \mathrm{BCLT}(\mathrm{P})$, then*

$$\sup_{f \in \mathcal{F}} L(F_{n,f}, F_f) = O_P\left(\left(\frac{\log n}{n}\right)^{1/4}\right) \quad n \to \infty. \tag{31}$$

*Moreover, for some $\alpha \in (0, 2)$ and for some $D > 0$*

$$H_{d_{P_n},2}(\mathcal{F}; u) \le Du^{-\alpha} \log n, \quad u > 0, \quad a.s., \tag{32}$$

*then*

$$\sup_{f \in \mathcal{F}} L(F_{n,f}, F_f) = O\big(n^{-\frac{1}{2+\alpha}} \log n\big) \quad n \to \infty, \quad a.s., \tag{33}$$

*Proof.* Appendix H. □

## 1.4 Bounding the generalization error of convex combinations of classifiers

We start with an application of the inequalities in Subsection 1.1 to bounding the generalization error in general classification problems. Assume that the labels take values in a finite set $\mathcal{Y}$ with $|\mathcal{Y}| = K$. Consider a class $\tilde{\mathcal{F}}$ of functions from $\mathcal{S} \times \mathcal{Y}$ into $\mathbb{R}$. A function $f \in \tilde{\mathcal{F}}$ predicts a label $y \in \mathcal{Y}$ for an example $x \in \mathcal{S}$ iff

$$f(x, y) > \max_{y' \neq y} f(x, y'). \tag{34}$$

In practice, $f(x, y)$ can be set equal to $P(y|x)$, so $\mathcal{F}$ can be assumed to be uniformly bounded. The margin of a labelled example $(x, y)$ is defined as

$$m_{f,y}(x) := f(x, y) - \max_{y' \neq y} f(x, y'), \tag{35}$$

so $f$ mis-classifies the label example $(x, y)$ if and only if $m_{f,y} \le 0$. Let

$$\mathcal{F} := \big\{f(\cdot, y) : y \in \mathcal{Y}, f \in \tilde{\mathcal{F}}\big\}. \tag{36}$$

Then, we can show the following theorem.

**Theorem 13.** *For all $t > 0$, it holds that*

$$\mathbb{P}\bigg(\exists f \in \mathcal{F} : P\{m_{f,y} \le 0\} > \inf_{\delta \in (0,1]} \bigg[P_n\{m_{f,y} \le \delta\} + \frac{8}{\delta}(2K - 1)R_n(\mathcal{F})$$

$$+ \big(t + \sqrt{\log \log_2 2\delta^{-1}}\big)\sqrt{\frac{\tau_{\min}}{n}} + B_n\bigg]\bigg) \le \frac{\pi^2}{3} \exp(-2t^2), \tag{37}$$

*where $B_n$ is defined in (5). Here,*

$$P\{m_{f,y} \leq 0\} := \sum_{x \in \mathcal{S}} \pi(x)\mathbf{1}\{m_{f,y}(x) \leq 0\}, \tag{38}$$

*and*

$$P_n\{m_{f,y} \leq \delta\} = \frac{1}{n}\sum_{k=1}^{n}\mathbf{1}\{m_{f,y}(X_k) \leq \delta\} \tag{39}$$

*is the empirical distribution of the Markov process $\{m_{f,y}(X_n)\}_{n=1}^{\infty}$ given $f$.*

*Proof.* First, we need to bound the Rademacher's complexity for the class of functions $\{m_{f,y} : f \in \tilde{\mathcal{F}}\}$. Observe that

$$\mathbb{E}\left[\sup_{f \in \tilde{\mathcal{F}}}\left|n^{-1}\sum_{j=1}^{n}\varepsilon_j m_{f,y}(X_j)\right|\right]. \tag{40}$$

By [2, Proof of Theorem 11], we have

$$\mathbb{E}\left[\sup_{f \in \tilde{\mathcal{F}}}\left|n^{-1}\sum_{j=1}^{n}\varepsilon_j m_{f,y}(X_j)\right|\right] \leq (2K-1)R_n(\mathcal{F}), \tag{41}$$

where $R_n(\mathcal{F})$ is *the Rademacher complexity function* of the class $\mathcal{F}$. Now, assume that this class of function is uniformly bounded as in practice. Hence, by Theorem 3 for $\varphi$ that is equal to 1 on $(-\infty, 0]$, is equal to 0 on $[1, +\infty)$ and is linear in between, we obtain (37). $\square$

In addition, by using the fact that $(X_1, Y_1) - (X_2, Y_2)\cdots - (X_n, Y_n)$ forms a Markov chain with stationary distribution $\tilde{\pi}$ (see the discussion on Section 2.2 in the main document), by applying Theorem 3, we obtain the following result:

**Theorem 14.** *Let $\varphi$ is a nonincreasing function such that $\varphi(x) \geq \mathbf{1}_{(-\infty,0]}(x)$ for all $x \in \mathbb{R}$. For any $t > 0$,*

$$\mathbb{P}\left(\exists f \in \tilde{\mathcal{F}} : P\{\tilde{f} \leq 0\} > \inf_{\delta \in (0,1]}\left[P_n\varphi\left(\frac{\tilde{f}}{\delta}\right) + \frac{8L(\varphi)}{\delta}R_n(\mathcal{H})\right.\right.$$
$$\left.\left. + \left(t + \sqrt{\log\log_2 2\delta^{-1}}\right)\sqrt{\frac{\tau_{\min}}{n}} + B_n\right]\right) \leq \frac{\pi^2}{3}\exp(-2t^2), \tag{42}$$

*where $B_n$ is defined in (5).*

As in [2], in the voting methods of combining classifiers, a classifier produced at each iteration is a convex combination $\tilde{f}$ of simple base classifiers from the class $\mathcal{H}$. In addition, the Rademacher complexity can be bounded above by

$$R_n(\mathcal{H}) \leq C\sqrt{\frac{V(\mathcal{H})}{n}} \tag{43}$$

for some constant $C > 0$, where $V(\mathcal{H})$ is the VC-dimesion of $\mathcal{H}$. Let $\varphi$ be equal to 1 on $(-\infty, 0]$, is equal to 0 on $[1, +\infty)$ and is linear in between. By setting $t_\alpha = \sqrt{\frac{1}{2}\log\frac{\pi^2}{3\alpha}}$, from Theorem 14, with probability at least $1 - \alpha$, it holds that

$$P\{\tilde{f} \leq 0\} \leq \inf_{\delta \in (0,1]}\left[P_n\{\tilde{f} \leq \delta\} + \frac{8C}{\delta}\sqrt{\frac{V(\mathcal{H})}{n}} + \left(t_\alpha + \sqrt{\log\log_2 2\delta^{-1}}\right)\sqrt{\frac{\tau_{\min}}{n}} + B_n\right], \tag{44}$$

which extends the result of Bartlett et al. [4] to Markov dataset (PAC-bound).

## 1.5 Bounding the generalization error in neural network learning

In this section, we consider the same example as [2, Section 6]. However, we assume that feature vectors in this dataset is generated by a Markov chain instead of an i.i.d. process. Let $\mathcal{H}$ be a class of measurable functions from $\mathcal{S} \to \mathbb{R}$ (base functions). Let $\tilde{\mathcal{H}}$ be the set of function $\tilde{f} : \mathcal{S} \times \mathcal{Y} \to \mathbb{R}$. The introduction of $\tilde{\mathcal{H}}$ is to deal with the new Markov chain $\{(X_n, Y_n)\}_{n=1}^{\infty}$ which is generated by both feature vectors and their labels instead of the feature-based Markov chain $\{X_n\}_{n=1}^{\infty}$.

Consider a feed-forward neural network where the set $V$ of all the neurons is divided into layers

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

$$\leq \frac{\pi^2}{3} \big( 3 - 2\zeta(\alpha) \big)^{-1} \exp\big( - 2t^2 \big), \qquad (61)$$

*where $B_n$ is defined in* (5).

*Proof.* Let

$$\Delta_k := \begin{cases} [2^{k-1}, 2^k) & \text{for} \quad k \in \mathbb{Z}, k \neq 0, 1 \\ [1/2, 2) & \text{for} k = 1. \end{cases} \tag{62}$$

Then, using the following partition:

$$\{f \in \mathcal{H}_\infty\} = \bigcup_{l=0}^{\infty} \bigcup_{k_1 \in \mathbb{Z}\backslash\{0\}} \bigcup_{k_2 \in \mathbb{Z}\backslash\{0\}} \cdots \bigcup_{k_l \in \mathbb{Z}\backslash\{0\}} \left\{ f \in \mathcal{H}_\infty : l(f) = l, W_j(f) \in \Delta_{k_j}, \; \forall j \in [l] \right\}. \tag{63}$$

On each subset $\left\{ f \in \mathcal{H}_\infty : l(f) = l, W_j(f) \in \Delta_{k_j}, \; \forall j \in [l] \right\}$, we can lower bound $\Lambda(f)$ and $\Gamma_\alpha(f)$ by

$$\Lambda(f) \geq \prod_{j=1}^{l} (2L_j 2^{k_j} + 1), \tag{64}$$

$$\Gamma_\alpha(f) \geq \sum_{j=1}^{l} \sqrt{\frac{\alpha}{2} \log(|k_j| + 1)}. \tag{65}$$

By replacing $t$ by $t + \sum_{j=1}^{l} \sqrt{\frac{\alpha}{2} \log(k_j + 1)}$ and using Theorem 3 to bound the probability of each event and then using the union bound, we can show that

$$\mathbb{P}\left( \exists f \in \mathcal{H}_\infty : P\{\tilde{f} \leq 0\} > \inf_{\delta \in (0,1]} \left[ P_n \varphi\left(\frac{\tilde{f}}{\delta}\right) + \frac{2\sqrt{2\pi}L(\varphi)}{\delta} \Lambda(f) G_n(\mathcal{H}) \right.\right.$$
$$\left.\left. + \frac{2}{\sqrt{n}} + \left(t + \Gamma_\alpha(f) + \sqrt{\log \log_2 2\delta^{-1}}\right) \sqrt{\frac{\tau_{\min}}{n}} + B_n \right] \right) \tag{66}$$

$$\leq \mathbb{P}\left( \exists f \in \mathcal{H}_\infty : P\{\tilde{f} \leq 0\} \right.$$
$$> \inf_{\delta \in (0,1]} \left[ P_n \varphi\left(\frac{\tilde{f}}{\delta}\right) + \frac{2\sqrt{2\pi}L(\varphi)}{\delta} \Lambda(f) G_n(\mathcal{H}) \right.$$
$$\left.\left. + \left(t + \sum_{j=1}^{l} \sqrt{\frac{\alpha}{2} \log(k_j + 1)} + \sqrt{\log \log_2 2\delta^{-1}}\right) \sqrt{\frac{\tau_{\min}}{n}} + \frac{2}{\sqrt{n}} + B_n \right] \right) \tag{67}$$

$$\leq \sum_{l=0}^{\infty} \sum_{k_1 \in \mathbb{Z}\backslash\{0\}} \cdots \sum_{k_l \in \mathbb{Z}\backslash\{0\}} 2 \exp\left( -2\left(t + \sum_{j=1}^{l} \sqrt{\frac{\alpha}{2} \log(k_j + 1)}\right)^2 \right) \tag{68}$$

$$\leq \frac{\pi^2}{3} (3 - 2\zeta(\alpha))^{-1} \exp\left( -2t^2 \right), \tag{69}$$

where the last equation is followed by using some algebraic manipulations. $\qquad\square$

## 2 Extension to High-Order Markov Chains

In this section, we extend our results in previous sections to $m$-order Markov chains and a mixture of $m$ independent Markov services.

### 2.1 Extend to $m$-order Markov chain

In this subsection, we extend our results in previous sections to $m$-order homogeneous Markov chain. The main idea is to convert $m$-order homogeneous Markov chains to $1$-order homogeneous Markov chain and use our results in previous sections to bound the generalization error.

We start with the following simple example.

**Example 17.** *[m-order moving average process without noise] Consider the following m-order Markov chain*

$$X_k = \sum_{i=1}^{m} a_i X_{k-i}, \qquad k \in \mathbb{Z}_+. \tag{70}$$

*Let $Y_k := [X_{k+m-1}, X_{k+m-2}, \cdots, X_k]^T$. Then, from (70), we obtain*

$$Y_{k+1} = \mathbf{G} Y_k, \qquad \forall k \in \mathbb{Z}_+ \tag{71}$$

*where*

$$\mathbf{G} := \begin{pmatrix} a_1 & a_2 & \cdots & a_{m-1} & a_m \\ 1 & 0 & \cdots & 0 & 0 \\ 0 & 1 & \cdots & 0 & 0 \\ \vdots & \vdots & \ddots & \vdots & \vdots \\ 0 & 0 & \cdots & 1 & 0 \end{pmatrix}. \tag{72}$$

*It is clear that $\{\mathbf{Y}_n\}_{n=1}^{\infty}$ is an order-1 Markov chain. Hence, instead of directly working with the m-order Markov chain $\{\mathbf{X}_n\}_{n=1}^{\infty}$, we can find an upper bound for the Markov chain $\{Y_n\}_{n=1}^{\infty}$.*

*To derive generalization error bounds for the Markov chain $\{Y_n\}_{n=1}^{\infty}$, we can use the following arguments. For all $f \in \mathcal{F}$ and $(x_k, x_{k+1}, \cdots, x_{k+m-1}) \in \mathcal{S}^m$, by setting $\tilde{f}(x_k, x_{k+1}, \cdots, x_{k+m-1}) = f(x_k)$ where $\tilde{f} : \mathcal{S}^m \to \mathbb{R}$, we obtain*

$$\frac{1}{n} \sum_{i=1}^{n} \mathbf{1}\{f(X_i) \leq 0\} = \frac{1}{n} \sum_{i=1}^{n} \mathbf{1}\{\tilde{f}(Y_i) \leq 0\}. \tag{73}$$

*Hence, by applying all the results for 1-order Markov chain $\{Y_n\}_{n=1}^{\infty}$, we obtain corresponding upper bounds for the sequence of m-order Markov chain $\{X_n\}_{n=1}^{\infty}$.*

This approach can be extended to more general $m$-order Markov chain $X_k = g(X_{k-1}, X_{k-2} \cdots, X_{k-m})$ where $g : \mathcal{S}^m \to \mathbb{R}$. More specifically, for any tuple $(x_1, x_2, \cdots, x_m) \in \mathcal{S}^m$, observe that

$$dg = \frac{\partial g}{\partial x_1} dx_1 + \frac{\partial g}{\partial x_2} dx_2 + \cdots + \frac{\partial g}{\partial x_m} dx_m. \tag{74}$$

Hence, if $\frac{\partial g}{\partial x_i} = \alpha_i$ for some constant $\alpha_i$ and for each $i \in [m]$, from (74), we have

$$g(x_1, x_2, \cdots, x_m) = g(c_1, c_2, \cdots, c_m) + \sum_{i=1}^{m} a_i x_i + \sum_{i=1}^{m} a_i \nu_i, \tag{75}$$

where $\nu_i$'s are constants. One specific example where the function $g : \mathcal{S}^m \to \mathbb{R}$ satisfies this property is $g(x_1, x_2, \cdots, x_m) = a_1 x_1 + a_2 x_2 + \cdots a_m x_m$, where $a_1, a_2, \cdots, a_m$ are constants as in Example 17.

Now, by choosing $u = (g(c_1, c_2, \cdots, c_m) + \sum_{i=1}^{m} \alpha_i \nu_i)/(1 - \sum_{i=1}^{m} a_i \nu_i)$, from (75), we have

$$g(x_1, x_2, \cdots, x_m) + u = \sum_{i=1}^{m} a_i (x_i + u). \tag{76}$$

By setting $Y_k = [X_{k+m-1} + u \quad X_{k+m-2} + u \quad \cdots \quad X_k + u]^T$, from (76), we have:

$$Y_{k+1} = \begin{pmatrix} a_1 & a_2 & \cdots & a_{m-1} & a_m \\ 1 & 0 & \cdots & 0 & 0 \\ 0 & 1 & \cdots & 0 & 0 \\ \vdots & \vdots & \ddots & \vdots & \vdots \\ 0 & 0 & \cdots & 1 & 0 \end{pmatrix} Y_k. \tag{77}$$

In a more general setting, if $X_k = g(X_{k-1}, \cdots, X_{k-m}, V_k)$ for some random variable $V_k$ which is independent of $\{X_{k-i}\}_{i=1}^m$ such as the Autoregressive model (AR), where

$$X_k = c + \sum_{i=1}^m a_i X_{t-i} + V_k, \tag{78}$$

we can use the following conversion procedure. First, by using Taylor's approximation (to the first-order), we obtain

$$g(x_1, x_2, \cdots, x_m, \xi) \approx g(c_1, c_2, \cdots, c_m, \xi_0) + \sum_{i=1}^m \left.\frac{\partial g}{\partial x_i}\right|_{(c_1, c_2, \cdots, c_m, \zeta_0)} (x_i - c_i)$$
$$+ \left.\frac{\partial g}{\partial v}\right|_{(c_1, c_2, \cdots, c_m, \xi_0)} (\xi - \xi_0) \tag{79}$$

for some good choice of $(c_1, c_2, \cdots, c_m, \xi_0) \in \mathcal{S}^m \times \mathcal{V}$, where $\mathcal{V}$ is the alphabet of $V_k$. Using the above trick with $Y_k = [X_{k+m-1} + u \quad X_{k+m-2} + u \quad \cdots \quad X_k + u]^T$, $a_i = \left.\frac{\partial g}{\partial x_i}\right|_{(c_1, c_2, \cdots, c_m, \xi_0)}$, we can replace the recursion $X_k = g(X_{k-1}, \cdots, X_{k-m}, V_k)$ by the following equivalent recursion:

$$Y_{k+1} = \begin{pmatrix} a_1 & a_2 & \cdots & a_{m-1} & a_m \\ 1 & 0 & \cdots & 0 & 0 \\ 0 & 1 & \cdots & 0 & 0 \\ \vdots & \vdots & \ddots & \vdots & \vdots \\ 0 & 0 & \cdots & 1 & 0 \end{pmatrix} Y_k + \begin{pmatrix} \left.\frac{\partial g}{\partial v}\right|_{(c_1, c_2, \cdots, c_m, \xi_0)} V_k \\ 0 \\ 0 \\ \vdots \\ 0 \end{pmatrix}. \tag{80}$$

Since $V_k$ is independent of $\{X_{k+m-i}\}_{i=2}^{m+1}$ or $Y_k$, (80) models a new 1-order Markov chain $\{Y_k\}_{k=1}^\infty$. Then, by using the the same arguments to obtain (73), we can derive bounds on generalization error for this model.

For a general $m$-order homogeneous Markov chain, it holds that

$$P_{X_k | X_{k-1} = x_1, X_{k-2} = x_2, \cdots, X_{k-m} = x_m} \sim T_{(x_1, x_2, \cdots, x_m)} \tag{81}$$

for all $(x_1, x_2, \cdots, x_m) \in \mathcal{S}^m$, where $T_{(x_1, x_2, \cdots, x_m)}$ is a random variable which depends only on $x_1, x_2, \cdots, x_m$ and does not depend on $k$. Hence, we can represent the

$$X_k = \tilde{g}(X_{k-1}, X_{k-2}, \cdots, X_{k-m}, T_{(X_{k-1}, X_{k-2}, \cdots, X_{k-m})}), \tag{82}$$

where $T_{(X_{k-1}, X_{k-2}, \cdots, X_{k-m})} = f(\varepsilon_k, V_k, V_{k-1}, \cdots, V_{k-q}, X_{k-1}, X_{k-2}, \cdots, X_{k-m})$. Here, $\varepsilon_k$ represents new noise at time $k$ which is independent of the past. Hence, in a general $m$-order homogeneous Markov chain, we can represent the $m$-order homogeneous Markov chain by the following recursion:

$$X_k = g(X_{k-1}, X_{k-2}, \cdots, X_{k-m}, \varepsilon_k, V_k, V_{k-1}, \cdots, V_{k-q}), \tag{83}$$

where $\varepsilon_k$ represents new noise at time $k$ and $q \in \mathbb{Z}_+$. By using Taylor expansion to the first order, we can approximate the Markov chain in (83) by an Autoregressive Moving-Average Model (ARMA$(m, q)$) model as following:

$$X_k = c + \varepsilon_k + \sum_{i=1}^m a_i X_{k-i} + \sum_{i=1}^q \theta_i \varepsilon_{k-i}, \tag{84}$$

where $c$ and $a_i$'s are constants, and $\{\varepsilon_k\}_{k=1}^\infty$ are i. i. d. Gaussian random variables $\mathcal{N}(0, \sigma^2)$. For this model, let

$$V_k := \sum_{i=1}^k \varepsilon_i \tag{85}$$

and

$$Y_k := \begin{pmatrix} X_{k+m-1} + u \\ X_{k+m-2} + u \\ \vdots \\ X_k + u \\ V_{k+m-1} \\ V_{k+m-2} \\ \vdots \\ V_{k+m-q} \\ V_{k+m-q-1} \end{pmatrix}, \tag{86}$$

where

$$u := \frac{c}{1 - \sum_{i=1}^{m} a_i}. \tag{87}$$

Let $V_k := \sum_{i=1}^{k} \varepsilon_i$ for all $k \geq 1$. Observe that

$$V_{k+m} = \varepsilon_{k+m} + V_{k+m-1}. \tag{88}$$

On the other hand, we have

$$X_{k+m} = c + \varepsilon_{k+m} + \sum_{i=1}^{m} a_i X_{k+m-i} + \sum_{i=1}^{q} \theta_i \varepsilon_{k+m-i} \tag{89}$$

$$= c + \varepsilon_{k+m} + \sum_{i=1}^{m} a_i X_{k+m-i} + \sum_{i=1}^{q} \theta_i \big( V_{k+m-i} - V_{k+m-1-i} \big) \tag{90}$$

$$= c + \varepsilon_{k+m} + \sum_{i=1}^{m} a_i X_{k+m-i} + \theta_1 V_{k+m-1} + \sum_{i=1}^{q-1} \big( \theta_{i+1} - \theta_i \big) V_{k+m-1-i} - \theta_q V_{k+m-q-1}. \tag{91}$$

Then, we have

$$Y_{k+1} = \mathbf{G} Y_k + \begin{pmatrix} \varepsilon_{k+m} \\ 0 \\ \vdots \\ 0 \\ \varepsilon_{k+m} \\ 0 \\ \vdots \\ 0 \end{pmatrix}, \tag{92}$$

where

$$\mathbf{G} := \begin{pmatrix} \mathbf{G}_{11} & \mathbf{G}_{12} \\ \mathbf{G}_{21} & \mathbf{G}_{22} \end{pmatrix}. \tag{93}$$

Here,

$$\mathbf{G}_{11} := \begin{pmatrix} a_1 & a_2 & \cdots & a_{m-1} & a_m \\ 1 & 0 & \cdots & 0 & 0 \\ 0 & 1 & \cdots & 0 & 0 \\ \vdots & \vdots & \ddots & \vdots & \vdots \\ 0 & 0 & \cdots & 1 & 0 \end{pmatrix}_{m \times q+1}, \tag{94}$$

$$\mathbf{G}_{12} := \begin{pmatrix} \theta_1 & \theta_2 - \theta_1 & \cdots & \theta_q - \theta_{q-1} & -\theta_q \\ 0 & 0 & \cdots & 0 & 0 \\ 0 & 0 & \cdots & 0 & 0 \\ \vdots & \vdots & \ddots & \vdots & \vdots \\ 0 & 0 & \cdots & 0 & 0 \end{pmatrix}_{m \times q+1}, \tag{95}$$

$$\mathbf{G}_{21} := \begin{pmatrix} 0 & 0 & \cdots & 0 & 0 \\ 0 & 0 & \cdots & 0 & 0 \\ 0 & 0 & \cdots & 0 & 0 \\ \vdots & \vdots & \ddots & \vdots & \vdots \\ 0 & 0 & \cdots & 0 & 0 \end{pmatrix}_{q+1 \times m} , \tag{96}$$

and

$$\mathbf{G}_{22} := \begin{pmatrix} 1 & 0 & \cdots & 0 & 0 \\ 1 & 0 & \cdots & 0 & 0 \\ 0 & 1 & \cdots & 0 & 0 \\ \vdots & \vdots & \ddots & \vdots & \vdots \\ 0 & 0 & \cdots & 1 & 0 \end{pmatrix}_{q+1 \times q+1} . \tag{97}$$

Since $\varepsilon_{k+m}$ is independent of $Y_k$, (92) models a 1-order Markov chain. Hence, we can use the above arguments to derive new generalization error bounds for the $m$-order homogeneous Markov chain where ARMA model is a special case.

## 2.2 Mixture of $m$ Services

In this section, we consider the case that $Y_k = \sum_{l=1}^{m} \alpha_l X_k^{(l)}$ for all $k = 1, 2, \cdots$, where $\{X_k^{(l)}\}_{k=1}^{\infty}$ are independent Markov chains on $\mathcal{S}$ with stationary distribution for all $l \in [m]$. This setting usually happens in practice, for example, video is a mixture of voice, image, and text, where each service can be modelled as a high-order Markov chain and the order of the Markov chain depends on the type of service.

Let

$$Z_k := \begin{pmatrix} \alpha_1 X_k^{(1)} + \alpha_2 X_k^{(2)} + \cdots \alpha_m X_k^{(m)} \\ \alpha_2 X_k^{(2)} + \cdots \alpha_m X_k^{(m)} \\ \vdots \\ \alpha_m X_k^{(m)} \end{pmatrix} = \begin{pmatrix} Y_k \\ \alpha_2 X_k^{(2)} + \cdots \alpha_m X_k^{(m)} \\ \vdots \\ \alpha_m X_k^{(m)} \end{pmatrix}. \tag{98}$$

Then, it holds that

$$Z_k = \mathbf{G} X_k \tag{99}$$

where

$$\mathbf{G} := \begin{pmatrix} \alpha_1 & \alpha_2 & \cdots & \alpha_{m-1} & \alpha_m \\ 0 & \alpha_2 & \cdots & \alpha_{m-1} & \alpha_m \\ \vdots & \vdots & \vdots & \vdots & \vdots \\ 0 & 0 & \cdots & 0 & \alpha_m \end{pmatrix}, \tag{100}$$

and

$$X_k := \begin{pmatrix} X_k^{(1)} \\ X_k^{(2)} \\ \vdots \\ X_k^{(m)} \end{pmatrix}. \tag{101}$$

It is obvious that $\mathbf{G}$ is non-singular since $\det(\mathbf{G}) = \prod_{l=1}^{m} \alpha_l \neq 0$. Therefore, for fixed pair $(x, y) \in \mathcal{S}^m \times \mathcal{S}^m$, we have

$$\mathbb{P}\big(Z_{k+1} = y \big| Z_k = x\big) = \mathbb{P}\big(X_{k+1} = \mathbf{G}^{-1} y \big| X_k = \mathbf{G}^{-1} x\big). \tag{102}$$

Now, assume that $\mathbf{G}^{-1}x = \left(\beta_x^{(1)}, \beta_x^{(2)}, \cdots \beta_x^{(m)})\right)$ and $\mathbf{G}^{-1}y = \left(\beta_y^{(1)}, \beta_y^{(2)}, \cdots \beta_y^{(m)})\right)$. Then, from (101) and (102), we have

$$\mathbb{P}\big(Z_{k+1} = y | Z_k = x\big) = \mathbb{P}\bigg(\bigcap_{l=1}^{m} \{X_{k+1}^{(l)} = \beta_y^{(l)}\} \bigg| \bigcap_{l=1}^{m} \{X_k^{(l)} = \beta_x^{(l)}\}\bigg) \tag{103}$$

$$= \prod_{l=1}^{m} \mathbb{P}\big(X_{k+1}^{(l)} = \beta_y^{(l)} | X_k^{(l)} = \beta_x^{(l)}\big) \tag{104}$$

$$= \prod_{l=1}^{m} Q_l(\beta_x^{(l)}, \beta_y^{(l)}), \tag{105}$$

where $Q_l$ is the transition probability of the Markov chain $l$. It follows that $\{Z_k\}_{k=1}^{\infty}$ is a 1-order Markov chain. It is easy to see that $\{Z_k\}_{k=1}^{\infty}$ has stationary distribution if all the Markov chains $\{X_k^{(l)}\}_{l=1}^{m}$ have stationary distributions.

Now, as Subsection 2.1, to derive generalization error bounds for the Markov chain $\{X_n\}_{n=1}^{\infty}$, we can use the following arguments. For all $f \in \mathcal{F}$ and by setting $\tilde{f}(z_1, z_2, \cdots, z_m) := f(z_1)$ where $\tilde{f} : \mathcal{S}^m \to \mathbb{R}$, we obtain $\tilde{f}(GX_k) = f(Y_k)$ and

$$\frac{1}{n}\sum_{i=1}^{n} \mathbf{1}\{f(Y_i) \le 0\} = \frac{1}{n}\sum_{i=1}^{n} \mathbf{1}\{\tilde{f}(\mathbf{G}X_i) \le 0\}. \tag{106}$$

Hence, by applying all the results for 1-order Markov chain $\{Z_n\}_{n=1}^{\infty}$ where $Z_n = \mathbf{G}X_n$, we obtain corresponding upper bounds for the sequence of $m$-order Markov chain $\{X_n\}_{n=1}^{\infty}$.

## A   Proof of Lemma 1

Before going to prove Lemma 1, we observe the following interesting fact.

**Lemma 18.** *Let $\{X_n\}_{n=1}^{\infty}$ be an arbitrary process on a Polish space $\mathcal{S}$, and let $\{Y_n\}_{n=1}^{\infty}$ be a independent copy (replica) of $\{X_n\}_{n=1}^{\infty}$. Denote by $\mathbf{X} = (X_1, X_2, \cdots, X_n)$, $\mathbf{Y} = (Y_1, Y_2, \cdots, Y_n)$, and $\mathcal{F}$ a class of uniformly bounded functions from $\mathcal{S} \to \mathbb{R}$. Let $\boldsymbol{\epsilon} := (\varepsilon_1, \varepsilon_2, \cdots, \varepsilon_n)$ be a vector of i.i.d. Rademacher's random variables. Then, the following holds:*

$$\mathbb{E}_{\boldsymbol{\epsilon}}\left[\mathbb{E}_{\mathbf{X},\mathbf{Y}}\left[\sup_{f\in\mathcal{F}}\left|\sum_{i=1}^{n}\varepsilon_i(f(X_i) - f(Y_i))\right|\right]\right] = \mathbb{E}_{\mathbf{X},\mathbf{Y}}\left[\sup_{f\in\mathcal{F}}\left|\sum_{i=1}^{n}(f(X_i) - f(Y_i))\right|\right]. \tag{107}$$

**Remark 19.** *Our lemma generalizes a similar fact for i.i.d. processes. In the case that $\{X_n\}_{n=1}^{\infty}$ is an i.i.d. random process, (107) holds with equality since $P_{X^n,Y^n}(x_1, x_2, \cdots, x_n, y_1, y_2, \cdots, y_n)$ is invariant under permutation. However, for the Markov case, this fact does not hold in general. Hence, in the following, we provide a new proof for (107), which works for any process $\{X_n\}_{n=1}^{\infty}$ by making use of the properties of Rademacher's process.*

*Proof.* Let $g(x, y) := f(x) - f(y)$ and $\mathcal{G} := \{g : \mathcal{S} \times \mathcal{S} \to \mathbb{R} : g(x, y) := f(x) - f(y) \text{ for some } f \in \mathcal{F}\}$. Then, it holds that

$$\mathbb{E}_{\boldsymbol{\epsilon}}\left[\mathbb{E}_{\mathbf{X},\mathbf{Y}}\left[\sup_{f\in\mathcal{F}}\left|\sum_{i=1}^{n}\varepsilon_i(f(X_i) - f(Y_i))\right|\right]\right] = \mathbb{E}_{\boldsymbol{\epsilon}}\left[\mathbb{E}_{\mathbf{X},\mathbf{Y}}\left[\sup_{g\in\mathcal{G}}\left|\sum_{i=1}^{n}\varepsilon_i g(X_i, Y_i)\right|\right]\right]. \tag{108}$$

Observe that $g(X_i, Y_i) = -g(Y_i, X_i)$ for all $i \in [n]$.

For all $j \in [n]$, denote by

$$\mathcal{N}_j := [n] \setminus \{j\}, \tag{109}$$

and

$$\varepsilon_{\mathcal{N}_j} := \{\varepsilon_i : i \in \mathcal{N}_j\}. \tag{110}$$

Now, for each $j \in [n]$, observe that $\varepsilon_j$ is independent of $X_1^n, Y_1^n, \varepsilon_{\mathcal{N}_j}$. Hence, we have

$$
\mathbb{E}_{\varepsilon_1^n, X_1^n, Y_1^n} \left[ \sup_{g \in \mathcal{G}} \left| \sum_{i=1}^n \varepsilon_i g(X_i, Y_i) \right| \right]
$$

$$
= \frac{1}{2} \mathbb{E}_{\varepsilon_{\mathcal{N}_j}, X_1^n, Y_1^n} \left[ \sup_{g \in \mathcal{G}} \left| \sum_{i \in \mathcal{N}_j} \varepsilon_i g(X_i, Y_i) + g(X_j, Y_j) \right| \right]
$$

$$
+ \frac{1}{2} \mathbb{E}_{\varepsilon_{\mathcal{N}_j}, X_1^n, Y_1^n} \left[ \sup_{g \in \mathcal{G}} \left| \sum_{i \in \mathcal{N}_j} \varepsilon_i g(X_i, Y_i) - g(X_j, Y_j) \right| \right] \tag{111}
$$

$$
= \frac{1}{2} \mathbb{E}_{\varepsilon_{\mathcal{N}_j}, X_1^n, Y_1^n} \left[ \sup_{g \in \mathcal{G}} \left| \sum_{i \in \mathcal{N}_j} \varepsilon_i g(X_i, Y_i) + g(X_j, Y_j) \right| \right]
$$

$$
+ \frac{1}{2} \mathbb{E}_{\varepsilon_{\mathcal{N}_j}, X_1^n, Y_1^n} \left[ \sup_{g \in \mathcal{G}} \left| \sum_{i \in \mathcal{N}_j} \varepsilon_i g(X_i, Y_i) + g(Y_j, X_j) \right| \right] \tag{112}
$$

where (112) follows from $g(X_j, Y_j) = -g(Y_j, X_j)$.

Now, by setting $\tilde{\epsilon}_i := -\varepsilon_i$ for all $i \in n$. Then, we obtain

$$
\mathbb{E}_{\varepsilon_{\mathcal{N}_j}, X_1^n, Y_1^n} \left[ \sup_{g \in \mathcal{G}} \left| \sum_{i \in \mathcal{N}_j} \varepsilon_i g(X_i, Y_i) + g(Y_j, X_j) \right| \right]
$$

$$
= \mathbb{E}_{\tilde{\varepsilon}_{\mathcal{N}_j}, X_1^n, Y_1^n} \left[ \sup_{g \in \mathcal{G}} \left| \sum_{i \in \mathcal{N}_j} \tilde{\epsilon}_i g(X_i, Y_i) + g(Y_j, X_j) \right| \right] \tag{113}
$$

$$
= \mathbb{E}_{\varepsilon_{\mathcal{N}_j}, X_1^n, Y_1^n} \left[ \sup_{g \in \mathcal{G}} \left| \sum_{i \in \mathcal{N}_j} \varepsilon_i g(Y_i, X_i) + g(Y_j, X_j) \right| \right] \tag{114}
$$

$$
= \mathbb{E}_{\varepsilon_{\mathcal{N}_j}, X_1^n, Y_1^n} \left[ \sup_{g \in \mathcal{G}} \left| \sum_{i \in \mathcal{N}_j} \varepsilon_i g(X_i, Y_i) + g(X_j, Y_j) \right| \right], \tag{115}
$$

where (113) follows from $(\tilde{\epsilon}_i : i \in \mathcal{N}_j)$ has the same distribution as $(\varepsilon_i : i \in \mathcal{N}_j)$, (114) follows from $g(X_i, Y_i) = -g(Y_i, X_i)$ and $\tilde{\epsilon}_i = -\varepsilon_i$, and (115) follows from $g(X_i, Y_i) = -g(Y_i, X_i)$ for all $i \in [n]$.

From (112) and (115), we have

$$
\mathbb{E}_{\varepsilon_1^n, X_1^n, Y_1^n} \left[ \sup_{g \in \mathcal{G}} \left| \sum_{i=1}^n \varepsilon_i g(X_i, Y_i) \right| \right] = \mathbb{E}_{\varepsilon_{\mathcal{N}_j}, X_1^n, Y_1^n} \left[ \sup_{g \in \mathcal{G}} \left| \sum_{i \in \mathcal{N}_j} \varepsilon_i g(X_i, Y_i) + g(X_j, Y_j) \right| \right] \quad \forall j \in [n].
$$

$$\tag{116}$$

It follows that

$$
\mathbb{E}_{\varepsilon_1^n, X_1^n, Y_1^n} \left[ \sup_{g \in \mathcal{G}} \left| \sum_{i=1}^n \varepsilon_i g(X_i, Y_i) \right| \right]
$$

$$
= \mathbb{E}_{\varepsilon_1^{n-1}, X_1^n, Y_1^n} \left[ \sup_{g \in \mathcal{G}} \left| \sum_{i=1}^{n-1} \varepsilon_i g(X_i, Y_i) + g(X_n, Y_n) \right| \right] \tag{117}
$$

$$
= \frac{1}{2} \mathbb{E}_{\varepsilon_1^{n-2}, X_1^n, Y_1^n} \left[ \sup_{g \in \mathcal{G}} \left| \sum_{i=1}^{n-2} \varepsilon_i g(X_i, Y_i) + g(X_{n-1}, Y_{n-1}) + g(X_n, Y_n) \right| \right]
$$

$$
+ \frac{1}{2} \mathbb{E}_{\varepsilon_1^{n-2}, X_1^n, Y_1^n} \left[ \sup_{g \in \mathcal{G}} \left| \sum_{i=1}^{n-2} \varepsilon_i g(X_i, Y_i) - g(X_{n-1}, Y_{n-1}) + g(X_n, Y_n) \right| \right] \tag{118}
$$

$$
= \frac{1}{2} \mathbb{E}_{\varepsilon_1^{n-2}, X_1^n, Y_1^n} \left[ \sup_{g \in \mathcal{G}} \left| \sum_{i=1}^{n-2} \varepsilon_i g(X_i, Y_i) + g(X_{n-1}, Y_{n-1}) + g(X_n, Y_n) \right| \right]
$$

$$+ \frac{1}{2}\mathbb{E}_{\varepsilon_1^{n-2}, X_1^n, Y_1^n}\left[\sup_{g\in\mathcal{G}}\left|\sum_{i=1}^{n-2}\varepsilon_i g(X_i, Y_i) + g(Y_{n-1}, X_{n-1}) + g(X_n, Y_n)\right|\right], \quad (119)$$

where (117) follows from setting $j = n$ in (116), and (119) follows from $g(Y_{n-1}, X_{n-1}) = -g(X_{n-1}, Y_{n-1})$.

Now, for any fixed tuple $(x_1^{n-1}, y_1^{n-1}, \varepsilon_1^{n-1}) \in \mathcal{S}^{n-1} \times \mathcal{S}^{n-1} \times \{-1, 1\}^{n-1}$, observe that

$$P_{X_n, Y_n | X_1^{n-1}, Y_1^{n-1}, \varepsilon_1^{n-2}}(x_n, y_n | x_1^{n-1}, y_1^{n-1}, \varepsilon_1^{n-2})$$

$$= P_{X_n | X_1^{n-1}}(x_n | x_1^{n-1}) P_{Y_n | Y_1^{n-1}}(y_n | y_1^{n-1}) \quad (120)$$

$$= P_{Y_n | Y_1^{n-1}}(x_n | x_1^{n-1}) P_{X_n | X_1^{n-1}}(y_n | y_1^{n-1}) \quad (121)$$

$$= P_{Y_n, X_n | Y_1^{n-1}, X_1^{n-1}, \varepsilon_1^{n-2}}(x_n, y_n | x_1^{n-1}, y_1^{n-1}, \varepsilon_1^{n-2}). \quad (122)$$

On the other hand, we also have

$$P_{X_1^{n-1}, Y_1^{n-1}, \varepsilon_1^{n-2}}(x_1^{n-1}, y_1^{n-1}, \varepsilon_1^{n-2})$$

$$= P_{\varepsilon_1^{n-2}}(\varepsilon_1^{n-2}) P_{X_1^{n-1}}(x_1^{n-1}) P_{Y_1^{n-1}}(y_1^{n-1}) \quad (123)$$

$$= P_{\varepsilon_1^{n-2}}(\varepsilon_1^{n-2}) P_{Y_1^{n-1}}(x_1^{n-1}) P_{X_1^{n-1}}(y_1^{n-1}) \quad (124)$$

$$= P_{Y_1^{n-1}, X_1^{n-1}, \varepsilon_1^{n-2}}(x_1^{n-1}, y_1^{n-1}, \varepsilon_1^{n-2}). \quad (125)$$

Hence, from (122) and (125), we obtain

$$P_{X_1^n, Y_1^n, \varepsilon_1^{n-2}}(x_1^n, y_1^n, \varepsilon_1^{n-2})$$

$$= P_{X_1^{n-1} Y_1^{n-1} \varepsilon_1^{n-2}}(x_1^{n-1}, y_1^{n-1}, \varepsilon_1^{n-2}) P_{X_n Y_n | X_1^{n-1} Y_1^{n-1} \varepsilon_1^{n-2}}(x_n, y_n | x_1^{n-1}, y_1^{n-1}, \varepsilon_1^{n-2})$$

$$\quad (126)$$

$$= P_{Y_1^{n-1}, X_1^{n-1}, \varepsilon_1^{n-2}}(x_1^{n-1}, y_1^{n-1}, \varepsilon_1^{n-2}) P_{Y_n, X_n | Y_1^{n-1}, X_1^{n-1}, \varepsilon_1^{n-2}}(x_n, y_n | x_1^{n-1}, y_1^{n-1}, \varepsilon_1^{n-2})$$

$$\quad (127)$$

$$= P_{Y_1^n, X_1^n, \varepsilon_1^{n-2}}(x_1^n, y_1^n, \varepsilon_1^{n-2}). \quad (128)$$

Now, from (128), we also have

$$P_{X_1^n, Y_1^n}(x_1^n, y_1^n) = P_{Y_1^n X_1^n}(x_1^n, y_1^n). \quad (129)$$

It follows from (129) that

$$P_{X_{n-1}, Y_{n-1}}(x_{n-1}, y_{n-1}) = \sum_{x_1^{n-2}, y_1^{n-2}, x_n, y_n} P_{X_1^n, Y_1^n}(x_1^n, y_1^n) \quad (130)$$

$$= \sum_{x_1^{n-2}, y_1^{n-2}, x_n, y_n} P_{Y_1^n X_1^n}(x_1^n, y_1^n) \quad (131)$$

$$= P_{Y_{n-1} X_{n-1}}(x_{n-1}, y_{n-1}). \quad (132)$$

Hence, from (128) and (132), we have

$$P_{X_n, Y_n, X_1^{n-2}, Y_1^{n-2}, \varepsilon_1^{n-2} | X_{n-1}, Y_{n-1}}(x_n, y_n, x_1^{n-2}, y_1^{n-2}, \varepsilon_1^{n-2} | x_{n-1}, y_{n-1}) \quad (133)$$

$$= \frac{P_{X_1^n, Y_1^n, \varepsilon_1^{n-2}}(x_1^n, y_1^n, \varepsilon_1^{n-2})}{P_{X_{n-1} Y_{n-1}}(x_{n-1}, y_{n-1})} \quad (134)$$

$$= \frac{P_{Y_1^n, X_1^n, \varepsilon_1^{n-2}}(x_1^n, y_1^n, \varepsilon_1^{n-2})}{P_{Y_{n-1} X_{n-1}}(x_{n-1}, y_{n-1})} \quad (135)$$

$$= P_{X_n, Y_n, X_1^{n-2}, Y_1^{n-2}, \varepsilon_1^{n-2} | X_{n-1}, Y_{n-1}}(y_n, x_n, y_1^{n-2}, x_1^{n-2}, \varepsilon_1^{n-2} | y_{n-1}, x_{n-1}). \quad (136)$$

From (136), for each fixed $(x_{n-1}, y_{n-1}) \in \mathcal{S} \times \mathcal{S}$, we have

$$\mathbb{E}_{\varepsilon_1^{n-2}, X_1^{n-2}, Y_1^{n-2}, X_n, Y_n} \left[ \sup_{g \in \mathcal{G}} \left| \sum_{i=1}^{n-2} \varepsilon_i g(X_i, Y_i) + g(Y_{n-1}, X_{n-1}) \right. \right.$$

$$+ g(X_n, Y_n) \Bigg| \Bigg| X_{n-1} = x_{n-1}, Y_{n-1} = y_{n-1} \Bigg]$$

$$= \mathbb{E}_{\varepsilon_1^{n-2}, X_1^{n-2}, Y_1^{n-2}, X_n, Y_n} \left[ \sup_{g \in \mathcal{G}} \left| \sum_{i=1}^{n-2} \varepsilon_i g(X_i, Y_i) + g(y_{n-1}, x_{n-1}) \right. \right.$$

$$+ g(X_n, Y_n) \Bigg| \Bigg| X_{n-1} = x_{n-1}, Y_{n-1} = y_{n-1} \Bigg]$$

$$= \mathbb{E}_{\varepsilon_1^{n-2}, X_1^{n-2}, Y_1^{n-2}, X_n, Y_n} \left[ \sup_{g \in \mathcal{G}} \left| \sum_{i=1}^{n-2} \varepsilon_i g(Y_i, X_i) + g(y_{n-1}, x_{n-1}) \right. \right.$$

$$+ g(Y_n, X_n) \Bigg| \Bigg| Y_{n-1} = x_{n-1}, X_{n-1} = y_{n-1} \Bigg] \tag{137}$$

$$= \mathbb{E}_{\varepsilon_1^{n-2}, X_1^{n-2}, Y_1^{n-2}, X_n, Y_n} \left[ \sup_{g \in \mathcal{G}} \left| \sum_{i=1}^{n-2} \varepsilon_i g(X_i, Y_i) + g(x_{n-1}, y_{n-1}) \right. \right.$$

$$+ g(X_n, Y_n) \Bigg| \Bigg| Y_{n-1} = x_{n-1}, X_{n-1} = y_{n-1} \Bigg] \tag{138}$$

where (137) follows from (136), and (138) follows from the fact that $g(x, y) = -g(y, x)$ for all $x, y \in \mathcal{S} \times \mathcal{S}$.

From (138) and (132), we obtain

$$\mathbb{E}_{\varepsilon_1^{n-2}, X_1^n, Y_1^n} \left[ \sup_{g \in \mathcal{G}} \left| \sum_{i=1}^{n-2} \varepsilon_i g(X_i, Y_i) + g(Y_{n-1}, X_{n-1}) + g(X_n, Y_n) \right| \right]$$

$$= \mathbb{E}_{\varepsilon_1^{n-2}, X_1^n, Y_1^n} \left[ \sup_{g \in \mathcal{G}} \left| \sum_{i=1}^{n-2} \varepsilon_i g(X_i, Y_i) + g(X_{n-1}, Y_{n-1}) + g(X_n, Y_n) \right| \right]. \tag{139}$$

From (119) and (139), we obtain

$$\mathbb{E}_{\varepsilon_1^n, X_1^n, Y_1^n} \left[ \sup_{g \in \mathcal{G}} \left| \sum_{i=1}^{n} \varepsilon_i g(X_i, Y_i) \right| \right]$$

$$= \mathbb{E}_{\varepsilon_1^{n-2}, X_1^n, Y_1^n} \left[ \sup_{g \in \mathcal{G}} \left| \sum_{i=1}^{n-2} \varepsilon_i g(X_i, Y_i) + g(X_{n-1}, Y_{n-1}) + g(X_n, Y_n) \right| \right]. \tag{140}$$

By using induction, we finally obtain

$$\mathbb{E}_{\varepsilon_1^n, X_1^n, Y_1^n} \left[ \sup_{g \in \mathcal{G}} \left| \sum_{i=1}^{n} \varepsilon_i g(X_i, Y_i) \right| \right] = \mathbb{E}_{X_1^n, Y_1^n} \left[ \sup_{g \in \mathcal{G}} \left| \sum_{i=1}^{n} g(X_i, Y_i) \right| \right], \tag{141}$$

or equation (107) holds. □

Next, recall the following result which was developed base on the spectral method [6]:

**Lemma 20.** *[7, Theorems 3.41] Let $X_1, X_2, \cdots, X_n$ be a stationary Markov chain on some Polish space with $L_2$ spectral gap $\lambda$ defined in Section 2.1 in the main document and the initial distribution $\nu \in \mathcal{M}_2$. Let $f \in \mathcal{F}$ and define*

$$S_{n, n_0}(f) = \frac{1}{n} \sum_{j=1}^{n} f(X_{j+n_0}) \tag{142}$$

*for all $n_0 \geq 0$. Then, it holds that*

$$\mathbb{E} \left[ \left| S_{n, n_0}(f) - \mathbb{E}_\pi [f(X)] \right|^2 \right] \leq \frac{2M}{n(1-\lambda)} + \frac{64M^2}{n^2(1-\lambda)^2} \lambda^{n_0} \left\| \frac{d\nu}{d\pi} - 1 \right\|_2. \tag{143}$$

In addition, we recall the following McDiarmid's inequality for Markov chain:

**Lemma 21.** *[10, Cor. 2.10] Let $\mathbf{X} := (X_1, X_2, \cdots, X_n)$ be a homogeneous Markov chain, taking values in a Polish state space $\Lambda = \underbrace{\mathcal{S} \times \mathcal{S} \times \cdots \times \mathcal{S}}_{n \text{ times}}$, with mixing time $t_{\min}(\varepsilon)$ (for $0 \leq \varepsilon \leq 1$). Recall the definition of $\tau_{\min}$ in Eq. (3) in the main document. Suppose that $f : \Lambda \to \mathbb{R}$ satisfies*

$$f(\mathbf{x}) - f(\mathbf{y}) \leq \sum_{i=1}^{n} c_i \mathbf{1}\{x_i \neq y_i\} \tag{144}$$

*for every $\mathbf{x}, \mathbf{y} \in \Lambda$, for some $c \in \mathbb{R}_+^n$. Then, for any $t \geq 0$,*

$$\mathbb{P}_{\mathbf{X}}\big[\big|f(\mathbf{X}) - \mathbb{E}[f(\mathbf{X})]\big| \geq t\big] \leq 2\exp\left(-\frac{2t^2}{\|c\|_2^2 \tau_{\min}}\right). \tag{145}$$

Now, we return to the proof of Lemma 1.

*Proof of Lemma 1.* For each $f \in \mathcal{F}$, observe that

$$\frac{1}{n}\sum_{i=1}^{n} f(X_i) - \int_{\mathcal{S}} \pi(x)f(x)dx$$

$$= \frac{1}{n}\sum_{i=1}^{n} f(X_i) - \mathbb{E}[f(X_i)] + \frac{1}{n}\sum_{i=1}^{n} \mathbb{E}[f(X_i)] - \int_{\mathcal{S}} \pi(x)f(x)dx. \tag{146}$$

On the other hand, we have

$$\left|\frac{1}{n}\sum_{i=1}^{n} \mathbb{E}[f(X_i)] - \int_{\mathcal{S}} \pi(x)f(x)dx\right|$$

$$= \mathbb{E}\left[\left|S_{n,0}(f) - \mathbb{E}_\pi[f(X)]\right|\right] \tag{147}$$

$$\leq \sqrt{\mathbb{E}\left[\left|S_{n,0}(f) - \mathbb{E}_\pi[f(X)]\right|^2\right]} \tag{148}$$

$$\leq \sqrt{\frac{2M}{n(1-\lambda)} + \frac{64M^2}{n^2(1-\lambda)^2}\left\|\frac{dv}{d\pi} - 1\right\|_2} \tag{149}$$

$$= A_n, \tag{150}$$

where (150) follows from Lemma 20 with $n_0 = 0$.

By using $|a + b| \leq |a| + |b|$, from (146) and (150), we obtain

$$\mathbb{E}\big[\|P_n - P\|_{\mathcal{F}}\big]$$

$$\leq \mathbb{E}\left[\sup_{f \in \mathcal{F}}\left|\frac{1}{n}\sum_{i=1}^{n} f(X_i) - \mathbb{E}[f(X_i)]\right|\right] + A_n. \tag{151}$$

On the other hand, let $Y_1, Y_2, \cdots, Y_n$ is a replica of $X_1, X_2, \cdots, X_n$. It holds that

$$\mathbb{E}\left[\sup_{f \in \mathcal{F}}\left|\frac{1}{n}\sum_{i=1}^{n} f(X_i) - \mathbb{E}[f(X_i)]\right|\right]$$

$$= \mathbb{E}_{\mathbf{X}}\left[\sup_{f \in \mathcal{F}}\left|\frac{1}{n}\sum_{i=1}^{n} f(X_i) - \mathbb{E}[f(Y_i)]\right|\right] \tag{152}$$

$$= \mathbb{E}_{\mathbf{X}}\left[\sup_{f \in \mathcal{F}}\left|\mathbb{E}_{\mathbf{Y}}\left[\frac{1}{n}\sum_{i=1}^{n} f(X_i) - f(Y_i)\right]\right|\right] \tag{153}$$

$$\leq \mathbb{E}_{\mathbf{X}}\left[\mathbb{E}_{\mathbf{Y}}\left[\sup_{f \in \mathcal{F}}\left|\frac{1}{n}\sum_{i=1}^{n} f(X_i) - f(Y_i)\right|\right]\right]. \tag{154}$$

Now, by Lemma 18 and the triangle inequality for infinity norm, we have

$$\mathbb{E}\left[\sup_{f\in\mathcal{F}}\left|\frac{1}{n}\sum_{i=1}^{n}f(X_i)-f(Y_i)\right|\right]=\mathbb{E}_{\varepsilon}\mathbb{E}_{\mathbf{X},\mathbf{Y}}\left[\left\|\frac{1}{n}\sum_{i=1}^{n}\varepsilon_i\big(f(X_i)-f(Y_i)\big)\right\|_{\mathcal{F}}\right] \tag{155}$$

$$\leq\mathbb{E}_{\varepsilon}\mathbb{E}_{\mathbf{X}}\left[\left\|\frac{1}{n}\sum_{i=1}^{n}\varepsilon_i\big(f(X_i)\big)\right\|_{\mathcal{F}}\right]+\mathbb{E}_{\varepsilon}\mathbb{E}_{\mathbf{Y}}\left[\left\|\frac{1}{n}\sum_{i=1}^{n}\varepsilon_i\big(f(Y_i)\big)\right\|_{\mathcal{F}}\right] \tag{156}$$

$$=2\mathbb{E}\big[\|P_n^0\|_{\mathcal{F}}\big], \tag{157}$$

where (157) follows from the fact that $\mathbf{Y}$ is a replica of $\mathbf{X}$.

From (151) and (157), we finally obtain

$$\mathbb{E}\big[\|P_n-P\|_{\mathcal{F}}\big]\leq2\mathbb{E}\big[\|P_n^0\|_{\mathcal{F}}\big]+A_n. \tag{158}$$

Now, by using the triangle inequality, we have

$$\mathbb{E}[\|P_n^0\|_{\mathcal{F}}]=\mathbb{E}_{\mathbf{X},\varepsilon}\left[\sup_{f\in\mathcal{F}}\left|\frac{1}{n}\sum_{i=1}^{n}\varepsilon_if(X_i)\right|\right] \tag{159}$$

$$\leq\mathbb{E}\left[\sup_{f\in\mathcal{F}}\left|\frac{1}{n}\sum_{i=1}^{n}\varepsilon_i\big(f(X_i)-\mathbb{E}_{\mathbf{Y}}\big[f(Y_i)\big]\big)\right|\right]+\mathbb{E}\left[\sup_{f\in\mathcal{F}}\left|\frac{1}{n}\sum_{i=1}^{n}\varepsilon_i\mathbb{E}_{\mathbf{Y}}[f(Y_i)]\right|\right] \tag{160}$$

$$=\mathbb{E}_{\mathbf{X},\varepsilon}\left[\sup_{f\in\mathcal{F}}\left|\mathbb{E}_{\mathbf{Y}}\left[\frac{1}{n}\sum_{i=1}^{n}\varepsilon_i\big(f(X_i)-f(Y_i)\big)\right]\right|\right]+\mathbb{E}\left[\sup_{f\in\mathcal{F}}\left|\frac{1}{n}\sum_{i=1}^{n}\varepsilon_i\mathbb{E}_{\mathbf{Y}}[f(Y_i)]\right|\right] \tag{161}$$

$$\leq\mathbb{E}_{\mathbf{X},\mathbf{Y},\varepsilon}\left[\sup_{f\in\mathcal{F}}\left|\frac{1}{n}\sum_{i=1}^{n}\varepsilon_i\big(f(X_i)-f(Y_i)\big)\right|\right]+\mathbb{E}\left[\sup_{f\in\mathcal{F}}\left|\frac{1}{n}\sum_{i=1}^{n}\varepsilon_i\mathbb{E}_{\mathbf{Y}}[f(Y_i)]\right|\right] \tag{162}$$

$$=\mathbb{E}_{\mathbf{X},\mathbf{Y}}\left[\sup_{f\in\mathcal{F}}\left|\frac{1}{n}\sum_{i=1}^{n}\big(f(X_i)-f(Y_i)\big)\right|\right]+\mathbb{E}\left[\sup_{f\in\mathcal{F}}\left|\frac{1}{n}\sum_{i=1}^{n}\varepsilon_i\mathbb{E}_{\mathbf{Y}}[f(Y_i)]\right|\right], \tag{163}$$

where (163) follows from Lemma 18.

Now, we have

$$\mathbb{E}_{\mathbf{X},\mathbf{Y}}\left[\sup_{f\in\mathcal{F}}\left|\frac{1}{n}\sum_{i=1}^{n}\big(f(X_i)-f(Y_i)\big)\right|\right]$$

$$=\mathbb{E}_{\mathbf{X},\mathbf{Y}}\left[\sup_{f\in\mathcal{F}}\left|\frac{1}{n}\sum_{i=1}^{n}\big(f(X_i)-Pf\big)-\big(f(Y_i)-Pf\big)\right|\right] \tag{164}$$

$$\leq\mathbb{E}_{\mathbf{X},\varepsilon}\left[\sup_{f\in\mathcal{F}}\left|\frac{1}{n}\sum_{i=1}^{n}f(X_i)-Pf\right|\right]+\mathbb{E}_{\mathbf{Y},\varepsilon}\left[\sup_{f\in\mathcal{F}}\left|\frac{1}{n}\sum_{i=1}^{n}f(Y_i)-Pf\right|\right] \tag{165}$$

$$=2\mathbb{E}_{\mathbf{X},\varepsilon}\left[\sup_{f\in\mathcal{F}}\left|\frac{1}{n}\sum_{i=1}^{n}f(X_i)-Pf\right|\right] \tag{166}$$

$$=2\mathbb{E}\big[\|P_n-P\|_{\mathcal{F}}\big]. \tag{167}$$

On the other hand, by using the triangle inequality, we also have

$$\mathbb{E}\left[\sup_{f\in\mathcal{F}}\left|\frac{1}{n}\sum_{i=1}^{n}\varepsilon_i\mathbb{E}_{\mathbf{Y}}[f(Y_i)]\right|\right]$$

$$\leq\mathbb{E}\left[\sup_{f\in\mathcal{F}}\left|\frac{1}{n}\sum_{i=1}^{n}\varepsilon_i\big(\mathbb{E}_{\mathbf{Y}}[f(Y_i)]-\mathbb{E}_{\pi}[f(Y)]\big)\right|\right]+\mathbb{E}\left[\sup_{f\in\mathcal{F}}\left|\frac{1}{n}\sum_{i=1}^{n}\varepsilon_i\mathbb{E}_{\pi}[f(Y)]\right|\right]. \tag{168}$$

Now, observe that

$$\mathbb{E}\left[\sup_{f\in\mathcal{F}}\left|\frac{1}{n}\sum_{i=1}^{n}\varepsilon_i\mathbb{E}_\pi[f(Y)]\right|\right] \le \left(\sup_{f\in\mathcal{F}}\left|\mathbb{E}_\pi[f(Y)]\right|\right)\mathbb{E}_{\boldsymbol{\varepsilon}}\left[\sum_{i=1}^{n}\frac{1}{n}\sum_{i=1}^{n}\varepsilon_i\right] \tag{169}$$

$$\le M\mathbb{E}_{\boldsymbol{\varepsilon}}\left[\sum_{i=1}^{n}\frac{1}{n}\sum_{i=1}^{n}\varepsilon_i\right] \tag{170}$$

$$\le M\sqrt{\mathbb{E}_{\boldsymbol{\varepsilon}}\left[\left(\frac{1}{n}\sum_{i=1}^{n}\varepsilon_i\right)^2\right]} \tag{171}$$

$$= \frac{M}{\sqrt{n}}. \tag{172}$$

In addition, for each fixed $(\varepsilon_1,\varepsilon_2,\cdots,\varepsilon_n)\in\{-1,+1\}^n$ and $f\in\mathcal{F}$, we have

$$\left|\frac{1}{n}\sum_{i=1}^{n}\varepsilon_i\big(\mathbb{E}_{\mathbf{Y}}[f(Y_i)]-\mathbb{E}_\pi[f(Y)]\big)\right|$$

$$\le \frac{1}{n}\mathbb{E}_{\mathbf{Y}}\left|\sum_{i=1}^{n}\varepsilon_i\big(f(Y_i)-\mathbb{E}_\pi[f(Y)]\big)\right| \tag{173}$$

$$= \frac{1}{n}\mathbb{E}_{\mathbf{Y}}\big[\big|g_\varepsilon(\mathbf{Y})\big|\big], \tag{174}$$

where

$$g_\varepsilon(\mathbf{y}) := \sum_{i=1}^{n}\varepsilon_i\big(f(y_i)-\mathbb{E}_\pi[f(Y)]\big), \qquad \forall\mathbf{y}\in\mathbb{R}^n. \tag{175}$$

Now, for all $\mathbf{x},\mathbf{y}\in\mathbf{b}^n$, we have

$$g_\varepsilon(\mathbf{x})-g_\varepsilon(\mathbf{y}) = \sum_{i=1}^{n}\varepsilon_i\big(f(x_i)-f(y_i)\big) \tag{176}$$

$$\le \sum_{i=1}^{n}|f(x_i)-f(y_i)| \tag{177}$$

$$\le \sum_{i=1}^{n}2M\mathbf{1}\{x_i\ne y_i\}. \tag{178}$$

Hence, by Lemma 21, we have

$$\mathbb{P}_{\mathbf{Y}}\big[\big|g_\varepsilon(\mathbf{Y})\big|\ge M\sqrt{2\tau_{\min}n\log n}\big] \le \frac{2}{n}. \tag{179}$$

It follows that

$$\mathbb{E}_{\mathbf{Y}}\big[\big|g_\varepsilon(\mathbf{Y})\big|\big] \le \mathbb{E}_{\mathbf{Y}}\big[\big|g_\varepsilon(\mathbf{Y})\big|\big|\big|g_\varepsilon(\mathbf{Y})\big|<M\sqrt{2\tau_{\min}n\log n}\big]$$
$$+ 2nM\mathbb{P}_{\mathbf{Y}}\big[\big|g_\varepsilon(\mathbf{Y})\big|\ge M\sqrt{2\tau_{\min}n\log n}\big] \tag{180}$$

$$\le M\sqrt{2\tau_{\min}n\log n}+2nM\left(\frac{2}{n}\right) \tag{181}$$

$$= M\sqrt{2\tau_{\min}n\log n}+4M, \tag{182}$$

where (180) follows from the total law of expectation and $|g_\varepsilon(\mathbf{y})|\le\sum_{i=1}^{n}\big|f(y_i)-\mathbb{E}_\pi[f(Y)]\big|\le 2nM$ for all $\mathbf{y}\in\mathbb{R}^n$.

From (174) and (182), we have

$$\left|\frac{1}{n}\sum_{i=1}^{n}\varepsilon_i\big(\mathbb{E}_{\mathbf{Y}}[f(Y_i)]-\mathbb{E}_\pi[f(Y)]\big)\right| \le \frac{1}{n}\left(M\sqrt{2\tau_{\min}n\log n}+4M\right) \tag{183}$$

for all $f \in \mathcal{F}, (\varepsilon_1, \varepsilon_2, \cdots, \varepsilon_n) \in \{-1, +1\}^n$.

From (183), we obtain

$$\mathbb{E}_{\boldsymbol{\varepsilon}} \left[ \sup_{f \in \mathcal{F}} \left| \frac{1}{n} \sum_{i=1}^{n} \varepsilon_i \big( \mathbb{E}_{\mathbf{Y}}[f(Y_i)] - \mathbb{E}_{\pi}[f(Y)] \big) \right| \right] \leq \frac{1}{n} \left( M \sqrt{2\tau_{\min} n \log n} + 4M \right). \tag{184}$$

From (168), (172), and (184), we have

$$\mathbb{E} \left[ \sup_{f \in \mathcal{F}} \left| \frac{1}{n} \sum_{i=1}^{n} \varepsilon_i \mathbb{E}_{\mathbf{Y}}[f(Y_i)] \right| \right]$$

$$\leq \frac{1}{n} \left( M \sqrt{2\tau_{\min} n \log n} + 4M \right) + \frac{M}{\sqrt{n}}. \tag{185}$$

From (163), (167), and (185), we obtain

$$\mathbb{E}[\|P_n^0\|_{\mathcal{F}}] \leq 2\mathbb{E}\big[\|P_n - P\|_{\mathcal{F}}\big] + \frac{1}{n} \left( M \sqrt{2\tau_{\min} n \log n} + 4M \right) + \frac{M}{\sqrt{n}}, \tag{186}$$

and we finally have

$$\mathbb{E}\big[\|P_n - P\|_{\mathcal{F}}\big] \geq \frac{1}{2} \mathbb{E}[\|P_n^0\|_{\mathcal{F}}] - \tilde{A}_n. \tag{187}$$

$\square$

## B    Proof of Theorem 2

The proof of Theorem 2 is based on [2]. First, we prove (6). Without loss of generality, we can assume that each $\varphi \in \Phi$ takes its values in $[0, 1]$ (otherwise, it can be redefined as $\varphi \wedge 1$). Then, it is clear that $\varphi(x) = 1$ for $x \leq 0$. Hence, for each fixed $\varphi \in \Phi$ and $f \in \mathcal{F}$, we obtain

$$P\{f \leq 0\} \leq P\varphi(f) \tag{188}$$
$$\leq P_n\varphi(f) + \|P_n - P\|_{\mathcal{G}_\varphi}, \tag{189}$$

where

$$\mathcal{G}_\varphi := \big\{ \varphi \cdot f : f \in \mathcal{F} \big\}. \tag{190}$$

Now, let $g(\mathbf{x}) = \sup_{f \in \mathcal{G}_\varphi} \left| \frac{1}{n} \sum_{i=1}^{n} f(x_i) - Pf \right|$. Then, for all $\mathbf{x} \neq \mathbf{y}$, we have

$$\big| g(\mathbf{x}) - g(\mathbf{y}) \big| = \left| \sup_{f \in \mathcal{G}_\varphi} \left| \frac{1}{n} \sum_{i=1}^{n} f(x_i) - Pf \right| - \sup_{f \in \mathcal{G}_\varphi} \left| \frac{1}{n} \sum_{i=1}^{n} f(y_i) - Pf \right| \right| \tag{191}$$

$$\leq \sup_{f \in \mathcal{G}_\varphi} \left| \left| \frac{1}{n} \sum_{i=1}^{n} f(x_i) - Pf \right| - \left| \frac{1}{n} \sum_{i=1}^{n} f(y_i) - Pf \right| \right| \tag{192}$$

$$\leq \sup_{f \in \mathcal{G}_\varphi} \left| \left( \frac{1}{n} \sum_{i=1}^{n} f(x_i) - Pf \right) - \left( \frac{1}{n} \sum_{i=1}^{n} f(y_i) - Pf \right) \right| \tag{193}$$

$$\leq \sup_{f \in \mathcal{G}_\varphi} \left| \frac{1}{n} \sum_{i=1}^{n} f(x_i) - f(y_i) \right| \tag{194}$$

$$\leq \sup_{f \in \mathcal{G}_\varphi} \frac{1}{n} \sum_{i=1}^{n} \big| f(x_i) - f(y_i) \big| \tag{195}$$

$$\leq \frac{1}{n} \sum_{i=1}^{n} \mathbf{1}\{x_i \neq y_i\}. \tag{196}$$

Hence, for $t > 0$, by Lemma 21 with we have

$$\mathbb{P} \left\{ \|P_n - P\|_{\mathcal{G}_\varphi} \geq \mathbb{E}\big[\|P_n - P\|_{\mathcal{G}_\varphi}\big] + t\sqrt{\frac{\tau_{\min}}{n}} \right\} \leq 2\exp\big(-2t^2\big). \tag{197}$$

Hence, with probability at least $1 - 2\exp\left(-2t^2\right)$ for all $f \in \mathcal{F}$

$$P\{f \le 0\} \le P_n \varphi(f) + \mathbb{E}[\|P_n - P\|_{\mathcal{G}_\varphi}] + t\sqrt{\frac{\tau_{\min}}{n}}. \tag{198}$$

Now, by Lemma 1 with $M = 1$ (since $\sup_{f \in \mathcal{G}_\varphi} \|f\|_\infty = 1$), it holds that

$$\mathbb{E}\left[\|P_n - P\|_{\mathcal{G}_\varphi}\right] \le 2\mathbb{E}\left[\|P_n^0\|_{\mathcal{G}_\varphi}\right] + A_n|_{M=1} \tag{199}$$

$$= 2\mathbb{E}\left[\left\|n^{-1}\sum_{i=1}^n \varepsilon_i \delta_{X_i}\right\|_{\mathcal{G}_\varphi}\right] + B_n. \tag{200}$$

Since $(\varphi - 1)/L(\varphi)$ is contractive and $\varphi(0) - 1 = 0$, by using Talagrand's contraction lemma [3, 11], we obtain

$$\mathbb{E}_\varepsilon\left\|n^{-1}\sum_{i=1}^n \varepsilon_i \delta_{X_i}\right\|_{\mathcal{G}_\varphi} \le 2L(\varphi)\mathbb{E}_\varepsilon\left\|n^{-1}\sum_{i=1}^n \varepsilon_i \delta_{X_i}\right\|_{\mathcal{F}} \tag{201}$$

$$= 2L(\varphi)R_n(\mathcal{F}). \tag{202}$$

From (197), (198), (200), and (202), with probability $1 - 2\exp\left(-2t^2\right)$, we have for all $f \in \mathcal{F}$,

$$P\{f \le 0\} \le P_n \varphi(f) + 4L(\varphi)R_n(\mathcal{F}) + t\sqrt{\frac{\tau_{\min}}{n}} + B_n. \tag{203}$$

Now, we use (203) with $\varphi = \varphi_k$ and $t$ is replaced by $t + \sqrt{\log k}$ to obtain

$$\mathbb{P}\left(\exists f \in \mathcal{F} : P\{f \le 0\} > \inf_{k>0}\left[P_n \varphi_k(f) + 4L(\varphi_k)R_n(\mathcal{F}) + \left(t + \sqrt{\log k}\right)\sqrt{\frac{\tau_{\min}}{n}} + B_n\right]\right)$$

$$\le 2\sum_{k=1}^\infty \exp\left(-2\left(t + \sqrt{\log k}\right)^2\right) \tag{204}$$

$$\le 2\sum_{k=1}^\infty k^{-2} \exp\left(-2t^2\right) \tag{205}$$

$$= \frac{\pi^2}{3}\exp\left(-2t^2\right), \tag{206}$$

where (206) follows from

$$\frac{\pi^2}{6} = \sum_{k=1}^\infty k^{-2}. \tag{207}$$

Next, we prove (7). By the equivalence of Rademacher and Gaussian complexity [12], we have

$$\mathbb{E}\left\|n^{-1}\sum_{i=1}^n \varepsilon_i \delta_{X_i}\right\|_{\mathcal{G}_\varphi} \le \sqrt{\frac{\pi}{2}}\mathbb{E}\left\|n^{-1}\sum_{i=1}^n g_i \delta_{X_i}\right\|_{\mathcal{G}_\varphi}. \tag{208}$$

Hence, from (200) and (208), we obtain

$$\mathbb{E}\left[\|P_n - P\|_{\mathcal{G}_\varphi}\right] \le \sqrt{2\pi}\mathbb{E}\left\|n^{-1}\sum_{i=1}^n g_i \delta_{X_i}\right\|_{\mathcal{G}_\varphi} + B_n. \tag{209}$$

Now, define Gaussian processes

$$Z_1(f, \sigma) := \sigma n^{-1/2}\sum_{i=1}^n g_i(\varphi \circ f)(X_i), \tag{210}$$

and

$$Z_2(f, \sigma) := L(\varphi)n^{-1/2}\sum_{i=1}^n g_i f(X_i) + \sigma g, \tag{211}$$

where $\sigma = \pm 1$ and $g$ is standard normal independent of the sequence $\{g_i\}$. Let $\mathbb{E}_g$ be the expectation on the probability space $(\Omega_g, \Sigma_g, \mathbb{P}_g)$ on which the sequence $\{g_i\}$ and $g$ are defined, then by [2, 11], we have

$$\mathbb{E}_g\left[\sup\{Z_1(f, \sigma) : f \in \mathcal{F}, \sigma = \pm 1\}\right] \leq \mathbb{E}_g\left[\sup\{Z_2(f, \sigma) : f \in \mathcal{F}, \sigma = \pm 1\}\right]. \tag{212}$$

On the other hand, it holds that

$$\mathbb{E}_g\left\|n^{-1/2}\sum_{i=1}^{n} g_i \delta_{X_i}\right\|_{\mathcal{G}_\varphi} = \mathbb{E}_g\left[n^{-1/2}\sup_{h \in \bar{\mathcal{G}}_\varphi}\sum_{i=1}^{n} g_i h(X_i)\right] \tag{213}$$

$$= \mathbb{E}_g\left[\sup\{Z_1(f, \sigma) : f \in \mathcal{F}, \sigma = \pm 1\}\right], \tag{214}$$

where $\bar{\mathcal{G}}_\varphi := \{\varphi(f), -\varphi(f) : f \in \mathcal{F}\}$, and similarly

$$L(\varphi)\mathbb{E}_g\left\|n^{-1/2}\sum_{i=1}^{n} g_i \delta_{X_i}\right\|_{\mathcal{F}} + \mathbb{E}|g| \geq \mathbb{E}_g\left[\sup\{Z_2(f, \sigma) : f \in \mathcal{F}, \sigma = \pm 1\}\right]. \tag{215}$$

From (212), (214), and (215), we have

$$\mathbb{E}_g\left\|n^{-1}\sum_{i=1}^{n} g_i \delta_{X_i}\right\|_{\mathcal{G}_\varphi} \leq L(\varphi)\mathbb{E}_g\left\|n^{-1}\sum_{i=1}^{n} g_i \delta_{X_i}\right\|_{\mathcal{F}} + n^{-1/2}\mathbb{E}|g|. \tag{216}$$

By combining (209) and (216), we obtain

$$\mathbb{E}\left[\|P_n - P\|_{\mathcal{G}_\varphi}\right] \leq \sqrt{2\pi}\left(L(\varphi)\mathbb{E}\left[\left\|n^{-1}\sum_{i=1}^{n} g_i \delta_{X_i}\right\|_{\mathcal{F}}\right] + n^{-1/2}\mathbb{E}|g|\right) + B_n. \tag{217}$$

Hence, from (198), (209), and (217), we finally obtain (7).

## C Proof of Theorem 3

We can assume, without loss of generality, that the range of $\varphi$ is $[0, 1]$ (otherwise, we can replace $\varphi$ by $\varphi \wedge 1$). Let $\delta_k = 2^{-k}$ for all $k \geq 0$. In addition, set $\Phi = \{\varphi_k : k \geq 1\}$, where

$$\varphi_k(x) := \begin{cases} \varphi(x/\delta_k), & x \geq 0, \\ \varphi(x/\delta_{k-1}), & x < 0 \end{cases}. \tag{218}$$

Now, for any $\delta \in (0, 1]$, there exists $k$ such that $\delta \in (\delta_k, \delta_{k-1}]$. Hence, if $f(X_i) \geq 0$, it holds that $f(X_i)/\delta_k \geq f(X_i)/\delta$, so we have

$$\varphi_k(f(X_i)) = \varphi\left(\frac{f(X_i)}{\delta_k}\right) \tag{219}$$

$$\leq \varphi\left(\frac{f(X_i)}{\delta}\right), \tag{220}$$

where (220) follows from the fact that $\varphi(\cdot)$ is non-increasing.

On the other hand, if $f(X_i) < 0$, then $f(X_i)/\delta_{k-1} \geq f(X_i)/\delta$. Hence, we have

$$\varphi_k(f(X_i)) = \varphi\left(\frac{f(X_i)}{\delta_{k-1}}\right) \tag{221}$$

$$\leq \varphi\left(\frac{f(X_i)}{\delta}\right), \tag{222}$$

where (220) follows from the fact that $\varphi(\cdot)$ is non-increasing.

From (220) and (222), we have

$$P_n \varphi_k(f) = \frac{1}{n} \sum_{i=1}^{n} \varphi_k(f(X_i)) \tag{223}$$

$$\leq \frac{1}{n} \sum_{i=1}^{n} \varphi\left(\frac{f(X_i)}{\delta}\right) \tag{224}$$

$$= P_n \varphi\left(\frac{f}{\delta}\right). \tag{225}$$

Moreover, we also have

$$\frac{1}{\delta_k} \leq \frac{2}{\delta}, \tag{226}$$

and

$$\log k = \log \log_2 \frac{1}{\delta_k} \leq \log \log_2 2\delta^{-1}. \tag{227}$$

Furthermore, observe that

$$L(\varphi_k) = \sup_{x \in \mathbb{R}} \left| \frac{d\varphi_k(x)}{dx} \right| \tag{228}$$

$$= \sup_{x \in \mathbb{R}} \left| \frac{d\varphi(x/\delta_k)}{dx} \right| \mathbf{1}\{x \geq 0\} + \left| \frac{d\varphi(x/\delta_{k-1})}{dx} \right| \mathbf{1}\{x < 0\} \tag{229}$$

$$\leq \frac{L(\varphi)}{\min\{\delta_k, \delta_{k-1}\}} \tag{230}$$

$$= \frac{L(\varphi)}{\delta_k} \tag{231}$$

$$\leq \frac{2}{\delta} L(\varphi). \tag{232}$$

By combining the above facts and using Theorem 2, we obtain (8) and (9).

## D   Proof of Theorem 6

Let $\varepsilon > 0$ and $\delta > 0$. Define recursively

$$r_0 := 1, \qquad r_{k+1} = C\sqrt{r_k \varepsilon} \wedge 1, \qquad \gamma_k := \sqrt{\frac{\varepsilon}{r_k}} \tag{233}$$

some sufficiently large constant $C > 1$ (to be determined later) such that $\varepsilon < C^{-4}$. Denote by

$$\delta_0 := \delta, \tag{234}$$
$$\delta_k := \delta(1 - \gamma_0 - \cdots - \gamma_{k-1}), \tag{235}$$
$$\delta_{k,\frac{1}{2}} := \frac{1}{2}\left(\delta_k + \delta_{k+1}\right), \qquad k \geq 1. \tag{236}$$

For $k \geq 0$, let $\varphi_k$ be a continuous function from $\mathbb{R}$ into $[0,1]$ such that $\varphi_k(u) = 1$ for $u \leq \delta_{k,\frac{1}{2}}$, $\varphi_k(u) = 0$ for $u \geq \delta_k$, and linear $\delta_{k,\frac{1}{2}} \leq u \leq \delta_k$. For $k \geq 1$ let $\varphi'_k$ be a continuous function from $\mathbb{R}$ into $[0,1]$ such that $\varphi'_k(u) = 1$ for $u \leq \delta_k$, $\varphi'_k(u) = 0$ for $u \geq \delta_{k-1,\frac{1}{2}}$, and linear for $\delta_k \leq u \leq \delta_{k-1,\frac{1}{2}}$.

To begin with, we prove the following lemma.

**Lemma 22.** *Define $\mathcal{F}_0 := \mathcal{F}$, and further recursively*

$$\mathcal{F}_{k+1} := \left\{ f \in \mathcal{F}_k : P\{f \leq \delta_{k,\frac{1}{2}}\} \leq \frac{r_{k+1}}{2} \right\}. \tag{237}$$

*For all $k \geq 1$, define*

$$\mathcal{G}_k := \{\varphi_k \circ f : f \in \mathcal{F}_k\}, \qquad k \geq 0 \tag{238}$$

*and*

$$\mathcal{G}'_k := \{\varphi'_k \circ f : f \in \mathcal{F}_k\}, \qquad k \geq 1. \tag{239}$$

*Assume that*

$$E^{(k)} := \left\{ \|P_n - P\|_{\mathcal{G}_{k-1}} \leq \mathbb{E}\|P_n - P\|_{\mathcal{G}_{k-1}} + \left(K_2\sqrt{r_{k-1}\varepsilon} + K_3\varepsilon\right)\sqrt{\tau_{\min}} \right\}$$
$$\cap \left\{ \|P_n - P\|_{\mathcal{G}'_k} \leq \mathbb{E}\|P_n - P\|_{\mathcal{G}'_k} + \left(K_2\sqrt{r_k\varepsilon} + K_3\varepsilon\right)\sqrt{\tau_{\min}} \right\}, \qquad k \geq 1, \tag{240}$$

*and*

$$E_N := \bigcap_{k=1}^{N} E^{(k)}, \qquad N \geq 1. \tag{241}$$

*Then, it holds that*

$$\mathbb{P}\left[E_N^c\right] \leq 4N \exp\left(-\frac{n\varepsilon^2}{2}\right). \tag{242}$$

*Proof.* The proof is based on [2, Proof of Theorem 5].

For the case $C\sqrt{\varepsilon} \geq 1$, by a simple induction argument, we have $r_k = 1$. Now, without loss of generality, we assume that $C\sqrt{\varepsilon} < 1$. For this case, we have

$$r_k = C^{1 + 2^{-1} + \cdots + 2^{-(k-1)}} \varepsilon^{2^{-1} + \cdots + 2^{-(k-1)}} \tag{243}$$

$$= C^{2(1 - 2^{-k})} \varepsilon^{1 - 2^{-k}} \tag{244}$$

$$= (C\sqrt{\varepsilon})^{2(1 - 2^{-k})}. \tag{245}$$

From (245), it is easy to see that $r_{k+1} < r_k$ for any $k \geq 0$.

Now, observe that

$$\sum_{i=0}^{k} \gamma_i = C^{-1}\left[C\sqrt{\varepsilon} + (C\sqrt{\varepsilon})^{2^{-1}} + \cdots + (C\sqrt{\varepsilon})^{2^{-k}}\right] \tag{246}$$

$$= C^{-1}\left[(C\sqrt{\varepsilon})^{2^{-k}} + ((C\sqrt{\varepsilon})^{2^{-k}})^2 + ((C\sqrt{\varepsilon})^{2^{-k}})^{2^2} \cdots + ((C\sqrt{\varepsilon})^{2^{-k}})^{2^k}\right] \tag{247}$$

$$\leq C^{-1}\left[(C\sqrt{\varepsilon})^{2^{-k}} + ((C\sqrt{\varepsilon})^{2^{-k}})^2 + ((C\sqrt{\varepsilon})^{2^{-k}})^3 \cdots + ((C\sqrt{\varepsilon})^{2^{-k}})^k\right] \tag{248}$$

$$\leq C^{-1}\sum_{i=1}^{\infty} \left((C\sqrt{\varepsilon})^{2^{-k}}\right)^i \tag{249}$$

$$\leq C^{-1}(C\sqrt{\varepsilon})^{2^{-k}}\left(1 - (C\sqrt{\varepsilon})^{2^{-k}}\right)^{-1} \leq \frac{1}{2}, \tag{250}$$

for $\varepsilon \leq C^{-4}, C > 2(2^{\frac{1}{4}} - 1)^{-1}$ and $k \leq \log_2\log_2\varepsilon^{-1}$, where (248) follows from $i + 1 \leq 2^i$ for all $i \geq 1$ and $C\sqrt{\varepsilon} < 1$. Hence, for small enough $\varepsilon$ (note that our choice of $\varepsilon \leq C^{-4}$ implies $C\sqrt{\varepsilon} < 1$), we have

$$\gamma_0 + \gamma_1 + \cdots + \gamma_k \leq \frac{1}{2}, \qquad k \geq 1. \tag{251}$$

Therefore, for all $k \geq 1$, we get $\delta_k \in (\delta/2, \delta)$. Note also that below our choice of $k$ will be such that the restriction $k \leq \log_2\log_2\varepsilon^{-1}$ for any fixed $\varepsilon > 0$ will always be fulfilled.

From the definitions of (238) and (239), for $k \geq 1$, we have

$$\sup_{g \in \mathcal{G}_k} Pg^2 \leq \sup_{f \in \mathcal{F}_k} P\{f \leq \delta_k\} \leq \sup_{f \in \mathcal{F}_k} P\{f \leq \delta_{k-1,\frac{1}{2}}\} \leq \frac{r_k}{2} \leq r_k, \tag{252}$$

and

$$\sup_{g \in \mathcal{G}'_k} Pg^2 \leq \sup_{f \in \mathcal{F}_k} P\{f \leq \delta_{k-1,\frac{1}{2}}\} \leq \frac{r_k}{2} \leq r_k. \tag{253}$$

Since $r_0 = 1$, it is easy to see that (252) and (253) trivially holds at $k = 0$.

Now, by the union bound, from (240), we have

$$\mathbb{P}\big[\big(E^{(k)}\big)^c\big] \leq \mathbb{P}\bigg[\|P_n - P\|_{\mathcal{G}_{k-1}} > \mathbb{E}\|P_n - P\|_{\mathcal{G}_{k-1}} + K_2\sqrt{r_{k-1}\varepsilon} + K_3\varepsilon\bigg]$$
$$+ \mathbb{P}\bigg[\|P_n - P\|_{\mathcal{G}'_{k-1}} > \mathbb{E}\|P_n - P\|_{\mathcal{G}'_{k-1}} + K_2\sqrt{r_{k-1}\varepsilon} + K_3\varepsilon\bigg]. \tag{254}$$

In addition, by similar arguments which leads to (197), we have

$$\mathbb{P}\bigg\{\|P_n - P\|_{\mathcal{G}_{k-1}} \geq \mathbb{E}\big[\|P_n - P\|_{\mathcal{G}_{k-1}}\big] + u\sqrt{\frac{\tau_{\min}}{n}}\bigg\} \leq 2\exp\big(-2u^2\big). \tag{255}$$

and

$$\mathbb{P}\bigg\{\|P_n - P\|_{\mathcal{G}'_{k-1}} \geq \mathbb{E}\big[\|P_n - P\|_{\mathcal{G}'_{k-1}}\big] + u\sqrt{\frac{\tau_{\min}}{n}}\bigg\} \leq 2\exp\big(-2u^2\big). \tag{256}$$

By replacing $u = \big(K_2\sqrt{r_{k-1}\varepsilon} + K_3\varepsilon\big)\sqrt{n}$ to (255) and (256) for $K_2 > 0$ and $K_3 > 0$, we obtain

$$\mathbb{P}\bigg\{\|P_n - P\|_{\mathcal{G}_{k-1}} \geq \mathbb{E}\big[\|P_n - P\|_{\mathcal{G}_{k-1}}\big] + \big(K_2\sqrt{r_{k-1}\varepsilon} + K_3\varepsilon\big)\sqrt{\tau_{\min}}\bigg\}$$
$$\leq 2\exp\bigg(-2n(K_2\sqrt{r_{k-1}\varepsilon} + K_3)^2\bigg) \tag{257}$$

and

$$\mathbb{P}\bigg\{\|P_n - P\|_{\mathcal{G}_{k-1}} \geq \mathbb{E}\big[\|P_n - P\|_{\mathcal{G}'_{k-1}}\big] + \big(K_2\sqrt{r_{k-1}\varepsilon} + K_3\varepsilon\big)\sqrt{\tau_{\min}}\bigg\}$$
$$\leq 2\exp\bigg(-2n(K_2\sqrt{r_{k-1}\varepsilon} + K_3\varepsilon)^2\bigg). \tag{258}$$

Now, since $0 < C\sqrt{\varepsilon} \leq 1$, by (245), we have

$$r_{k-1} = (C\sqrt{\varepsilon})^{2(1-2^{-(k-1)})} \geq C^2\varepsilon. \tag{259}$$

Hence, from (254), (257), (258), and (259), that

$$\mathbb{P}\big[\big(E^{(k)}\big)^c\big] \leq 4\exp\bigg(-2n(K_2\sqrt{r_{k-1}\varepsilon} + K_3\varepsilon)^2\bigg) \tag{260}$$

$$\leq 4\exp\bigg(-2n(K_2C + K_3)^2\varepsilon^2\bigg) \tag{261}$$

$$\leq 4\exp\bigg(-\frac{n\varepsilon^2}{2}\bigg), \qquad \forall k \geq 1 \tag{262}$$

if we choose $K_2$ and $K_3$ such that

$$K_2C + K_3 \geq \frac{1}{2}. \tag{263}$$

Then, by the union bound and (262), we have

$$\mathbb{P}\big[E_N^c\big] \leq 4N\exp\bigg(-\frac{n\varepsilon^2}{2}\bigg). \tag{264}$$

$\square$

**Lemma 23.** *Let $\varepsilon > 0$ and $0 < \alpha < 2$ such that*

$$\varepsilon \geq \left(\frac{1}{n\delta^\alpha}\right)^{\frac{2}{2+\alpha}} \vee \sqrt{\frac{2\log n}{n}} \vee B_n \tag{265}$$

*for all large enough $n$, where $B_n$ is defined in (5). Denote by $\mathcal{L} := \{f \in \mathcal{F} : P_n\{f \leq \delta\} \leq \varepsilon\}$. Then, on the event $E_N \cap \mathcal{L}$, we have:*

- *(i)* $\sup_{f \in \mathcal{F}_k} P_n\{f \leq \delta_k\} \leq r_k, \qquad 0 \leq k \leq N$

*and*

- *(ii)* $\forall f \in \mathcal{L} \quad P_n\{f \leq \delta\} \leq \varepsilon \qquad \Rightarrow f \in \mathcal{F}_N$

*for all positive integer $N$ satisfying*

$$N \leq \frac{1}{\eta}\log_2\log_2\varepsilon^{-1} \qquad and \qquad r_N \geq \varepsilon, \tag{266}$$

*and $\eta$ is some implicit positive constant.*

*Proof.* We will use induction with respect to $N$. For $N = 0$, the statement is obvious. Suppose it holds for some $N \geq 0$, such that $N + 1$ still satisfies condition (266) of the lemma. Then, on the event $E_N \cap \mathcal{L}$ we have

$$(i) \qquad \sup_{f \in \mathcal{F}_k} P_n\{f \leq \delta_k\} \leq r_k, \qquad 0 \leq k \leq N \tag{267}$$

and

$$(ii) \qquad \forall f \in \mathcal{F} \quad P_n\{f \leq \delta\} \leq \varepsilon \quad \Rightarrow f \in \mathcal{F}_N. \tag{268}$$

Suppose now that $f \in \mathcal{F}$ is such that $P_n\{f \leq \delta\} \leq \varepsilon$. By the induction assumptions, on the event $E_N$ defined in (241), we have $f \in \mathcal{F}_N$. Because of this, we obtain on the event $E_{N+1}$

$$P\{f \leq \delta_{N,\frac{1}{2}}\} = P_n\{f \leq \delta_{N,\frac{1}{2}}\} + (P - P_n)\{f \leq \delta_{N,\frac{1}{2}}\} \tag{269}$$

$$\leq P_n\{f \leq \delta_N\} + (P - P_n)\{f \leq \delta_{N,\frac{1}{2}}\} \tag{270}$$

$$\leq P_n\{f \leq \delta_N\} + (P - P_n)(\varphi_N(f)) \tag{271}$$

$$\leq P_n\{f \leq \delta_N\} + \|P - P_n\|_{\mathcal{G}_N} \tag{272}$$

$$\leq \varepsilon + \mathbb{E}\|P_n - P\|_{\mathcal{G}_N} + \left(K_2\sqrt{r_N\varepsilon} + K_3\varepsilon\right)\sqrt{\tau_{\min}}. \tag{273}$$

For the class $\mathcal{G}_N$, define

$$\hat{R}_n(\mathcal{G}_N) := \left\|n^{-1}\sum_{i=1}^n \varepsilon_i\delta_{X_i}\right\|_{\mathcal{G}_N}, \tag{274}$$

where $\varepsilon_i$ is a sequence of i.i.d. Rademacher random variables. By Lemma 1, it holds that

$$\mathbb{E}\big[\|P_n - P\|_{\mathcal{G}_N}\big] \leq 2\mathbb{E}\big[\|P_n^0\|_{\mathcal{G}_N}\big] + B_n \tag{275}$$

$$= 2\mathbb{E}\big[\hat{R}_n(\mathcal{G}_N)\big] + B_n. \tag{276}$$

From (276), we have

$$\mathbb{E}\big[\|P_n - P\|_{\mathcal{G}_N}\big]$$

$$\leq 2\mathbb{E}\big[\hat{R}_n(\mathcal{G}_N)\big] + B_n \tag{277}$$

$$= 2\mathbb{E}\big[\mathbf{1}\{E_N\}\mathbb{E}_\varepsilon\big[\hat{R}_n(\mathcal{G}_N)\big]\big] + 2\mathbb{E}\big[\mathbf{1}\{E_N^c\}\mathbb{E}_\varepsilon\big[\hat{R}_n(\mathcal{G}_N)\big]\big] + B_n. \tag{278}$$

Next, by the well-known entropy inequalities for subgaussian process [12], we have

$$\mathbb{E}_\varepsilon\big[\hat{R}_n(\mathcal{G}_N)\big] \leq \inf_{g \in \mathcal{G}_N} \mathbb{E}_\varepsilon\left|n^{-1}\sum_{j=1}^n \varepsilon_j g(X_j)\right|$$

$$+ \frac{\eta}{\sqrt{n}}\int_0^{(2\sup_{g \in \mathcal{G}_N} P_n g^2)^{1/2}} H_{d_{P_n,2}}^{1/2}(\mathcal{G}_N; u)du \tag{279}$$

for some implicit positive constant $\eta > 0$.

By the induction assumption, on the event $E_N \cap \mathcal{L}$,

$$\inf_{g \in \mathcal{G}_N} \mathbb{E}_\varepsilon \left| n^{-1} \sum_{j=1}^n \varepsilon_j g(X_j) \right| \leq \inf_{g \in \mathcal{G}_N} \sqrt{\mathbb{E}_\varepsilon \left| n^{-1} \sum_{j=1}^n \varepsilon_j g(X_j) \right|^2} \tag{280}$$

$$\leq \frac{1}{\sqrt{n}} \inf_{g \in \mathcal{G}_N} \sqrt{P_n g^2} \tag{281}$$

$$\leq \frac{1}{\sqrt{n}} \inf_{f \in \mathcal{F}_N} \sqrt{P_n \{f \leq \delta_N\}} \tag{282}$$

$$\leq \sqrt{\frac{\varepsilon}{n}} \tag{283}$$

$$\leq \varepsilon, \tag{284}$$

where (283) follows from $\inf_{f \in \mathcal{F}_N} P_n\{f \leq \delta_N\} \leq \inf_{f \in \mathcal{F}_N} P_n\{f \leq \delta\} \leq P_n\{f \leq \delta\} \leq \varepsilon$ by the induction assumption with $f \in \mathcal{F}_N$.

We also have on the event $E_N \cap \mathcal{L}$, by (252), it holds that

$$\sup_{g \in \mathcal{G}_N} P_n g^2 \leq \sup_{f \in \mathcal{F}_N} P_n \{f \leq \delta_N\} \leq r_N. \tag{285}$$

The Lipschitz norm of $\varphi_{k-1}$ and $\varphi'_k$ is bounded by

$$L = 2(\delta_{k-1} - \delta_k)^{-1} = 2\delta^{-1}\gamma_{k-1}^{-1} = \frac{2}{\delta}\sqrt{\frac{r_{k-1}}{\varepsilon}} \tag{286}$$

which implies the following bound on the distance:

$$d^2_{P_n,2}(\varphi_N \circ f; \varphi_N \circ g) = n^{-1} \sum_{j=1}^n \left| \varphi_N(f(X_j)) - \varphi_N(g(X_j)) \right|^2 \tag{287}$$

$$\leq \left( \frac{2}{\delta}\sqrt{\frac{r_N}{\varepsilon}} \right)^2 d^2_{P_n,2}(f,g). \tag{288}$$

Therefore, on the event $\mathcal{E}_N \cap \mathcal{L}$,

$$\frac{1}{\sqrt{n}} \int_0^{(2\sup_{g \in \mathcal{G}_N} P_n g^2)^{1/2}} H^{1/2}_{d_{P_n},2}(\mathcal{G}_N; u) \, du$$

$$\leq \frac{1}{\sqrt{n}} \int_0^{(2r_N)^{1/2}} H^{1/2}_{d_{P_n},2}\left( \mathcal{F}; \frac{\delta\sqrt{\varepsilon}u}{2\sqrt{r_N}} \right) du \tag{289}$$

$$\leq \left( \frac{2\sqrt{D}}{1 - \alpha/2} \right) \left( \frac{r_N}{\varepsilon} \right)^{\alpha/4} \frac{r_N^{1/2 - \alpha/4}}{\sqrt{n}\delta^{\alpha/2}} \tag{290}$$

$$\leq \left( \frac{2^{1/2 + \alpha/4}\sqrt{D}}{1 - \alpha/2} \right) \frac{r_N^{1/2}}{\varepsilon^{\alpha/4}} \varepsilon^{\frac{2+\alpha}{4}} \tag{291}$$

$$= \left( \frac{2^{1/2 + \alpha/4}\sqrt{D}}{1 - \alpha/2} \right) \sqrt{r_N \varepsilon}, \tag{292}$$

where (292) follows from the condition (265), which implies that

$$\frac{1}{n^{1/2}\delta^{\alpha/2}} \leq \varepsilon^{\frac{2+\alpha}{4}}. \tag{293}$$

From (279) and (292), we obtain that on the event $E_N \cap \mathcal{L}$,

$$\mathbb{E}_\varepsilon \left[ \hat{R}_n(\mathcal{G}_N) \right] \leq \varepsilon + \frac{2^{1/2 + \alpha/4}\eta\sqrt{D}}{1 - \alpha/2} \sqrt{r_N \varepsilon}. \tag{294}$$

On the other hand, we also have

$$\mathbb{E}_\varepsilon\big[\hat{R}_n(\mathcal{G}_N)\big] \leq 1. \tag{295}$$

Hence, by combining with (294) and (295), from (278), we obtain

$$\mathbb{E}\big[\big(\big\|P_n - P\big\|_{\mathcal{G}_N}\big)\big] = 2\mathbb{E}\big[\mathbf{1}\{E_N\}\mathbb{E}_\varepsilon\big[\hat{R}_n(\mathcal{G}_N)\big]\big] + 2\mathbb{E}\big[\mathbf{1}\{E_N^c\}\mathbb{E}_\varepsilon\big[\hat{R}_n(\mathcal{G}_N)\big]\big] + B_n \tag{296}$$

$$\leq 2\bigg[\varepsilon + \eta\bigg(\frac{2^{1/2+\alpha/4}\eta\sqrt{D}}{1-\alpha/2}\bigg)\sqrt{r_N\varepsilon}\bigg] + 8N\exp\bigg(-\frac{n\varepsilon^2}{2}\bigg) + B_n. \tag{297}$$

Now, by the condition (265), it holds that

$$\varepsilon \geq B_n, \tag{298}$$

and

$$8\eta N\exp\bigg(-\frac{n\varepsilon^2}{2}\bigg) \leq \frac{8\eta N}{n} \tag{299}$$

$$\leq \frac{8\log_2\log_2\varepsilon^{-1}}{n} \tag{300}$$

$$\leq \frac{8\log_2\log_2\sqrt{\frac{n}{2\log n}}}{n} \tag{301}$$

$$\leq \eta\sqrt{\frac{2\log n}{n}} \tag{302}$$

$$\leq \eta\varepsilon, \tag{303}$$

for $n$ sufficiently large, where (299) and (301) follows from $\varepsilon \geq \sqrt{\frac{2\log n}{n}}$, (300) follows from (266), and (302) holds for $n$ sufficiently large.

From (297), (298), and (303), it holds that

$$\mathbb{E}\big[\big\|P_n - P\big\|_{\mathcal{G}_N}\big] \leq 2\bigg[\varepsilon + \eta\bigg(\frac{2^{1/2+\alpha/4}\sqrt{D}}{1-\alpha/2}\bigg)\sqrt{r_N\varepsilon}\bigg] + 2\varepsilon \tag{304}$$

$$= 4\varepsilon + 2\eta\bigg(\frac{2^{1/2+\alpha/4}\sqrt{D}}{1-\alpha/2}\bigg)\sqrt{r_N\varepsilon}. \tag{305}$$

In addition, we have

$$r_N = (C\sqrt{\varepsilon})^{2(1-2^{-N})} \geq C^2\varepsilon, \tag{306}$$

or

$$\varepsilon \leq \frac{r_N}{C^2}. \tag{307}$$

Hence, from (305) and (307), we conclude that with some constant $\tilde{\eta} > 0$,

$$\mathbb{E}\big[\big\|P_n - P\big\|_{\mathcal{G}_N}\big] \leq \tilde{\eta}\sqrt{r_N\varepsilon}. \tag{308}$$

From (273) and (308), on the event $E_{N+1} \cap \mathcal{L}$, we have

$$P\{f \leq \delta_{N,\frac{1}{2}}\} \leq \varepsilon + \tilde{\eta}\sqrt{r_N\varepsilon} + \big(K_2\sqrt{r_N\varepsilon} + K_3\varepsilon\big)\sqrt{\tau_{\min}} \tag{309}$$

$$\leq \frac{1}{2}C\sqrt{r_N\varepsilon} \tag{310}$$

$$= r_{N+1}/2, \tag{311}$$

by a proper choice of the constant $C > 0$, where (311) follows from (307). This means that $f \in \mathcal{F}_{N+1}$ and the induction step for (ii) is proved.

Now, we prove (i). We have on the event $E_{N+1}$,

$$\sup_{f\in\mathcal{F}_{N+1}} P_n\{f\le\delta_{N+1}\} \le \sup_{f\in\mathcal{F}_{N+1}} P\{f\le\delta_{N+1}\} + \sup_{f\in\mathcal{F}_{N+1}} (P_n-P)\{f\le\delta_{N+1}\} \tag{312}$$

$$\le \sup_{f\in\mathcal{F}_{N+1}} P\{f\le\delta_{N+1}\} + \|P_n-P\|_{\mathcal{G}'_{N+1}} \tag{313}$$

$$\le r_{N+1}/2 + \mathbb{E}\|P_n-P\|_{\mathcal{G}'_{N+1}} + \big(K_2\sqrt{r_{N+1}\varepsilon}+K_3\varepsilon\big)\sqrt{\tau_{\min}}. \tag{314}$$

By Lemma 1, we have

$$\mathbb{E}\big\|P_n-P\big\|_{\mathcal{G}'_{N+1}}$$

$$\le 2\mathbb{E}\big[\mathbf{1}\{E_N\}\mathbb{E}_\varepsilon\big[\hat{R}_n(\mathcal{G}'_{N+1})\big]\big] + 2\mathbb{E}\big[\mathbf{1}\{E_N^c\}\mathbb{E}_\varepsilon\big[\hat{R}_n(\mathcal{G}'_{N+1})\big]\big] + B_n \tag{315}$$

$$\le 2\mathbb{E}\big[\mathbf{1}\{E_N\}\mathbb{E}_\varepsilon\big[\hat{R}_n(\mathcal{G}'_{N+1})\big]\big] + 2\mathbb{E}\big[\mathbf{1}\{E_N^c\}\mathbb{E}_\varepsilon\big[\hat{R}_n(\mathcal{G}'_{N+1})\big]\big] + \varepsilon, \tag{316}$$

where (316) follows from the condition (265).

As above, we have

$$\mathbb{E}_\varepsilon\big[\hat{R}_n(\mathcal{G}'_{N+1})\big] \le \inf_{g\in\mathcal{G}'_{N+1}} \mathbb{E}_\varepsilon\bigg|n^{-1}\sum_{j=1}^n \varepsilon_j g(X_j)\bigg|$$

$$+ \frac{\eta}{\sqrt{n}}\int_0^{(2\sup_{g\in\mathcal{G}'_{N+1}} P_n g^2)^{1/2}} H_{d_{P_n,2}}^{1/2}\big(\mathcal{G}'_{N+1};u\big)du. \tag{317}$$

Since we already proved (i), it implies that on the event $E_{N+1}\cap\mathcal{L}$,

$$\inf_{g\in\mathcal{G}'_{N+1}} \mathbb{E}_\varepsilon\bigg|n^{-1}\sum_{j=1}^n \varepsilon_j g(X_j)\bigg| \le \inf_{g\in\mathcal{G}'_{N+1}} \sqrt{\mathbb{E}_\varepsilon\bigg|n^{-1}\sum_{j=1}^n \varepsilon_j g(X_j)\bigg|^2} \tag{318}$$

$$\le \frac{1}{\sqrt{n}}\inf_{g\in\mathcal{G}'_{N+1}} \sqrt{P_n g^2} \tag{319}$$

$$\le \frac{1}{\sqrt{n}}\inf_{f\in\mathcal{F}_{N+1}} \sqrt{P_n\{f\le\delta_{N,1/2}\}} \tag{320}$$

$$\le \frac{1}{\sqrt{n}}\inf_{f\in\mathcal{F}_N} \sqrt{P_n\{f\le\delta_{N,1/2}\}} \tag{321}$$

$$\le \sqrt{\frac{\varepsilon}{n}} \le \varepsilon. \tag{322}$$

By the induction assumption, we also have on the event $E_{N+1}\cap\mathcal{L}$,

$$\sup_{g\in\mathcal{G}'_{N+1}} P_n g^2 \le \sup_{f\in\mathcal{F}_{N+1}} P_n\{f\le\delta_{N,1/2}\} \le \frac{r_{N+1}}{2} \le r_N. \tag{323}$$

The bound for the Lipschitz norm of $\varphi'_k$ gives the following bound on the distance

$$d_{P_n,2}^2\big(\varphi'_{N+1}\circ f; \varphi'_{N+1}\circ f\big) = n^{-1}\sum_{j=1}^n \big|\varphi'_{N+1}\circ f(X_j) - \varphi'_{N+1}\circ g(X_j)\big|^2 \tag{324}$$

$$\le \bigg(\frac{2}{\delta}\sqrt{\frac{r_N}{\varepsilon}}\bigg)^2 d_{P_n,2}^2(f,g). \tag{325}$$

Therefore, on the event $E_{N+1}\cap\mathcal{L}$, we get quite similarly to (292),

$$\frac{1}{\sqrt{n}}\int_0^{(2\sup_{g\in\mathcal{G}'_{N+1}} P_n g^2)^{1/2}} H_{d_{P_n,2}}^{1/2}\big(\mathcal{G}'_{N+1};u\big)du$$

$$\le \frac{1}{\sqrt{n}}\int_0^{(2r_N)^{1/2}} H_{d_{P_n,2}}^{1/2}\bigg(\mathcal{F}; \frac{\delta\sqrt{\varepsilon}u}{2\sqrt{r_N}}\bigg)du \tag{326}$$

$$\le \bigg(\frac{2^{1/2+\alpha/4}\sqrt{D}}{1-\alpha/2}\bigg)\bigg(\frac{r_N}{\varepsilon}\bigg)^{\alpha/4}\frac{r_N^{1/2-\frac{\alpha}{4}}}{\sqrt{n}\delta^{\alpha/2}} \tag{327}$$

$$= \bigg(\frac{2^{1/2+\alpha/4}\sqrt{D}}{1-\alpha/2}\bigg)\sqrt{r_N\varepsilon}. \tag{328}$$

We collect all bounds to see that on the event $\mathcal{E}_{N+1} \cap \mathcal{L}$,

$$\sup_{f \in \mathcal{F}_{N+1}} P_n\{f \leq \delta_{N+1}\} \leq \frac{r_{N+1}}{2} + \bar{\eta}\sqrt{r_N \varepsilon} \tag{329}$$

for some constant $\bar{\eta} > 0$.

Therefore, it follows that with a proper choice of constant $C > 0$ in the recurrence relationship defining the sequence $\{r_k\}$, we have on the event $E_{N+1} \cap \mathcal{L}$

$$\sup_{f \in \mathcal{F}_{N+1}} P_n\{f \leq \delta_{N+1}\} \leq C\sqrt{r_N \varepsilon} = r_{N+1}, \tag{330}$$

which proves the induction step for (i) and, therefore, the lemma is proved. Finally, using Lemma 24 and the same arguments as [2, Proof of Theorem 5], we can prove Theorem 6. $\qquad\square$

**Lemma 24.** *Suppose that for some $\alpha \in (0, 2)$ and for some $D > 0$ such that the condition (19) holds. Then for any constant $\xi > C^2$, for all $\delta \geq 0$ and*

$$\varepsilon \geq \left(\frac{1}{n\delta^\alpha}\right)^{\frac{2}{2+\alpha}} \vee \sqrt{\frac{2\log n}{n}} \vee B_n, \tag{331}$$

*and for all large enough $n$, the following:*

$$\mathbb{P}\left[\exists f \in \mathcal{F} : P_n\{f \leq \delta\} \leq \varepsilon \quad and \quad P\left\{f \leq \frac{\delta}{2}\right\} \geq \xi\varepsilon\right] \leq 4/\eta \log_2 \log_2 \varepsilon^{-1} \exp\left\{-\frac{n\varepsilon^2}{2}\right\} \tag{332}$$

*and*

$$\mathbb{P}\left[\exists f \in \mathcal{F} : P\{f \leq \delta\} \leq \varepsilon \quad and \quad P_n\left\{f \leq \frac{\delta}{2}\right\} \geq \xi\varepsilon\right] \leq 4/\eta \log_2 \log_2 \varepsilon^{-1} \exp\left\{-\frac{n\varepsilon^2}{2}\right\}, \tag{333}$$

*where $\eta$ is some constant.*

*Proof.* Observe that

$$\mathbb{P}\left[\exists f \in \mathcal{F} : P_n\{f \leq \delta\} \leq \varepsilon \wedge P\{f \leq \delta/2\} \geq \xi\varepsilon\right]$$

$$\leq \mathbb{P}\left[\left\{\exists f \in \mathcal{F} : \{P_n\{f \leq \delta\} \leq \varepsilon\} \wedge \{P\{f \leq \delta/2\} \geq \xi\varepsilon\}\right\} \cap E_N\right] + \mathbb{P}[E_N^c] \tag{334}$$

$$\leq \mathbb{P}\left[\left\{\exists f \in \mathcal{F}_N \cap \mathcal{L}\} \wedge \{P\{f \leq \delta/2\} \geq \xi\varepsilon\}\right\} \cap E_N\right] + \mathbb{P}[E_N^c] \tag{335}$$

$$\leq \mathbb{P}\left[\left\{\exists f \in \mathcal{F}_N \cap L\} \wedge \{P\{f \leq \delta_N\} \geq \xi\varepsilon\}\right\} \cap E_N\right] + \mathbb{P}[E_N^c] \tag{336}$$

$$\leq \mathbb{P}\left[\left\{\exists f \in \mathcal{F}_N \cap L\} \wedge \{P\{f \leq \delta_N\} > r_N\}\right\} \cap E_N\right] + \mathbb{P}[E_N^c] \tag{337}$$

$$= \mathbb{P}[E_N^c] \tag{338}$$

$$\leq 4N \exp\left(-\frac{n\varepsilon^2}{2}\right) \tag{339}$$

$$\leq 4/\eta\left(\log_2 \log_2 \varepsilon^{-1}\right) \exp\left(-\frac{n\varepsilon^2}{2}\right), \tag{340}$$

where (335) follows from (ii) in Lemma 23, (337) follows from $r_N \leq (C\sqrt{\varepsilon})^2 < \xi\varepsilon$ for some constant $\xi > C^2$, and (338) follows from (i) in Lemma 23, and (340) follows from the condition (266) in Lemma 23, which holds for $n$ sufficiently large. $\qquad\square$

Now, we return to prove Theorem 6.

*Proof of Theorem 6.* . Consider sequences $\delta_j := 2^{-j\frac{2}{\gamma}}$, and

$$\varepsilon_j := \left( \frac{1}{n\delta_j^{\alpha'}} \right)^{\frac{1}{2+\alpha'}}, \qquad j \geq 0, \tag{341}$$

where

$$\alpha' := \frac{2\gamma}{2-\gamma} \geq \alpha. \tag{342}$$

Then, we have

$$\varepsilon_j = n^{(\gamma-2)/4} 2^j. \tag{343}$$

Let

$$\mathcal{E} := \{ \exists j \geq 0 \quad \exists f \in \mathcal{F} : P_n\{f \leq \delta_j\} \wedge P\{f \leq \delta_j/2\} \geq \xi\varepsilon_j \}. \tag{344}$$

By Lemma 24, the condition (19) implies that there exists $\xi > 0$ such that

$$\mathbb{P}[\mathcal{E}] \leq 4/\eta \sum_{j=0}^{\infty} \left( \log_2 \log_2 \varepsilon_j^{-1} \right) \exp\left( -\frac{n\varepsilon_j^2}{2} \right) \tag{345}$$

$$\leq 4\upsilon' \sum_{j=0}^{\infty} \left( \log_2 \log_2 n \right) \sum_{j\geq 0} \exp\left[ -\frac{n^{\frac{\gamma}{2}}}{2} 2^{2j} \right] \tag{346}$$

$$\leq \upsilon \left( \log_2 \log_2 n \right) \exp\left[ -\frac{n^{\frac{\gamma}{2}}}{2} \right] \tag{347}$$

for some $\upsilon, \upsilon' > 0$. Now, since $\hat{\delta}_n(\gamma; f) \in (0, 1]$, there exists some $j \geq 1$ such that

$$\hat{\delta}_n(\gamma; f) \in (\delta_j, \delta_{j-1}]. \tag{348}$$

Then, by the definition of $\hat{\delta}_n(\gamma; f)$ in (18), we have

$$P_n\{f \leq \delta_j\} \leq P_n\{f \leq \hat{\delta}_n(\gamma; f)\} \tag{349}$$

$$\leq \sqrt{\delta_j^{-\gamma} n^{-1+\frac{\gamma}{2}}} \tag{350}$$

$$= \varepsilon_j. \tag{351}$$

Suppose that for some $f \in \mathcal{F}$, the inequality $\zeta^{-1}\hat{\delta}_n(\gamma; f) \leq \delta_n(\gamma; f)$ fails, which leads to

$$\delta_n(\gamma; f) < \zeta^{-1}\hat{\delta}_n(\gamma; f) \tag{352}$$

$$\leq \frac{\delta_{j-1}}{\zeta}. \tag{353}$$

Then, if $\zeta > 2^{1+\frac{2}{\gamma}}$, from the definition of $\delta_n(\gamma; f)$ in (17), it holds that

$$P\{f \leq \delta_j/2\} \geq P\left\{ f \leq \frac{\delta_{j-1}}{\zeta} \right\} \tag{354}$$

$$\geq \sqrt{\left( \frac{\delta_{j-1}}{\zeta} \right)^{-\gamma} n^{-1+\gamma/2}} \tag{355}$$

$$= \sqrt{\frac{1}{2} 2^{2j} \zeta^{\gamma} n^{-1+\frac{\gamma}{4}}} \tag{356}$$

$$= \varepsilon_j \sqrt{\frac{\zeta^{\gamma}}{4}} \tag{357}$$

$$> \xi\varepsilon_j \tag{358}$$

by choosing $\zeta$ sufficiently large, where $\xi$ is defined in Lemma 24. From (358), it holds that $\zeta^{-1}\hat{\delta}_n(\gamma; f) \leq \delta_n(\gamma; f)$ fails for some $f \in \mathcal{F}$ to hold with probability at most $\mathbb{P}[\mathcal{E}]$.

Similarly, we can show that the event $\delta_n(\gamma; f) \leq \zeta\hat{\delta}_n(\gamma; f)$ fails for some $f \in \mathcal{F}$ with probability at most $\mathbb{P}[\mathcal{E}]$.

By using the union bound, we finally obtain (20). $\qquad\square$

# E   Proof of Lemma 7

*Proof.* Observe that

$$\left|P_n(f \leq y) - P(f \leq y)\right|$$

$$= \frac{1}{n}\sum_{i=1}^{n}\left(\mathbf{1}\{f(X_i) \leq y\} - \mathbb{P}(f(X_i) \leq y)\right) + \frac{1}{n}\sum_{i=1}^{n}\mathbb{P}(f(X_i) \leq y) - P(f \leq y). \tag{359}$$

Now, let

$$f_n(x) := \mathbf{1}\{f(x) \leq y\} - \mathbb{P}(f(X_n) \leq y), \tag{360}$$

for all $x \in \mathcal{S}$ and $n \in \mathbb{Z}^+$. It is clear that

$$\|f_n\|_\infty \leq 1. \tag{361}$$

Now, let

$$g(\mathbf{x}) := \frac{1}{n}\sum_{i=1}^{n} f_i(x_i). \tag{362}$$

Then, we have

$$\left|g(\mathbf{x}) - g(\mathbf{y})\right| = \frac{1}{n}\left|\sum_{i=1}^{n}\left(f_i(x_i) - f_i(y_i)\right)\right| \tag{363}$$

$$\leq \frac{1}{n}\left|\sum_{i=1}^{n}\left(\mathbf{1}\{f(x_i) \leq y\} - \mathbf{1}\{f(y_i) \leq y\}\right)\right| \tag{364}$$

$$\leq \frac{1}{n}\sum_{i=1}^{n}\mathbf{1}\{x_i \neq y_i\}. \tag{365}$$

Then, by applying Lemma 21, it holds that

$$\mathbb{P}\left[\left|\frac{1}{n}\sum_{i=1}^{n} f_i(X_i)\right| \geq t\sqrt{\frac{\tau_{\min}}{n}}\right] \leq 2\exp\left(-2t^2\right). \tag{366}$$

On the other hand, for all $y \in \mathbb{R}$, let

$$\tilde{f}_y(x) := \mathbf{1}\{f(x) \leq y\}. \tag{367}$$

It is clear that $\sup_{y \in \mathbb{R}}\|\tilde{f}_y\|_\infty = 1$. Hence, by Lemma 1, it holds that

$$\left|\frac{1}{n}\sum_{i=1}^{n}\mathbb{P}(f(X_i) \leq y) - P(f \leq y)\right| = \left|\frac{1}{n}\sum_{i=1}^{n}\mathbb{E}[\tilde{f}_y(X_i)] - \mathbb{E}_\pi\left[\tilde{f}_y(X)\right]\right| \tag{368}$$

$$\leq \sqrt{B_n}, \tag{369}$$

where (369) follows from Lemma 20 (with $M = 1$) and the Cauchy–Schwarz inequality.

From (359), (366), and (369), we have

$$\sup_{f \in \mathcal{F}}\sup_{y \in \mathbb{R}}\left|P_n(f \leq y) - P(f \leq y)\right| \leq \sqrt{B_n} + t\sqrt{\frac{\tau_{\min}}{n}} \tag{370}$$

with probability at least $1 - 2\exp(-2t^2)$. $\qquad\square$

# F   Proof of Lemma 9

Let $\delta > 0$. Let $\varphi(x)$ be equal to 1 for $x \leq 0$, 0 for $x \geq 1$ and linear in between. Observe that

$$F_f(y) = P\{f \leq y\} \tag{371}$$

$$\leq P\varphi\left(\frac{f - y}{\delta}\right) \tag{372}$$

$$\leq P_n\varphi\left(\frac{f - y}{\delta}\right) + \|P_n - P\|_{\tilde{\mathcal{G}}_\varphi} \tag{373}$$

$$\leq F_{n,f}(y + \delta) + \|P_n - P\|_{\tilde{\mathcal{G}}_\varphi}, \tag{374}$$

and

$$F_{n,f}(y) \leq P_n\{f \leq y\} \tag{375}$$

$$\leq P_n\varphi\left(\frac{f-y}{\delta}\right) \tag{376}$$

$$\leq P\varphi\left(\frac{f-y}{\delta}\right) + \|P_n - P\|_{\tilde{\mathcal{G}}_\varphi} \tag{377}$$

$$\leq F_f(y+\delta) + \|P_n - P\|_{\tilde{\mathcal{G}}_\varphi}. \tag{378}$$

Now, by applying Lemma 21 (see (197)), we have

$$\mathbb{P}\left[\|P_n - P\|_{\tilde{\mathcal{G}}_\varphi} \geq \mathbb{E}\left[\|P_n - P\|_{\tilde{\mathcal{G}}_\varphi}\right] + t\sqrt{\frac{\tau_{\min}}{n}}\right] \leq 2\exp\left(-2t^2\right). \tag{379}$$

From (379), with probability at least $1 - 2\exp(-2t^2)$, it holds that

$$\|P_n - P\|_{\tilde{\mathcal{G}}_\varphi} \leq \mathbb{E}\left[\|P_n - P\|_{\tilde{\mathcal{G}}_\varphi}\right] + t\sqrt{\frac{\tau_{\min}}{n}}. \tag{380}$$

On the other hand, from Lemma 1, we have

$$\mathbb{E}\left[\|P_n - P\|_{\tilde{\mathcal{G}}_\varphi}\right] \leq 2\mathbb{E}\left[\|P_n^0\|_{\tilde{\mathcal{G}}_\varphi}\right] + B_n. \tag{381}$$

From (380) and (381), with probability at least $1 - 2\exp(-2t^2)$, it holds that

$$\|P_n - P\|_{\tilde{\mathcal{G}}_\varphi} \leq 2\mathbb{E}\left[\|P_n^0\|_{\tilde{\mathcal{G}}_\varphi}\right] + B_n + t\sqrt{\frac{\tau_{\min}}{n}}. \tag{382}$$

From (374), (378), and (382), with probability at least $1 - 2\exp(-2t^2)$, we have

$$L(F_f, F_{f,n}) \leq \delta + 2\mathbb{E}\left[\|P_n^0\|_{\tilde{\mathcal{G}}_\varphi}\right] + B_n + t\sqrt{\frac{\tau_{\min}}{n}}. \tag{383}$$

Furthermore, by Talagrand's contraction lemma [3, 11] for the class of function $\tilde{\varphi}(x) := \varphi(x) - 1$, we have

$$\mathbb{E}\left[\|P_n^0\|_{\tilde{\mathcal{G}}_\varphi}\right] \leq 2\mathbb{E}\left[\sup_{f\in\mathcal{F}, y\in[-M,M]}\left|\sum_{i=1}^n \varepsilon_i \frac{f(X_i) - y}{\delta}\right|\right] \tag{384}$$

$$= \frac{2}{\delta}\mathbb{E}\left[n^{-1}\sup_{f\in\mathcal{F}}\left|\sum_{i=1}^n f(X_i)\right|\right] + \frac{2M}{\delta n}\mathbb{E}\left|\sum_{i=1}^n \varepsilon_i\right| \tag{385}$$

$$\leq \frac{2}{\delta}\mathbb{E}\left[\|P_n^0\|_{\mathcal{F}}\right] + \frac{2M}{\delta\sqrt{n}}. \tag{386}$$

Hence, by setting $\delta := \sqrt{4\mathbb{E}[\|P_n^0\|_{\mathcal{F}}] + 4M/\sqrt{n}}$, from (383) and (386), it holds with probability at least $1 - 2\exp(-2t^2)$ that

$$L(F_f, F_{f,n}) \leq 4\sqrt{\mathbb{E}[\|P_n^0\|_{\mathcal{F}}] + M/\sqrt{n}} + B_n + t\sqrt{\frac{\tau_{\min}}{n}}. \tag{387}$$

## G  Proof of Theorem 10

Fix $M > 0$. Since $\mathcal{F}_M \in \mathrm{GC(P)}$, we have

$$\mathbb{E}\left[\|P_n - P\|_{\mathcal{F}_M}\right] \to 0 \quad \text{a.s.} \quad n \to \infty, \tag{388}$$

which, by Lemma 1 with $t = \sqrt{\log n}$,

$$\mathbb{E}\left[\|P_n - P\|_{\mathcal{F}_M}\right] \geq \frac{1}{2}\mathbb{E}\left[\|P_n^0\|_{\mathcal{F}_M}\right] - \tilde{A}_n. \tag{389}$$

By taking $n \to \infty$, from (389), we obtain

$$\mathbb{E}\big[\|P_n^0\|_{\mathcal{F}_M}\big] \to 0, \tag{390}$$

or

$$\mathbb{E}\bigg[\Big\|n^{-1}\sum_{i=1}^{n}\varepsilon_i\delta_{X_i}\Big\|\bigg]_{\mathcal{F}_M} \to 0, \quad \text{as} \quad n \to \infty. \tag{391}$$

Furthermore, with $t = \sqrt{\log n}$, by Lemma 9, we have

$$\mathbb{P}\bigg\{\sup_{f \in \mathcal{F}_M} L(F_{n,f}, F_f) \ge 4\sqrt{\mathbb{E}[\|P_n^0\|_{\mathcal{F}}] + M/\sqrt{n}} + B_n + \log n\sqrt{\frac{\tau_{\min}}{n}}\bigg\}$$

$$\le 2\exp\big(-2\log n\big) = \frac{2}{n^2}. \tag{392}$$

It follows from (392) that

$$\sum_{n=1}^{\infty}\mathbb{P}\bigg\{\sup_{f \in \mathcal{F}_M} L(F_{n,f}, F_f) \ge 4\sqrt{\mathbb{E}[\|P_n^0\|_{\mathcal{F}}] + M/\sqrt{n}} + B_n + \log n\sqrt{\frac{\tau_{\min}}{n}}\bigg\}$$

$$\le 2\sum_{n=1}^{\infty}\frac{1}{n^2} < \infty. \tag{393}$$

Hence, by Borel-Cantelli's lemma [13], $B_n \to 0$, and (391), we obtain

$$\sup_{f \in \mathcal{F}_M} L(F_{n,f}, F_f) \to 0, \qquad a.s.. \tag{394}$$

Since $\sup_{f \in \mathcal{F}} L(F_{n,f_M}, F_{f_M}) = \sup_{f \in \mathcal{F}_M} L(F_{n,f}, F_f)$, from (394), we have

$$\sup_{f \in \mathcal{F}} L(F_{n,f_M}, F_{f_M}) = \sup_{f \in \mathcal{F}_M} L(F_{n,f}, F_f) \to 0, \qquad a.s.. \tag{395}$$

Now, by [2], the following facts about Levy's distance holds:

$$\sup_{f \in \mathcal{F}} L(F_f, F_{f_M}) \le \sup_{f \in \mathcal{F}} P\{|f| \ge M\} \tag{396}$$

and

$$\sup_{f \in \mathcal{F}} L(F_{n,f}, F_{n,f_M}) \le \sup_{f \in \mathcal{F}} P_n\{|f| \ge M\}. \tag{397}$$

Now, by the condition (29), we have

$$\sup_{f \in \mathcal{F}} P\{|f| \ge M\} \to 0 \qquad a.s. \qquad M \to \infty, \tag{398}$$

so

$$\sup_{f \in \mathcal{F}} L(F_f, F_{f_M}) \to 0, \qquad a.s. \qquad M \to \infty. \tag{399}$$

To prove that

$$\lim_{M \to \infty}\limsup_{n \to \infty}\sup_{f \in \mathcal{F}} L(F_{n,f}, F_{n,f_M}) = 0, \qquad a.s., \tag{400}$$

it is enough to show that

$$\lim_{M \to \infty}\limsup_{n \to \infty}\sup_{f \in \mathcal{F}} P_n\{|f| \ge M\} = 0, \qquad a.s. \tag{401}$$

To this end, consider the function $\varphi$ from $\mathbb{R}$ into $[0,1]$ that is equal to 0 for $|u| \le M-1$, is equal to 1 for $|u| > M$ and is linear in between. We have

$$\sup_{f \in \mathcal{F}} P_n\{|f| \ge M\} = \sup_{f \in \mathcal{F}_M} P_n\{|f| \ge M\} \tag{402}$$

$$\le \sup_{f \in \mathcal{F}_M} P_n\varphi(|f|) \tag{403}$$

$$\le \sup_{f \in \mathcal{F}_M} P\varphi(|f|) + \|P_n - P\|_{\mathcal{G}} \tag{404}$$

$$\le \sup_{f \in \mathcal{F}_M} P\{|f| \ge M-1\} + \|P_n - P\|_{\mathcal{G}}, \tag{405}$$

where

$$\mathcal{G} := \big\{\varphi \circ |f| : f \in \mathcal{F}_M\big\}. \tag{406}$$

Then, by using the same arguments to obtain (197), it holds with probability $1 - 2\exp(-2t^2)$ that

$$\|P_n - P\|_{\mathcal{G}} \leq \mathbb{E}\big[\|P_n - P\|_{\mathcal{G}}\big] + t\sqrt{\frac{\tau_{\min}}{n}} \tag{407}$$

$$\leq 2\mathbb{E}[\|P_n^0\|_{\mathcal{F}}] + A_n + t\sqrt{\frac{\tau_{\min}}{n}}. \tag{408}$$

Then, by setting $t = \log n$ and using the Borel-Cantelli's lemma [13], the following holds almost surely:

$$\|P_n - P\|_{\mathcal{G}} \leq 2\mathbb{E}\big[\|P_n^0\|_{\mathcal{G}}\big] + A_n + \log n\sqrt{\frac{\tau_{\min}}{n}}. \tag{409}$$

Now, since $\varphi \circ f \in \varphi \circ \mathcal{F}_M$, by (391) and Talagrand contraction lemma [3, 12], we have

$$\mathbb{E}\big[\|P_n^0\|_{\mathcal{G}}\big] \to 0 \qquad \text{as} \quad n \to \infty. \tag{410}$$

From (409) and (410), we obtain

$$\|P_n - P\|_{\mathcal{G}} \to 0 \quad a.s.. \tag{411}$$

Hence, we obtain (ii) from (i), the condition (29), and (411).

To prove that (ii) implies (i), we use the following bound [14]

$$\left| \int_{-M}^{M} t \, d(F - G)(t) \right| \leq \upsilon L(F, G), \tag{412}$$

which holds with some constant $\upsilon = \upsilon(M)$ for any two distribution functions on $[-M, M]$. This bound implies that

$$\|P_n - P\|_{\mathcal{F}_M} = \sup_{f \in \mathcal{F}_M} |P_n f - P f| \tag{413}$$

$$\leq \sup_{f \in \mathcal{F}_M} \left| \int_{-M}^{M} t \, d(F_{n,f} - F_f)(t) \right| + M \sup_{f \in \mathcal{F}_M} \big|P(|f| \geq M) - P_n(|f| \geq M)\big| \tag{414}$$

$$\leq \upsilon \sup_{f \in \mathcal{F}_M} L(F_{n,f}, F_f) + M \sup_{f \in \mathcal{F}_M} \big|P(|f| \geq M) - P_n(|f| \geq M)\big| \tag{415}$$

$$\leq \upsilon \sup_{f \in \mathcal{F}} L(F_{n,f}, F_f) + M \sup_{f \in \mathcal{F}_M} \big|P(|f| \geq M) - P_n(|f| \geq M)\big|. \tag{416}$$

Now, by Lemma 7, with probability at least $1 - 2\exp(-2t^2)$, the following holds:

$$\sup_{y \in \mathbb{R}} \sup_{f \in \mathcal{F}_M} \big|P_n(f \leq y) - P(f \leq y)\big| \leq t\sqrt{\frac{\tau_{\min}}{n}} + \sqrt{B_n}. \tag{417}$$

By setting $t = \sqrt{\log n}$ and using the Borel-Cantelli's lemma, from (417), we obtain

$$\sup_{y \in \mathbb{R}} \sup_{f \in \mathcal{F}_M} \big|P_n(f \leq y) - P(f \leq y)\big| \to 0, \quad a.s.. \tag{418}$$

Finally, from (417), (418), and (ii), we obtain (i). This concludes our proof of Theorem 10.

## H    Proof of Theorem 12

The proof is based on [2, Proof of Theorem 9]. Since $\mathcal{F}$ is uniformly bounded, we can choose $M > 0$ such that $\mathcal{F}_M = \mathcal{F}$. To prove the first statement, note that $\mathcal{F} \in \mathrm{BCLT}(\mathrm{P})$ means that

$$\mathbb{E}\big[\|P_n - P\|_{\mathcal{F}}\big] = O(n^{-1/2}). \tag{419}$$

Now, from Lemma 1, we have

$$\mathbb{E}\big[\big\|P_n - P\big\|_{\mathcal{F}}\big] \geq \frac{1}{2}\mathbb{E}\big[\|P_n^0\|_{\mathcal{F}}\big] - \tilde{A}_n, \tag{420}$$

for all $t > 0$. By applying (420), it easy to see that

$$\mathbb{E}\bigg[\bigg\|n^{-1}\sum_{i=1}^{n}\varepsilon_i\delta_{X_i}\bigg\|_{\mathcal{F}}\bigg] = \mathbb{E}\big[\|P_n^0\|_{\mathcal{F}}\big] \tag{421}$$

$$\leq 2\tilde{A}_n + 2\mathbb{E}\big[\big\|P_n - P\big\|_{\mathcal{F}}\big] \tag{422}$$

$$\leq O\bigg(\sqrt{\frac{\log n}{n}}\bigg) \tag{423}$$

since $\tilde{A}_n = O\big(\sqrt{\frac{\log n}{n}}\big)$ by (1).

Now, from Lemma 9, for $t = \sqrt{\log n}$,

$$\mathbb{P}\bigg\{\sup_{f\in\mathcal{F}} L(F_f, F_{f,n}) \geq 4\sqrt{\mathbb{E}[\|P_n^0\|_{\mathcal{F}}] + M/\sqrt{n}} + B_n + \sqrt{\frac{\tau_{\min}\log n}{n}}\bigg\} \leq \frac{2}{n^2}. \tag{424}$$

From (423) and (424), it holds that

$$\mathbb{P}\bigg\{\bigg(\frac{n}{\log n}\bigg)^{1/4}\sup_{f\in\mathcal{F}} L(F_{n,f}, F_f) \geq D\bigg\} \to 0 \tag{425}$$

as $n \to \infty$ for some constant $D$, or

$$\sup_{f\in\mathcal{F}} L(F_{n,f}, F_f) = O_P\bigg(\bigg(\frac{\log n}{n}\bigg)^{1/4}\bigg). \tag{426}$$

Now, recall

$$\tilde{\mathcal{G}}_\varphi := \bigg\{\varphi\circ\bigg(\frac{f-y}{\delta}\bigg) - 1 : f\in\mathcal{F}, y\in[-M, M]\bigg\}. \tag{427}$$

To prove the second statement, we use the following fact [2, p.29]:

$$\mathbb{E}_\varepsilon\big[\|P_n^0\|_{\tilde{\mathcal{G}}_\varphi}\big] \leq \frac{d}{\sqrt{n}}\bigg[\int_0^{\sqrt{2}} H_{d_{P_n,2}}^{1/2}(\mathcal{F};\delta u)du + \sqrt{\log\frac{4M}{\delta}} + 1\bigg] \tag{428}$$

for some constant $d$, which, under the condition (32), satisfies

$$\mathbb{E}_\varepsilon\big[\|P_n^0\|_{\tilde{\mathcal{G}}_\varphi}\big] \leq d\bigg[\frac{1}{\delta^{\alpha/2}}\sqrt{\frac{1}{n}} + \frac{1}{\sqrt{n}}\bigg(\sqrt{\log\frac{4M}{\delta}} + 1\bigg)\bigg]. \tag{429}$$

Now, by Lemma 9, it holds for all $t > 0$ and $\delta > 0$ that

$$\mathbb{P}\bigg\{\sup_{f\in\mathcal{F}} L(F_f, F_{f,n}) \geq \delta + \mathbb{E}\big[\|P_n^0\|_{\tilde{\mathcal{G}}_\varphi}\big] + B_n + t\sqrt{\frac{\tau_{\min}}{n}}\bigg\} \leq 2\exp(-2t^2). \tag{430}$$

Since $\mathbb{E}\big\|n^{-1}\sum_{i=1}^{n}\varepsilon_i\delta_{X_i}\big\|_{\tilde{\mathcal{G}}_\varphi} = \mathbb{E}[\|P_n^0\|_{\tilde{\mathcal{G}}_\varphi}] \leq d\big[\sqrt{\frac{\log n}{n}}\delta^{-\alpha/2} + \frac{1}{\sqrt{n}}\big(\sqrt{\log\frac{4M}{\delta}} + 1\big)\big]$, from (430), for all $t > 0$, we have

$$\mathbb{P}\bigg\{\sup_{f\in\mathcal{F}} L(F_f, F_{f,n}) \geq \delta + d\bigg[\sqrt{\frac{\log n}{n}}\delta^{-\alpha/2} + \frac{1}{\sqrt{n}}\bigg(\sqrt{\log\frac{4M}{\delta}} + 1\bigg)\bigg] + B_n + t\sqrt{\frac{\tau_{\min}}{n}}\bigg\}$$
$$\leq 2\exp\big(-2t^2\big). \tag{431}$$

Now, by choosing $t = \sqrt{\log n}$ and $\delta = (\log n)n^{-\frac{1}{2+\alpha}}$, we have

$$\mathbb{P}\bigg\{\sup_{f\in\mathcal{F}} L(F_f, F_{f,n}) \geq \nu(\log n)n^{-\frac{1}{2+\alpha}}\bigg\} \leq \frac{2}{n^2} \tag{432}$$

for some constant $\nu$ and for $n \geq N_0$ for some finite $N_0$ big enough.

From (432), we have

$$\sum_{n=1}^{\infty} \mathbb{P}\left\{ \sup_{f \in \mathcal{F}} L(F_f, F_{f,n}) \geq \nu(\log n) n^{-\frac{1}{2+\alpha}} \right\} \leq N_0 + \sum_{n=N_0}^{\infty} \frac{2}{n^2} < \infty. \qquad (433)$$

Hence, by Borel-Cantelli's lemma [13], it holds that

$$\sup_{f \in \mathcal{F}} L(F_f, F_{f,n}) = O_P\big((\log n) n^{-\frac{1}{2+\alpha}}\big), \qquad a.s. \qquad (434)$$

This concludes our proof of Theorem 12.