# OpenReview forum: "Generalization Error Bounds on Deep Learning with Markov Datasets"
_NeurIPS.cc/2022/Conference — NeurIPS 2022 Accept_

### Official Review · Reviewer_MmSs · 2022-07-07

**Rating:** 3
**Confidence:** 4
**Soundness:** 3 good
**Presentation:** 1 poor
**Contribution:** 2 fair

**Summary:**

This work uses recent high dimensional symmetrization inequalities to extend existing i.i.d. generalization bound approaches to prove new generalization error bounds for machine learning on Markov datasets. An approach is proposed to extend generalization bounds to Bayesian settings.

**Questions:**

none

**Limitations:**

While the authors motivate the problem by observing that many real world datasets consist of time series data instead of i.i.d. samples, the results are severely limited by assuming stationarity of the Markov chain. In fact, to my knowledge, none of the motivating applications listed in the introduction typically statisfy this property. I would have liked to see a better and more complete motivation for considering stationary Markov chains from the machine learning literature.

**Strengths And Weaknesses:**

Strength: The paper addresses generalization bounds in the setting where data is generated by a Markov chain, a setting which to my knowledge has not previously been treated in the literature.

Weaknesses:
Stationary assumption: While the authors motivate the problem by observing that many real world datasets consist of time series data instead of i.i.d. samples, the results are severely limited by assuming stationarity of the Markov chain. In fact, to my knowledge, none of the motivating applications listed in the introduction typically statisfy this property. I would have liked to see a better and more complete motivation for considering stationary Markov chains from the machine learning literature.

Presentation:
Section 4 appears to be unfinished, without even a theorem statement - the way it points to a corresponding vastly expanded section in the supplement is troubling as it appears to be an attempt to bypass the main paper page limit. Given the higher-order results are critical to the contribution of the paper, Section 3 should have been greatly condensed so Section 4 could be expanded and fit the page limit. Furthermore, the supplement is not written as such, it almost looks as if that is the real main paper and the "main paper" is just a haphazard and inefficient extraction of portions of it - see the many sections that are present in both almost verbatim. The supplement should instead be essentially a collection of appendices well-tied to the main paper.

Minor:
Line 15: [27] is a book, not "seminar papers"

Line 47 should be clarified: [29] did not "introduce Bayesian deep learning", see citations in [29] for instance.

Line 156: Presumably there are l layers of neurons, not "V".

---------------------------------------
Edit after rebuttal: Unfortunately the authors do not acknowledge my significant concerns with the presentation and in general do not promise or implement any changes to the paper. I cannot recommend acceptance of this work as-is.

---

> ### Author Response · Authors · 2022-07-28
> **Reply to the reviewer's review**
>
> Thank you very much for reviewing our paper.  However, your concern related to the stationary assumption looks not totally correct.
>
> In our opinion, the Markov model is used in a vast application in machine learning. For example, it models time-series data in machine learning, or hidden Markov models (a special type of Markov chains) are widely used in machine learning for speech recognition, pattern recognition, handwriting, etc as we mentioned in the introduction part and in our results.
>
> You are right in the sense that some real-world datasets (but not all) are not exactly stationary Markov chains. However,  in practice, in many machine learning applications, datasets are samples of time series achieved by using MCMC modeling (which generates stationary Markov chains). Hence, we can approximate time-series datasets as stationary Markov chains in many applications.  There are also some other methods of approximating non-stationary Markov chains by stationary ones by using AR and ARMA models in the statistical research literature.  Our work can be considered the first step toward real-world problems. Any progress in a challenging problem should be acknowledged.  The second reviewer gives us some published papers related to the Markov chain in machine learning.  You can refer to these papers as well. We also note that the i.i.d. dataset (which is currently assumed in most machine learning research literature) is a special case of the Markov dataset with the stationary distribution. Hence, your current decision is totally not correct.
>
> Thank you very much for your suggestions related to the presentation. We will change our presentations to your suggestions. More specifically, we will compress Section 3 (put some results in the appendices) and add more details about the high-order Markov in Section 4. We also remove redundant statements in the Appendix which are already stated in the main section and correct three minor points as your suggestions. We will also make the motivation clearer in the introduction based on your concerns. We admit that our presentation is not perfect, but it is readable by other researchers.  We don't see any serious presentation problems from your analysis that we cannot fix. Removing the restatements of some theorems and lemmas in the Appendix since they are already stated in the main part makes the paper more beautiful, but it is not a serious presentation error, and it can easily be fixed.

---

### Official Review · Reviewer_pV3G · 2022-07-12

**Rating:** 7
**Confidence:** 2
**Soundness:** 3 good
**Presentation:** 2 fair
**Contribution:** 3 good

**Summary:**

The submission extends Rademacher and Gaussian complexity generalization bounds from iid data to Markov Chains. The resulting bounds for neural networks are also provided and then extended to Bayesian neural networks. Extensions to m-order Markov Chains and mixtures of Markov chains are given in the appendix.

**Questions:**

See the previous section

**Limitations:**

See the weaknesses section

**Strengths And Weaknesses:**

I have read other papers and worked on theory of Markov Chains but still had a hard time following and verifying the results. I am voting for rejection but this is a low-confidence review as I might have misunderstood the terminology and notation. The techniques appear general and, if correct, could help for extending results other than Rademacher complexity to Markov Chains. The current manuscript is however hard to follow and it will benefit from reorganization and other revisions that I will elaborate on below.

1. In line 74, what does $\nu << \pi$ mean for start distribution $\nu$ and stationary distribution $\pi$? If this is a comparison, how are these distributions compared? In Eq (5) below it, what is $dv/d\pi$? $v$ is not defined. If this is $\nu$, does this mean that the start distribution is a function of the stationary distribution?

2. I think the paper should shift its focus from neural networks to a general machine learning model. The title and introduction emphasize the generalization bound on deep learning yet the main result (Theorem 2) is for a general model. The extension to deep learning (Theorem 3) is in fact a corollary of the previous theorem. In the iid setting there is a vast literature on measures of complexity special to neural networks which the authors can follow-up on if they want to focus on deep learning. A good starting point is [1] although there has been a substantial amount of work after this review.

3. One of the paper's main contributions is extensions beyond simple Markov Chain models (Section 4). This section has only one paragraph in the main paper and all the results are moved to the appendix. The required space for bringing these results in the main paper can be freed up by making some of the short Equations (7) up to (17) in Section 2 inline and reducing the whitespace.

4. There are other techniques e.g. [2] and Section 3 in [3] for extending results from iid to Markov Chain using mixing time. While this previous work is not about Rademacher complexity and generalization I wonder in what ways the underlying techniques are similar or different from the submission.

Minor comments:

(i) I suggest defining "Polish space" in line 55 and g* in line 60

(ii) The formulation of neural network in Section 3.2 is a bit confusing. See [1] for an example of a more standard notation.


[1] Behnam Neyshabur, "Implicit Regularization in Deep Learning." PhD thesis, 2017

[2] Duchi, John C., et al. "Ergodic mirror descent." SIAM Journal on Optimization (2012)

[3] Wang, Gang, Bingcong Li, and Georgios B. Giannakis. "A multistep Lyapunov approach for finite-time analysis of biased stochastic approximation." arXiv (2019).

-------------------------------------------------------
Update after discussions:

Thanks for the clarifications. I could go through the main result and its proof in the appendix after these clarifications and changed my vote to accept. The initial score was tentative.

I still strongly believe the paper should assume less about the reader's background in this part of measure theory. Perhaps the current definitions are adequate for a paper in a statistics journal, but a large part of NeurIPS audience whose research is related to generalization and can benefit from this result will have trouble following this paper. Even more fundamental measure theoretic concepts are often named or fully defined in machine learning papers. This is something that can be fixed in camera-ready and I do not want to lower my score because of it but it worries me that the initial rebuttal (before update) insisted on not incorporating this change and no revision has been submitted so far.

The stationary assumption is fine to me although I do share reviewer MmSs's concern about organization.

---

> ### Author Response · Authors · 2022-07-28
> **Reply to the reviewer's review**
>
> Thank you very much for spending time reviewing our paper. Although there are some suggestions that can help us to improve our paper, however, your decision is not correct. The following are our answers to your questions.
>
> 1. The first definition $\nu<<\pi$ and $d\nu/ d \pi$ are well-known concepts in statistics. Hence, we think that giving those definitions is optional. These concepts refer to absolute continuous and Radon-Nikodym derivatives, not the simple comparison between two numbers. Based on your concerns, we will define them in the revised version.
>
> 2.  We agree that the title of the paper can shift from neural networks to general machine learning. We would like to focus on deriving generalization bounds for deep neural networks, but indeed, our results can apply to general machine learning as you mentioned.
>
> 3.  Our main focus is the $1$-order Markov chain. In Section IV, we mentioned how to extend to the $m$-order Markov chain.
> We will put some equations in Section 3 to Appendix and extend this section as per your suggestions.
>
> 4. The papers that you suggest mainly focus on designing specific learning algorithms. In our paper, we show that we can guarantee generalization errors for classes of learning algorithms. However, we will cite these papers since they also studied the Markov chain.
>
> From our answers to your questions, the reason to reject our paper looks not clear.  We admit that our presentation is not perfect, but it is readable by other researchers. We don't see any serious presentation problems from your analysis that we cannot fix. Expanding some parts and reducing the others is not a serious problem.

---

### Official Review · Reviewer_XGoR · 2022-07-12

**Rating:** 8
**Confidence:** 4
**Soundness:** 4 excellent
**Presentation:** 3 good
**Contribution:** 4 excellent

**Summary:**

This paper develops an empirical process theory for irreducible reversible Markov Chains with applications to statistical learning theory for non i.i.d. sampling. It adopts an ambitious and historically faithful approach, following the classic [1] quite closely and generalizing most of their results to the non i.i.d. case.

In very simple terms, the most basic idea relevant to both this paper and to [2] is that empirical averages of functions will converge to their population versions, where the latter are defined in terms of the stationary distribution of the background Markov chain process, as a result of the convergence of the samples to this stationary distribution.

However, the results are of course finite sample and in particular, also apply before approaching the stationary distribution and perfectly quantify the convergence. The proofs usually follow the proofs of [1] quite closely, but must also invoke in passing some powerful results  from [2,3,4,5]. Most notably this includes Lemma 22 from the sup., i.e. Theorem 3.41 from [3], which quantifies the convergence of the background Markov process to the stationary distribution (which constitutes one more source of "failure"), as well as the Markov Hoeffding's lemma (lemma 23, page 27) from [2,6]

The work is composed of the main paper and a supplementary in two parts since the latter contains an appendix. The 2-part supplementary is reasonably self-contained and there is little need to read the main paper in too much detail.
 In my opinion, one of the main results of the paper is Theorem 5 in the supplementary, which provides a uniform high probability bound on the difference between P(f_n\leq 0) and P(f\leq 0) where f_n is an empirical average of f and the bound is uniform over f running in an arbitrary function class. The bound naturally involves the Rademacher complexity of the function class in question. In the main paper and much of the rest of the "main supplementary", the results are expressed in a more general but harder-to-parse form (which can nonetheless be easily translated to theorems of the form of this Theorem 5. )

The above result can be directly applied to obtain generalization bounds for classification problems via the classic lemma in statistical learning theory, for an arbitrary method, as long as one can bound the complexity of the underlying function class. However, "the main supplementary" goes much further and (in section 2.3) establishes tighter results on the convergence of the empirical processes not just in distribution but in terms of Lévy distance. The work also provides applications of the results to convex combinations of classifiers and to neural networks, all of which mirroring [1]. The application to Neural Networks is reasonably high-level (could theoretically be applied to other methods such as deep kernels for instance), and features the complexity of the one-layer set of basis functions without computing it.

Finally, there are sections with an extension of the results to the case of higher-order Markov chains and to Bayesian deep learning for good measure.





=======================================
References:


[1] V. Koltchinskii and D. Pachenko. "Empirical margin distributions and bounding the generalization error of combined classifiers", Annals of stats 2002.

[2] Jianqing Fan, Bai Jiang, Qiang Sun. "Hoeffding's inequality for general Markov chains and applications to statistical learning.", JMLR 2021

[3] Daniel Paulin. "Concentration inequalities for Markov chains by Marton Couplings and spectral methods", electronic journal of Probability 2015.

[4] Ledoux and Talagrand. Probability in Banach spaces.

[5] D. Rudolf. Explicit error bounds for Markov chain Monte Carlo. (arxiv 2011)//Explicit error bounds for lazy reversible Markov chain Monte Carlo,  Journal of Complexity 2009.

[6] Shravas Rao. A Hoeffding inequality for Markov Chains.  Electronic communications in Probability 2019.




**Questions:**

In this section, I list both "questions" and the most significant doubts/mistakes I have about the actual mathematics of the paper.  In the "limitations" section I list comparatively minor issues. (This should hopefully make it easier for the authors to prioritize their work during the rebuttal)


1. QUESTION: In Lemma 26 (main supplementary), your constants are explicit, whereas the equivalent result (Theorem 5 on page 16 of [1]) involves a nonexplicit constant. It would thus appear that your result is tighter even in the i.i.d. case. Is that the case and can you explain what made this possible?


2. PRESUMED ERROR: Lemma 22 page 25 is cited from 11, theorem 3.41, which is on page 60 and originally involves a parameter "p". The value of p which yields the result as cited is p=4, but then the constant next to M^2 should be 128 instead of 64, to account for the additional factor $\frac{p}{p-2}=2$. Of course this also influences several theorem statements throughout all three parts of the submission, including the definition of "A_n" wherever it appears, equation (18) in the main text, etc.



3.  QUESTION/ PRESUMED ERROR: I think there are a couple of issues with the proof of Lemma 7 (from page 5) on page 39. It doesn't seem that the statement is correctly proved with the current order between the classifiers. It looks like there is a single proof which works for an arbitrary value of y, but that is not enough to put y inside the high probability statement (unless a suitable union bound and limiting argument were applied). In addition, the proof is a bit sloppy on lines 636-638: It would seem that equation (400) holds "for any y" (which should replace the "for any t" from line 636), and that it shouldn't involve the term $\frac{4t}{\sqrt{n}}$, which instead comes from equation (394). It is after this (at equation (401)) that the quantifier is slipped back into the high probability statement without proof.

3.2 (Minor) For the ease of the reader I would also recommend adding that equation (214) is on page 25, or better, restructure the proof of Lemma 1 so that equation (214) becomes a separate lemma, which it seems to deserve to be.


4. MISSING PROOF (confirm solution below): The proof of Theorem 5 (page 4 of the main sup.) is not complete at all. Note that first of all, the theorem makes use of the notation "\gamma_{ps}", which has not been introduced in the paper. It turns out (see equation (3.3) of [3]) that this refers to the "pseudo spectral gap". Applying the results from Theorem 3 as dictated in the proof here does not yield equation (17) but instead a different equation with the term 4M\exp(...) replaced by something similar to the quantity "A_n". I believe that the correct way to reach theorem 5 is by using the Bernstein-type inequality from page 13 of [3] instead of Lemma 22 in dealing with the "burn-in" part of failure estimation.




5. QUESTION: Do you think it is possible to obtain simpler results of a more "classic SLT" flavor (with expectation instead of probability, and generalization bounds involving margins), analogous to the treatment of [7], by using only the symmetrization parts of the present paper and using a McDiarmid type inequality? If so, has it been done and where is the "Markov McDiarmid inequality" in the literature?

5.2 Have you tried just starting with the Hoeffding type lemma from the closely related work [2], applying a union bound (which should reach a finite class result already) and then using classic covering number arguments to obtain a result in terms of Theorem 10.1 in [8]?




=======================================
References:

[1] V. Koltchinskii and D. Pachenko. "Empirical margin distributions and bounding the generalization error of combined classifiers", Annals of stats 2002.

[2] Jianqing Fan, Bai Jiang, Qiang Sun. "Hoeffding's inequality for general Markov chains and applications to statistical learning.", JMLR 2021

[3] Daniel Paulin. "Concentration inequalities for Markov chains by Marton Couplings and spectral methods", electronic journal of Probability 2015.

[4] Ledoux and Talagrand. Probability in Banach spaces.

[5] D. Rudolf. Explicit error bounds for Markov chain Monte Carlo. (arxiv 2011)//Explicit error bounds for lazy reversible Markov chain Monte Carlo,  Journal of Complexity 2009.

[6] Shravas Rao. A Hoeffding inequality for Markov Chains.  Electronic communications in Probability 2019.

[7] Bartlett and Mendelson. "Rademacher and Gaussian Complexities: Risk Bounds and Structural Results" 2002.

[8] Anthony and Bartlett. Neural Network Learning: Theoretical foundations. 1999.


**Limitations:**

For actual limitations, see the above questions.  See more minor issues below.



==============Minor problems with notation=============


MAIN:

Equation (4) of the main paper (page 2): the mixing time is not defined (see def 1.3 on page 4 of [2]).
Line 72: \gamma_{ps} is not defined.  See (4.5) in [9], or middle of page 2 in [2].
Lines 74 to 76 are a little bit sloppy: the condition "\nu\ll \pi$ should be in equation (5) inside the brackets rather than before the equation, \nu and v shouldn't be used interchangeably and a clash with the running variable measure in eq (5) and the initial distribution \nu should be avoided. Later in the equation after line 76, it is the initial distribution that is meant by \nu/ v. There is also an actual constant "v" in Theorem 6 (sup) which should ideally not clash either.

Line 105 when you claim that the MC (X_n,Y_n) is irreducible and recurrent, I guess some more conditions are required, including the assumption that the original MC is recurrent and irreducible (which I agree is implied) and the fact that g(x,y) \neq 0 for all (x,y). Sounds like this condition should creep up in a lot of other main theorems in some form of other as well.

Equation (19), at the bottom of page 4 (line 130). P_0 is not defined! This is definitely unpleasant for the reader. I guess P_0 refers to the Rademacher or gaussian complexity, from the rest of the context.  It appears in technical appendices for the first time in equation (221) page 26 as far as I am concerned.

Equation (49), top of page 9: I think there is a constant "c" missing after \pi^2 before the exponential.

SUP

Lemma 19 and its proof (pages 18 and 19): there is some inconsistency between the originally introduced notation where bold version of epsilon, X and Y refer to the vector of the first n instances of the corresponding variables and the later notation. In the proof there is some notation of the type \epsilon^1_n for the first n instances of \epsilon (and similar notation for X and Y). This should be introduced.

equation (154), line 414 page 20: there is some inconsistency in the use of comas in the indices which define the transition probabilities.
cf. also equations (157, 159 and 160)

I would mention Jensen's inequality to aid the reader at line 463 top of page 26.



line 500 page 29: I really dislike it when authors "prove" a statement with "by [Reference]" as a mere justification. You could write " By a Slepian type Lemma  (c.f. the middle of page 7 in [1] or [4]m pp 76-77), ... " Or even better, write the used result in a separate Lemma.



=========minor typos========

MAIN:

Equation (10) should have a period instead of a comma.
line 78 "homogeneousness" should be "homogeneous"
Line 145 (p 5) "because the dependency" =>"because of the dependency"
Line 149 (p5) "Let \mathcal{H} be a the" ===> see the section 2.5 page 8 of the sup which corresponds to this section and is better written than in the main.
162 (p5) should be "for example, THE sigmoid function)
146 "constant is larger than IN  the i.i.d. case"
165 "Neural networks can" (not "networks works can")
214 (p 8) "we only derive a new bound. Other bounds can be derived in a similar fashion"
==> " We only derive one bound:  equation (45). The other bounds in this paper (such as eq. (46)) can be derived in a similar fashion"

227 "associated" is repeated



SUP


125: "especially, ..." => "In particular, ..."
Footnote at the bottom of page 27: "...change of measure is appeared in..."==> ""...change of measure appeared in..."

499 "and g is standard normal independent" ===> "  "and g is A standard normal independent"

Line 549 "replacing u=... to (294)"  ===> "substituting u=... into equation (294)..."



Lemma 26 page 37 lines 612-613: Improve sentence structure.



=======================================
References:


[1] V. Koltchinskii and D. Pachenko. "Empirical margin distributions and bounding the generalization error of combined classifiers", Annals of stats 2002.

[2] Jianqing Fan, Bai Jiang, Qiang Sun. "Hoeffding's inequality for general Markov chains and applications to statistical learning.", JMLR 2021

[3] Daniel Paulin. "Concentration inequalities for Markov chains by Marton Couplings and spectral methods", electronic journal of Probability 2015.

[4] Ledoux and Talagrand. Probability in Banach spaces.

[5] D. Rudolf. Explicit error bounds for Markov chain Monte Carlo. (arxiv 2011)//Explicit error bounds for lazy reversible Markov chain Monte Carlo,  Journal of Complexity 2009.

[6] Shravas Rao. A Hoeffding inequality for Markov Chains.  Electronic communications in Probability 2019.

[7] Bartlett and Mendelson. "Rademacher and Gaussian Complexities: Risk Bounds and Structural Results" 2002.

[8] Anthony and Bartlett. Neural Network Learning: Theoretical foundations. 1999.

[9] Wolfer and Kontorovich. Estimating the mixing time of ergodic Markov chains. 2019 COLT.





**Strengths And Weaknesses:**

This is a very solid, very high-quality paper overall.

Strengths:
1. The breadth of the results and the systematic approach to the treatment. A lot is covered and it represents a lot of work for the authors that required extreme proficiency with difficult topics. The results are as strong as possible and very general.
2. Applicability: this is a highly fundamental problem and the potential for further research (mostly on specific models) is great. This paper will get cited.


Weaknesses:

1. The readability and presentation could definitely be improved. Several results are just there in the supplementary without any proper explanation of the context. I am thinking of sections 2.2 and 2.3 in the main supplementary in particular. There is nothing about what the results mean.  In addition, the actual main paper is useless without the supplementary unless one is completely familiar with the complex baroque and overly general style of [1].
2. There are notations that are not properly introduced (see "questions"), making readability more problematic. Reader friendliness is, therefore, a definite issue. There are also parts of proofs that follow [1] very closely without disclosing the corresponding result despite it having the potential to make it easier for the reader.
3. I believe there are two MINOR ERRORS, including one with the numerical constants involved in several theorems (see "questions" section), and the proof of what is pretty much the main theorem is far from complete.
4. (Very slight) It is a lot of abominable bookkeeping but I couldn't spot a truly original proof technique (not that this should be grounds for rejection of course).


Regarding 1 and 2, I think it would be helpful to have a graph of dependencies between the sections,  and a table with equivalents from the literature.  Here is a tentative, non exhaustive list of correspondences:

Theorem 5 (p 4) = Theorem 4 in [1], page 12.
Theorem 6 (p4) = Theorem 5 in [1]
Theorem 10 (p6) = THeorem 7 page 24 of [1]
Theorem 3 (proof on page 29) = Theorem 2 in [1]
Lemma 26 p37: cf. Theorem 5 page 16 of [1]





=======================================
References:


[1] V. Koltchinskii and D. Pachenko. "Empirical margin distributions and bounding the generalization error of combined classifiers", Annals of stats 2002.

[2] Jianqing Fan, Bai Jiang, Qiang Sun. "Hoeffding's inequality for general Markov chains and applications to statistical learning.", JMLR 2021

[3] Daniel Paulin. "Concentration inequalities for Markov chains by Marton Couplings and spectral methods", electronic journal of Probability 2015.

[4] Ledoux and Talagrand. Probability in Banach spaces.

[5] D. Rudolf. Explicit error bounds for Markov chain Monte Carlo. (arxiv 2011)//Explicit error bounds for lazy reversible Markov chain Monte Carlo,  Journal of Complexity 2009.

[6] Shravas Rao. A Hoeffding inequality for Markov Chains.  Electronic communications in Probability 2019.

---

> ### Author Response · Authors · 2022-07-28
> **Reply to reviewer's comments**
>
> Thank you very much for your careful reading of our paper. Many suggestions are very useful to us. We will change our paper based on your suggestions, including presentations. For your questions and concerns in the Questions, the following are our answers:
>
> 1. We had a mistake here. The constants in Lemma 26 are non-explicit which is caused by the entropy inequalities for subgaussian processes (van der Vaart and Wellner (1996)). We use the same notation $c$ for two different concepts which lead to the mistake in Lemma 26's statement. In Lemma 26, we should state that $c$ is some constant.
>
> 2. We agree that $64M$ should be changed to $128M$ in the definition of $A_n$.
>
> 3. We had a typo in (298). The definition of $\tilde{f}_y$ should be changed to $1(f(x)\leq y)-P(f(x)\leq y)$,  which explains why $4t/\sqrt{n}$ should appear in Lemma 7.  We missed the second part of this definition, which causes confusion. Sorry for this.
>
> 4. Thank you for your suggestions. We will change the concentration inequalities here. In the first version, we used the pseudo-spectral gap, but later we changed to another one plus the use of a burn-in time to improve the convergence rate, which caused this problem.
>
> 5. We don't fully understand your question. However, as a consequence of this work, by choosing the PAC probability $\frac{1}{\sqrt{n}}$  for example, we can easily draw a result for the expected generalization bound, which is of order $O(\sqrt{\frac{\log n}{n}})$  plus the symmetrization part plus the empirical one. But, we think that the first term may reduce this to  $O(\sqrt{\frac{1}{n}})$ by some tricks.

---

### Meta-Review · Area_Chair_broZ · 2022-08-26

**Recommendation:** Accept
**Confidence:** Less certain

**Metareview:**

The reviewers support acceptance of the manuscript based on the quality of the results, but have expressed concerns about the organisation of the results which appear to be a result of the page length limits imposed on NeurIPS submissions.  There reviewers would welcome publication if these organisational issues were results, but are unsure if the authors will do so.   I would encourage acceptance as the authors have assured the reviewers that Section 3 will be compressed and Section 4 expanded upon so as to emphasise the extension to High-order Markov Chains; clearly Section 4 shouldn't be one paragraph and I appreciate the authors would likely have preferred to put more of the associated supplementary material here.  Potentially abridging the rather length Section 2 would be beneficial, with Section 2 edited to take up far less space as most of (7) to (17) need not be stand alone numbered equations.

**Award:**

No

---

### Decision · Program_Chairs · 2022-09-14

Accept